# FAST TRAINING AND SAMPLING OF RESTRICTED BOLTZMANN MACHINES

**Nicolas Béreux**
INRIA-Saclay, LISN
Paris-Saclay University

**Aurélien Decelle**
Escuela Técnica Superior de Ingenieros Industriales
Universidad Politécnica de Madrid
Departamento de Física Teórica
Universidad Complutense de Madrid

**Cyril Furtlehner**
INRIA-Saclay, LISN
Paris-Saclay University

**Lorenzo Rosset**
LCQB, Sorbonne Université,
Laboratoire de Physique Théorique
École Normale Supérieure, Paris

**Beatriz Seoane**
Departamento de Física Teórica
Universidad Complutense de Madrid

## ABSTRACT

Restricted Boltzmann Machines (RBMs) are powerful tools for modeling complex systems and extracting insights from data, but their training is hindered by the slow mixing of Markov Chain Monte Carlo (MCMC) processes, especially with highly structured datasets. In this study, we build on recent theoretical advances in RBM training and focus on the stepwise encoding of data patterns into singular vectors of the coupling matrix, significantly reducing the cost of generating new samples and evaluating the quality of the model, as well as the training cost in highly clustered datasets. The learning process is analogous to the thermodynamic continuous phase transitions observed in ferromagnetic models, where new modes in the probability measure emerge in a continuous manner. We leverage the continuous transitions in the training process to define a smooth annealing trajectory that enables reliable and computationally efficient log-likelihood estimates. This approach enables online assessment during training and introduces a novel sampling strategy called Parallel Trajectory Tempering (PTT) that outperforms previously optimized MCMC methods. To mitigate the critical slowdown effect in the early stages of training, we propose a pre-training phase. In this phase, the principal components are encoded into a low-rank RBM through a convex optimization process, facilitating efficient static Monte Carlo sampling and accurate computation of the partition function. Our results demonstrate that this pre-training strategy allows RBMs to efficiently handle highly structured datasets where conventional methods fail. Additionally, our log-likelihood estimation outperforms computationally intensive approaches in controlled scenarios, while the PTT algorithm significantly accelerates MCMC processes compared to conventional methods.

## 1 INTRODUCTION

Energy-based models (EBMs) are a long-established approach to generative modeling, with the Boltzmann Machine (BM) (Hinton & Sejnowski, 1983) and Restricted Boltzmann Machine (RBM) (Smolensky, 1986; Ackley et al., 1985) being among the earliest examples. They offer a clear framework for capturing complex data interactions and, with simple energy functions, can reveal underlying patterns, making them especially valuable for scientific applications. Pairwise-interacting models (Nguyen et al., 2017) have been widely used for decades for inference applications in fields like neuroscience (Hertz et al., 2011) and computational biology (Cocco et al., 2018).

More recent research has shown that RBMs can also be reinterpreted as many-body physical models (Decelle et al., 2024; Bulso & Roudi, 2021), analyzed to identify hierarchical relationships in the data (Decelle et al., 2023), or used to uncover biologically relevant constituent features (Tubiana et al., 2019; Fernandez-de Cossio-Diaz, 2025). In essence, RBMs present a great compromise between expressibility and interpretability: they are universal approximators capable of modeling complex datasets and simple enough to allow a direct extraction of insights from data. This interpretative task is particularly challenging when dealing with more powerful models, such as generative convolutional network EBMs (Xie et al., 2016; Song & Kingma, 2021) and other state-of-the-art models in general. At this point, it is also important to stress that RBM are not only interpretable, but they perform particularly well when dealing with tabular datasets with discrete variables and where data are not abundant, which is often the case in genomics/proteomics or neural recording datasets. For example, RBMs have been shown to be among the most reliable approaches to generate synthetic human genomes (Yelmen et al., 2021; 2023) or to model the immune response (Bravi et al., 2021a;b; 2023) or neural activity (van der Plas et al., 2023; Quiroz Monnens et al., 2024).

RBMs, like other EBMs, are challenging to train due to the difficulty of computing the log-likelihood gradient, which requires ergodic exploration of a complex free-energy landscape via Markov Chain Monte Carlo (MCMC). Training with non-convergent MCMC introduces out-of-equilibrium memory effects, as shown in recent studies (Nijkamp et al., 2019; 2020; Decelle et al., 2021), which can be analytically explained using moment-matching arguments (Nijkamp et al., 2019; Agoritsas et al., 2023). While these effects enable fast and accurate generative models for structured data (Carbone et al., 2024a) and high-quality images with RBMs (Liao et al., 2022), they create a sharp disconnect between the model's Gibbs-Boltzmann and dataset distributions, undermining parameter interpretability (Agoritsas et al., 2023; Decelle et al., 2024). For meaningful insights, RBMs require proper chain mixing during training, emphasizing the need for *equilibrium models*.

Working with equilibrium RBMs poses three primary challenges: **training, evaluating model quality, and sampling from the trained model**. Both training and sampling rely on MCMC methods to draw equilibrium samples, but their efficiency depends heavily on the dataset's structure. While RBMs can be effectively trained on image datasets like MNIST or CIFAR-10 with sufficient MCMC steps, this approach falters for highly structured datasets (Béreux et al., 2023), such as genomics, neural recordings, or low-temperature physical systems. These datasets often exhibit multimodal distributions with distinct clusters, as revealed by PCA (Fig. 1), which hinder mixing. Clustering is prominent in the first four datasets studied here, in contrast to the more compact PCA pattern seen in CelebA or full MNIST. Dataset details are provided in Fig. 1. Multimodal distributions exacerbate sampling challenges since mixing times depend on the MCMC chains' ability to transition between clusters. In RBMs, these distant modes emerge during training through continuous phase transitions (e.g., second-order or critical transitions) that encode patterns (Decelle et al., 2017; 2018; Bachtis et al., 2024). These transitions lead to *critical slowdown* (Hohenberg & Halperin, 1977), where mixing times diverge with system size as the transition point nears. Critical slowdown along the training trajectory was recently analyzed in (Bachtis et al., 2024). For a brief introduction to phase transitions and associated dynamical effects, see Supplemental Information (SI) G. All this hampers both the training process and the model's ability to generate independent samples, as achieving adequate mixing requires prohibitively long simulations. Model quality evaluation adds to the challenge, particularly in unsupervised learning. A common qualitative approach compares equilibrium samples to data via PCA projections. Quantitatively, moment comparisons between data and generated distributions offer insights but fail to detect mode collapse. Log-likelihood computation provides a single-score metric and tracks overfitting during training, but exact computation is infeasible due to the intractable partition function. Instead, methods such as Annealed Importance Sampling (AIS) (Neal, 2001) are used for estimation but become unreliable for highly multimodal distributions and are computationally impractical for online evaluations (see Section 4.1).

In this paper, we build on recent advances in understanding the evolution of the landscape during the training of RBM models to propose 3 innovative approaches for tackling the challenges of evaluating, generating samples and training RBMs on highly structured datasets:

1. We propose a novel method for estimating log-likelihood (LL) by leveraging the softness of the training trajectory. This approach enables efficient online LL computation during training at minimal cost or offline reconstruction using trajectory annealing or parallel tempering. Moreover, we demonstrate that this method is not only less computationally expensive than existing techniques but also more accurate in controlled experiments. The method is introduced in Sect. 4.1.

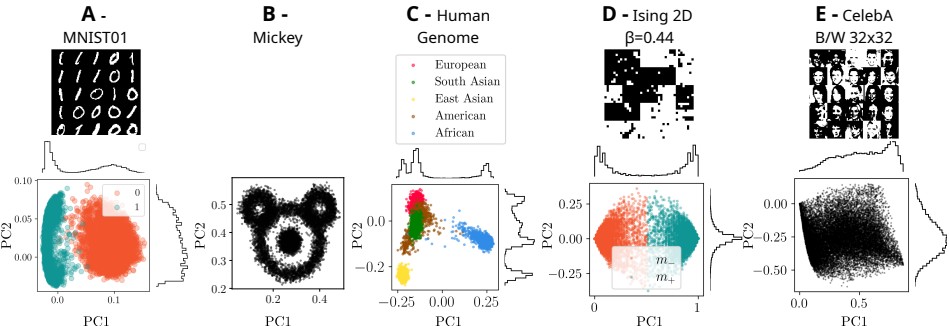

Figure 1: **Datasets.** Panels A-E display 5 distinct datasets projected onto their first two PCA components. In some instances, the dots are color-coded to indicate different labels. In A, the MNIST 01 dataset, featuring images of the digits 0 and 1 from the complete MNIST collection, along with a few sample images. In B, the "Mickey" dataset, an artificial dataset whose PCA representation forms the shape of Mickey Mouse's face. In C, the Human Genome Dataset (HGD), which consists of binary vectors representing mutations or non-mutations for individuals compared to a reference genome across selected genes. In D, the Ising dataset, showcasing equilibrium configurations of the 2D ferromagnetic Ising model at an inverse temperature of $\beta = 0.44$. In E, the CelebA dataset in black and white, resized to 32x32 pixels. For more details on these datasets, please refer to the SI.

2. We propose a variation of the standard parallel tempering algorithm, where parameters from different stages of the learning process are exchanged instead of temperatures. This new sampling approach significantly outperforms previous optimized methods and is detailed in Section 4.2.

3. We introduce a pretraining strategy based on the low-rank RBM framework of (Decelle & Furtlehner, 2021a) to mitigate gradient degradation in early training, caused by mixing time divergence when encoding distant modes in structured datasets. Our approach extends this framework to real datasets by increasing constrained directions and incorporating a trainable bias, essential for image data. We also refine static Monte Carlo sampling for efficient equilibrium sampling. See Section 5 and SI A for details.

We also provide an in-depth discussion in Sect. 2 on the physical reasons behind the failure of previous algorithms when handling clustered datasets.

## 2 RELATED WORK

Training EBMs by maximizing log-likelihood has long been a challenge (LeCun et al., 2006; Song & Kingma, 2021). The introduction of contrastive divergence (Hinton, 2002) popularized EBMs by using short MCMC runs initialized on minibatch examples, but this approach often produces models with poor equilibrium properties, limiting their effectiveness as generative models (Salakhutdinov & Murray, 2008; Desjardins et al., 2010; Decelle et al., 2021). Persistent contrastive divergence (PCD) (Tieleman, 2008) improves gradient estimation by maintaining a continuous MCMC chain between updates, acting like a slow annealing process. However, PCD struggles with clustered data, as the chain's properties can drift from the equilibrium measure, degrading the model (Béreux et al., 2023). This issue, tied to phase coexistence, can be mitigated with constrained MCMC methods, leveraging order parameters from the singular value decomposition of the RBM coupling matrix to reconstruct multimodal distributions (Béreux et al., 2023). While effective for model evaluation, this approach is too computationally intensive for practical training, despite producing models with strong equilibrium properties.

The population annealing algorithm, which adjusts the statistical weights of parallel chains after each parameter update, has been proposed as an alternative (Krause et al., 2018). Similarly, reweighting chains using non-equilibrium physics concepts like the Jarzynski equation has been explored (Carbone et al., 2024b). However, both methods face challenges with highly structured data, where frequent reweighting (duplicating only a few samples) skews the equilibrium measure due to an insufficient number of independent chains. To mitigate this, more sampling steps or lower learning rates are required, leading to excessively long training times to maintain equilibrium.

In our case, applying these methods to our datasets resulted in training that was either too unstable or prohibitively slow when auto-adjusting learning rates and Gibbs steps. Another approach uses EBMs as corrections for flow-based models (Nijkamp et al., 2022), simplifying sampling but sacrificing the interpretability of the energy function, which is central to our work. Alternatively, evolving flow models can act as fast sampling proposers for EBMs (Grenioux et al., 2023), but this requires training two separate networks and leads to lower acceptance rates as the EBM becomes more specialized.

The Parallel Tempering (PT) algorithm  (Hukushima & Nemoto, 1996), in which the model is simulated in parallel at different temperatures which are exchanged once in a while using the Metropolis rule, has been proposed to both speed up the sampling and to improve the training process (Salakhutdinov, 2009; Desjardins et al., 2010) showing important improvements in certain datasets (Krause et al., 2020).

However, PT is expensive because it requires sampling at multiple temperatures while using samples from only one model. It's also often ineffective with highly clustered data due to first-order phase transitions in EBMs, where modes disappear abruptly at certain temperatures, as discussed by (Decelle & Furtlehner, 2021a). This leads to sharp drops in acceptance rates for exchanges between temperatures near the transition. For this reason, first-order transitions typically hinder PT, as many intermediate temperatures are needed to allow sufficient acceptance rates in the exchanges. In contrast, continuous transitions, as seen during the training trajectory, cause modes to gradually separate as the model parameters are changed. The "Stacked Tempering" algorithm (Roussel et al., 2023), recently proposed for RBMs, accelerates sampling by training smaller RBMs with latent variables from previous models, enabling fast updates via a PT-like algorithm. While significantly faster than PT, it is unsuitable for learning, as it requires sequential training of multiple stacked models.

Separately, a theoretical study (Decelle & Furtlehner, 2021a) demonstrated that a low-rank RBM can be trained via a convex, fast optimization process to reproduce the statistics of the data projected along the $d$ first data principal directions through a convex and very fast optimization process (see (Decelle & Furtlehner, 2021a) and the discussion below) using a mapping to another model, the so-called Restricted Coulomb Machine. This low-rank model can be seen as a good approximation to the correct RBM needed to describe the data, and has the nice property that it can be efficiently sampled via a static Monte Carlo process. While this work showed the efficiency of this training process to describe simple synthetic datasets embedded in only one or two dimensions, the use of this algorithm to obtain reliable low-rank RBMs with real data requires further developments that will be discussed in this paper, in particular: (i) the training of a bias, which is crucial to gain almost for free an additional dimension and to deal with image datasets, and (ii) a tuning of the RBM fast sampling process, which was not properly investigated in the original paper.

In this paper, we show that the training trajectory can be exploited to estimate log-likelihood with minimal computational cost, enabling reliable online computation. We also introduce Parallel Trajectory Tempering (PTT), an efficient sampling method that outperforms optimized MCMC approaches like Parallel Tempering (PT) and Stacked Tempering (Roussel et al., 2023), requiring only a few model parameters stored at different stages of the training. Additionally, we demonstrate that initializing RBM training with a low-rank RBM in highly clustered datasets mitigates the effect of initial dynamical arrests, significantly improving model quality at a comparable cost. Combining these strategies allows us to train equilibrium models that capture diverse modes in datasets prone to mode collapse, efficiently evaluate RBMs during training, and sample from multimodal distributions.

## 3   THE RESTRICTED BOLTZMANN MACHINE

The RBM consists of $N_{\mathrm{v}}$ visible nodes and $N_{\mathrm{h}}$ hidden nodes. In our study, we primarily use binary variables $\{0, 1\}$ or $\pm 1$ for both layers. The visible and hidden layers interact through a weight matrix $\boldsymbol{W}$, with no intra-layer couplings. Each variable is further influenced by local biases, $\boldsymbol{\theta}$ for visible units and $\boldsymbol{\eta}$ for hidden units. The Gibbs-Boltzmann distribution for this model is given by

$$p(\boldsymbol{v}, \boldsymbol{h}) = \tfrac{1}{Z} \exp\left[-\mathcal{H}(\boldsymbol{v}, \boldsymbol{h})\right] \text{ where } \mathcal{H}(\boldsymbol{v}, \boldsymbol{h}) = -\sum_{ia} v_i W_{ia} h_a - \sum_i \theta_i v_i - \sum_a \eta_a h_a, \quad (1)$$

where $Z$ is the partition function of the system. As with other models containing hidden variables, the training objective is to minimize the distance between the empirical distribution of the data, $p_{\mathcal{D}}(\boldsymbol{v})$, and the model's marginal distribution over the visible variables, $p(\boldsymbol{v}) = \sum_{\boldsymbol{h}} \exp\left[-\mathcal{H}(\boldsymbol{v}, \boldsymbol{h})\right]/Z = \exp\left[-H(\boldsymbol{v})\right]/Z$. Minimizing the Kullback-Leibler divergence is equivalent to maximizing the

likelihood of observing the training dataset in the model. Thus, the log-likelihood $\mathcal{L} = \langle -H(\boldsymbol{v}) \rangle_{\mathcal{D}} - \log Z$ can be maximized using the classical stochastic gradient ascent. For a training dataset $\mathcal{D} = \{\boldsymbol{v}^{(m)}\}_{m=1,\dots,M}$, the log-likelihood gradient is given by

$$\frac{\partial \mathcal{L}}{\partial W_{ia}} = \langle v_i h_a \rangle_{\mathcal{D}} - \langle v_i h_a \rangle_{\text{RBM}}, \quad \frac{\partial \mathcal{L}}{\partial \theta_i} = \langle v_i \rangle_{\mathcal{D}} - \langle v_i \rangle_{\text{RBM}}, \quad \frac{\partial \mathcal{L}}{\partial \eta_a} = \langle h_a \rangle_{\mathcal{D}} - \langle h_a \rangle_{\text{RBM}}, \quad (2)$$

where $\langle f(\boldsymbol{v}, \boldsymbol{h}) \rangle_{\mathcal{D}} = M^{-1} \sum_m \sum_{\{\boldsymbol{h}\}} f(\boldsymbol{v}^{(m)}, \boldsymbol{h}) p(\boldsymbol{h}|\boldsymbol{v}^{(m)})$ denotes the average with respect to the entries in the dataset, and $\langle f(\boldsymbol{v}, \boldsymbol{h}) \rangle_{\text{RBM}}$ with respect to $p(\boldsymbol{v}, \boldsymbol{h})$.

Since $Z$ is intractable, gradient estimates typically rely on $N_{\text{s}}$ independent MCMC processes, replacing observable averages $\langle o(\boldsymbol{v}, \boldsymbol{h}) \rangle_{\text{RBM}}$ with $\sum_{r=1}^{R} o(\boldsymbol{v}^{(r)}, \boldsymbol{h}^{(r)})/R$, where $(\boldsymbol{v}^{(r)}, \boldsymbol{h}^{(r)})$ are the final states of $R$ parallel chains. Reliable estimates require proper mixing before each parameter update, but enforcing equilibrium at every step is impractical and slow. The widespread use of non-convergent MCMC processes underlies most training difficulties and anomalous dynamics in RBMs, as discussed in (Decelle et al., 2021). In order to minimize out-of-equilibrium effects, it is often useful to keep $R$ *permanent (or persistent)* chains, which means that the last configurations reached with the MCMC process used to estimate the gradient at a given parameter update $t$, $\boldsymbol{P}_t \equiv \{(\boldsymbol{v}_t^{(r)}, \boldsymbol{h}_t^{(r)})\}_{r=1}^R$, are used to initialize the chains of the subsequent update $t + 1$. This algorithm is typically referred to as PCD. In this scheme, the process of training can be mimicked as a slow cooling process, only that instead of varying a single parameter, e.g. the temperature, a whole set of parameters $\boldsymbol{\Theta}_t = (\boldsymbol{w}_t, \boldsymbol{\theta}_t, \boldsymbol{\eta}_t)$ is updated at every step to $\boldsymbol{\Theta}_{t+1} = \boldsymbol{\Theta}_t + \gamma \boldsymbol{\nabla} \mathcal{L}_t$ with $\boldsymbol{\nabla} \mathcal{L}_t$ being the gradient in equation 2 estimated using the configurations in $\boldsymbol{P}_t$, and $\gamma$ being the learning rate.

Typical MCMC mixing times in RBMs are initially short but increase as training progresses (Decelle et al., 2021). Studies reveal sharp mixing time peaks when the model encodes new modes (Bachtis et al., 2024), a phenomenon known as *critical slowing down* Hohenberg & Halperin (1977), associated with critical phase transitions. In RBMs, these transitions are related with the alignment of the singular vectors of $\boldsymbol{W}$ with the dataset's principal components (Decelle et al., 2017; 2018; Decelle & Furtlehner, 2021b; Bachtis et al., 2024). However, these dominant modes can be identified via PCA before training and efficiently encoded through convex optimization, circumventing standard maximum likelihood training. This approach prevents gradient degradation due to diverging mixing times at the first second-order transitions. The advantages of using the low-rank construction from Decelle & Furtlehner (2021a) as a pretraining method will be discussed in Section 5.

## 4 TRAJECTORY ANNEALING PARADIGM

One major challenge with structured datasets is quantifying the model's quality, since sampling the equilibrium measure of a well-trained model is often too time-consuming. This affects the reliability of generated samples and indirect measures such as log-likelihood (LL) estimation through Annealing Importance Sampling (AIS) (Krause et al., 2020), making them inaccurate and meaningless. The evolution of parameters during training can be seen as a smoother annealing process compared to the standard temperature annealing used in algorithms like annealing importance sampling (AIS) for LL estimation or in sampling algorithms like simulated annealing or PT. This is because it avoids the first-order transitions that typically occur with temperature in clustered datasets, see SI H for more detailed explanations of this phenomenon in clustered datasets. However, these methods can still be applied to trajectory annealing using the parameters of different stages of training, rather than scaling the model parameters by a common factor $\beta_i$, as is done in temperature-based protocols.

### 4.1 TRAJECTORY ANNEALING IMPORTANCE SAMPLING

The standard method for log-LL estimation in the literature is AIS (Neal, 2001); see (Krause et al., 2020) for a review on its application to RBMs. A detailed algorithm description is provided in SI D. AIS estimates the partition function of a model with parameters $\boldsymbol{\Theta} = \{\boldsymbol{W}, \boldsymbol{\theta}, \boldsymbol{\eta}\}$ by gradually increasing the inverse temperature $\beta$ from 0 to 1 over $N_\beta$ steps. At $\beta = 0$, where $p_0 = p_{\text{ref}}$, the partition function $Z_0$ is tractable, allowing estimation of $Z_i$ at neighboring $\beta_i$ values via path sampling. The reference distribution $p_{\text{ref}}$ is typically uniform, but improved performance has been reported when using the independent site distribution centered on the dataset, see SI D for details.

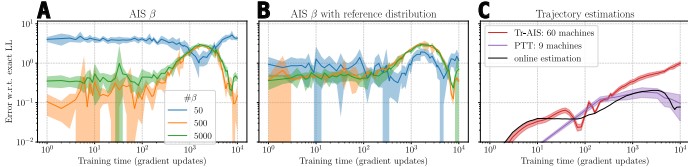

Figure 2: Comparison of LL estimation error (relative to the exact value) across different methods: AIS (A), AIS with a reference distribution fixed to the independent site distribution matching the dataset's empirical center (B), and Tr-AIS (ours, C). For Tr-AIS, we evaluate three approaches: online during training, offline using saved models, and with PTT. The RBM, pretrained and trained with 20 hidden nodes (allowing exact LL computation), for $10^4$ gradient steps on the HGD dataset. PTT selects a subset of saved models ensuring a 0.25 acceptance rate between consecutive models. Lines show the mean LL over 10 independent runs, with shaded areas representing one standard deviation.

AIS is quite costly because it requires a large number of annealing steps $N_\beta$ for reliable estimations, making it impractical to compute the log-likelihood online during training. As a result, it is often calculated offline using a set of models at different temperatures. However, since the training process itself follows an annealing trajectory, it makes more sense to use the evolution of parameters during training as annealing steps rather than relying on temperature. Additionally, at the start of training, parameters are set to zero, making $Z_0$ tractable, which also holds when initializing with a low-rank RBM. Since updating the partition function only involves calculating the energy difference between the new and old parameters, this can be efficiently computed online during each parameter update, allowing for very small integration steps. We refer to this method as *online* Tr-AIS. Alternatively, similar to standard AIS, multiple models can be saved during training for an *offline* Tr-AIS estimation.

To evaluate the reliability of our new approach, we trained several RBMs using datasets with a maximum of 20 hidden nodes, allowing for the exact computation of the partition function through direct enumeration. We then calculated the difference between the true partition function and the estimations obtained from various methods. The results for the HGD dataset are presented in Fig. 2, while similar qualitative results for the other datasets can be found in the SI, see Figs. 15, 16, 17 and 18. In Fig. 2, we display the estimation errors in the LL using three different methods: standard temperature AIS (panel A), AIS with non-flat reference distribution (panel B), and Tr-AIS (panel C). For the temperature AIS estimates, we show results for different values of $N_\beta$ in distinct colors. In panel C, we compare the online estimation in black with the offline Tr-AIS estimation, using the same models analyzed in panels A and B. Additionally, we present the error in LL estimation using just a few machines saved during the training and the trajectory parallel tempering (PTT), which will be detailed in the next section. Overall, the online Tr-AIS estimation consistently outperforms the other methods, and only the PTT estimation yields similar results in offline analyses.

## 4.2 EXPLOITING THE TRAINING TRAJECTORY FOR SAMPLING

The trajectory annealing paradigm can also be exploited to significantly accelerate the sampling of equilibrium configurations, as we show in Fig. 3. Before presenting our parallel trajectory annealing (PTT) algorithm, we first discuss the limitations of standard methods like Alternating Gibbs Sampling (AGS), and introduce the standard parallel tempering (PT) algorithm. The AGS procedure involves iteratively alternating between two steps: conditioning on all hidden variables given fixed visible variables, and then conditioning on all visible variables given fixed hidden variables.

**Problems with the standard AGS method–** Let us consider the MNIST01, HGD, and Ising 2D datasets. MNIST01 and Ising 2D exhibit strong bimodality in their PCA representations, while HGD is trimodal, as shown in Fig. 1. Suppose that we have trained an RBM on each of these datasets, for example using the low-rank RBM pretraining (see Section 5), and we aim to sample the equilibrium measure of these models. Typically, this involves running MCMC processes from random initializations and iterating them until convergence. The mixing time in such cases is determined by the jumping time between clusters. Accurate estimation of the relative weights between modes requires the MCMC processes to be ergodic, necessitating frequent back-and-forth jumps. However, as shown in the chains in Figs. 3–B1-3, AGS dynamics are exceedingly slow, rarely producing jumps

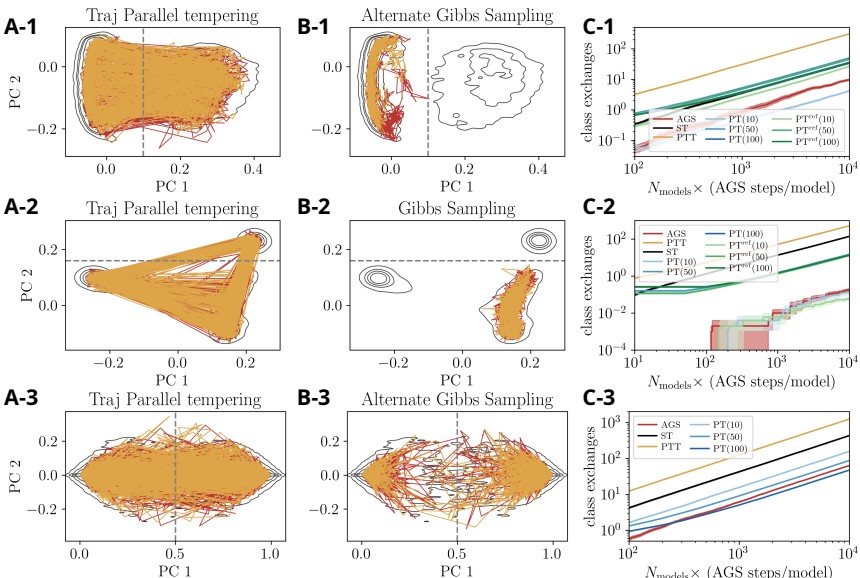

Figure 3: Comparison of the performance of different MCMC sampling methods on RBMs trained with the pretraining+PCD procedure on the MNIST01 (row-1), the HGD (row-2) and Ising 2D datasets (row-3). In A and B columns, we show the trajectory of two independent Markov chains (red and orange) iterated for $10^4$ MCMC steps using either PTT or AGS, projected onto the first two principal components of the dataset. The position of the chains is plotted every 10 MCMC steps. The black contour represents the density profile of the dataset. In column C, we show the averaged number of jumps between the two regions separated by the dashed grey line in the PCA plots using different MCMC methods: AGS, Parallel Tempering (PT) (Hukushima & Nemoto, 1996), without or with reference configuration (PTref) (Krause et al., 2020), the Stacked Tempering (Roussel et al., 2023), and the Trajectory Parallel Tempering (PTT) proposed in this work. The average is calculated over a population of 1000 chains and the shadow around the lines indicates the error of the mean.

even after $10^4$ MCMC steps. The red curves in Figs. 3–C1-3 illustrate the mean number of jumps over 1000 independent chains as a function of MCMC steps, indicating that achieving proper equilibrium generation would require at least $10^6$-$10^7$ MCMC steps. To accelerate sampling, we therefore explore optimized Monte Carlo methods inspired by Parallel Tempering.

**Trajectory Parallel Tempering–** Parallel tempering (PT) is used to simulate multiple systems in parallel at different inverse temperatures $\beta$ (Hukushima & Nemoto, 1996; Marinari & Parisi, 1992), with occasional temperature swaps between models based on the Metropolis rule. Since this move satisfies detailed balance, the dynamics will converge to the Boltzmann distribution $p_\beta(\boldsymbol{x}) \propto \exp(-\beta\mathcal{H}(\boldsymbol{x}))$ at each $\beta$. However, a more general approach involves simulating multiple systems with distincts set of parameters. For a set of parameters $\boldsymbol{\Theta}^t$ we associate the system $p_{\boldsymbol{\Theta}^t}(\boldsymbol{x})$. This includes the classic PT case, with $\boldsymbol{\Theta}^t_{\text{PT}} = \beta(t)\boldsymbol{\Theta}$ where the temperature ladder is typically defined as $\beta(t) = t/N_T$, where $t = 0, \ldots, N_T - 1$, and $N_T$ represents the number of systems simulated in parallel. Alternatively, recent work (Roussel et al., 2023) introduced a variant of this algorithm called Stacked Tempering, which uses a series of nested RBMs, where the hidden configurations of one machine are exchanged with the visible configurations of the next in the stack, analogous to the mechanism in deep belief networks.

We introduce a novel variant where model parameters from different training phases are exchanged. Using the learning trajectory, we define the parameter set $\boldsymbol{\Theta}^t$, with $t$ indexing models at specific training steps. We call this method **Parallel Trajectory Tempering (PTT)**. Configuration exchanges $\boldsymbol{x} = (\boldsymbol{v}, \boldsymbol{h})$ between neighboring models at $t$ and $t-1$ follow the rule:

$$p_{\text{acc}}(\boldsymbol{x}^t \leftrightarrow \boldsymbol{x}^{t-1}) = \min\left[1, \exp\left(\Delta\mathcal{H}^t(\boldsymbol{x}^t) - \Delta\mathcal{H}^t(\boldsymbol{x}^{t-1})\right)\right] \text{ with } \Delta\mathcal{H}_t(\boldsymbol{x}) \equiv \mathcal{H}_t(\boldsymbol{x}) - \mathcal{H}_{t-1}(\boldsymbol{x}), \quad (3)$$

where again the index $t$ indicates that the Hamiltonian is evaluated using the parameters $\mathbf{\Theta}^t$. This move satisfies detailed balance with our target equilibrium distribution $p(\boldsymbol{x}) = \exp(-\mathcal{H}(\boldsymbol{x}))/Z$, ensuring that the moves lead to the same equilibrium measure. The reason for using the trajectory is that PT-like algorithms work very well when a system undergoes a continuous phase transition. As discussed in the SI G, adjusting the inverse temperature can lead to discontinuous transitions for which PT is not known to perform very well, while following the trajectory, the system undergoes several continuous transitions as discussed in (Decelle et al., 2018; Bachtis et al., 2024). The pseudocodes for the standard PT and our new PTT algorithms are provided in SI B.1.

Figs. 3–A1-3 show the evolution of two independent Markov chains on the most trained model using the PTT sampling, showing a significant increase in the number of jumps compared to Figs. 3–B1-3. We also compare the number of mode jumps observed with PTT with those of other algorithms, as a function of the total number of AGS steps performed. For the PT algorithm, we test different values of $N_T$ and include the variant with a reference configuration (Krause et al., 2020). For Stacked Tempering, we replicate the method of (Roussel et al., 2023). The total AGS steps are taken as the product of the number of parallel models and the AGS steps per model, as shown in Fig. 3–C1-3. PTT consistently outperforms all other algorithms in all datasets.

Our PTT approach leverages the ease of sampling configurations from an initial RBM. During classical PCD training, early "non-specialized" models thermalize quickly, behaving like Gaussian distributions. The same is true for RBMs pre-trained from the low-rank RBM of Decelle & Furtlehner (2021a), as independent configurations for the first machine can be efficiently sampled by a static Monte Carlo algorithm (see SI A.1). The trajectory flow greatly accelerates equilibrium convergence, particularly in multimodal structures. The time interval between successive models is set to maintain an acceptance probability for exchanges (Eq. 3) around 0.25. Notably, the total time required to thermalize a PTT chain (AGS steps × number of machines) is unaffected by acceptance values within [0.2, 0.5] (see SI B.2 for details). In practice, effective models (i.e. the parameters of the model) can be saved during training by monitoring the acceptance rate between the last saved model and the current one. Pre-trained models require far fewer saved models, as many are naturally positioned near major phase transitions. The number of models used for each sampling process, along with a detailed description of the algorithm, is provided in the SI. PTT offers the advantage of requiring no extra training (unlike Stacked Tempering) or unnecessary temperature simulations and can be fully applied to other energy-based models. Instead, the equilibrium samples obtained from different models during sampling can be used to compute the log-likelihood, similar to the standard PT approach (Krause et al., 2020). In Fig. 2–right we show that the quality of this estimation is comparable with the online trajectory AIS. Another desirable advantage of the PT algorithms is that one can easily check the thermalisation of the MCMC process by investigating the ergodicity of the random walk in models (Banos et al., 2010), as shown in SI B.2. In addition, we evaluated other indicators of sample quality and potential overfitting in SI I. Our method consistently provides high-quality samples across all datasets and, thanks to precise likelihood computation, can effectively detect overfitting.

**On the failure of thermal algorithms–** Thermal sampling algorithms (such as Simulated Annealing or PT) have difficulties in efficiently sampling models that describe structured data sets. This is mainly because in these types of models, remote modes can simply disappear or appear discontinuously when the temperature is slightly changed. This phenomenon is known in physics as a discontinuous or first-order phase transition. For a detailed introduction to this effect and an illustration of its occurrence when changing the temperature in an RBM toy model, see SI H. The presence of this or these temperature transitions can affect the performance of the PT algorithm, as a change between two neighboring temperatures is rarely accepted (and therefore much denser temperature ladders are required), but in multimodal models the problem can be even more harmful and difficult to detect: PT can even forget configurations in certain minority clusters while maintaining high swap acceptances. We illustrate this problem in Fig. 4-A, where we compare the equilibrium distribution $p(\boldsymbol{m})$ of a well-trained machine trained with the HGD dataset when sampled at two different temperatures $\beta = 0.9$ and $\beta = 1$. This marginal distribution can be efficiently calculated using the tethered MC algorithm Béreux et al. (2023). We can see that from $\beta = 1$ to $\beta = 0.9$ the East Asian cluster (the upper right cluster) disappears from the measure. Since the jump to this cluster is strongly suppressed with AGS, as shown in Fig. 3-C2, simulating the system at higher temperatures does not help to find this cluster more easily, but prevents it from being found with a standard 1-temperature swap + 1AGS scheme (because it will be never nucleated at low temperatures). Indeed, in 4-B we show the distribution of samples generated at $\beta = 1$ by a long PT sampling run (with all swap acceptances

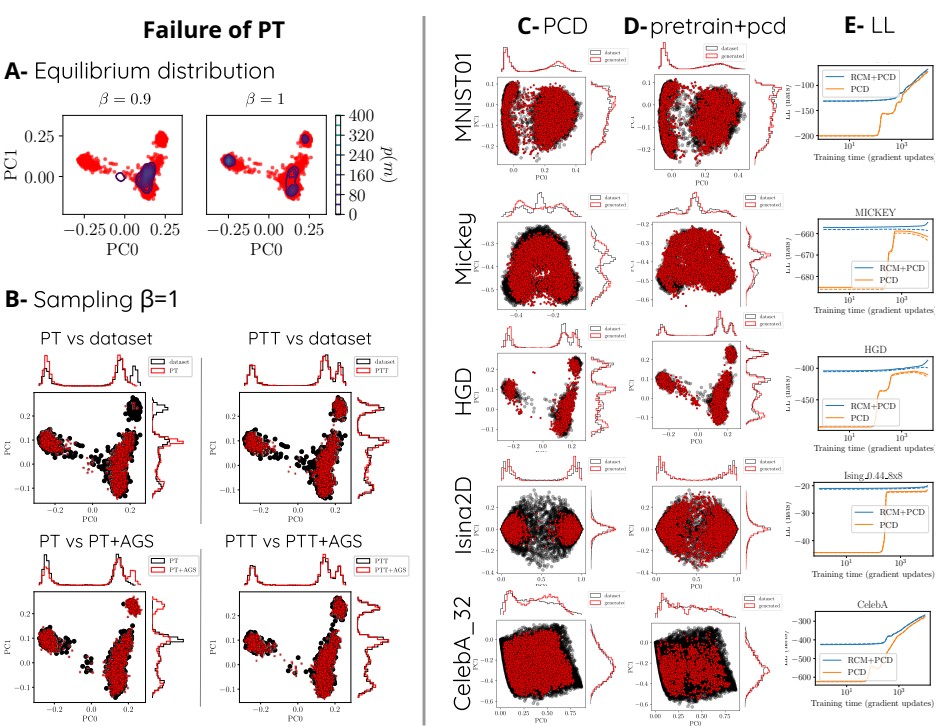

Figure 4: **On the left:** Panel A compares the marginal distribution $p(m)$ along the first two principal components for the pretrained+PCD RBM on the HGD dataset at inverse temperatures $\beta = 1$ and $\beta = 0.9$. Panel B presents sampling results at $\beta = 1$ using PT, PTT, and $10^6$ AGS steps, initialized from configurations generated by both algorithms. **Right panels:** Panels C and D compare equilibrium samples from RBMs trained with PCD alone vs. PCD initialized on low-rank RBMs for MNIST01, Mickey, HGD, Ising2D, and CelebA. Scatter plots and histograms of projections onto the first two principal components show data (black) vs. generated samples (red). Panel E tracks log-likelihood evolution (train: solid, test: dashed) using online Tr-AIS. Results, averaged over 10 low-rank RBM initializations, show minimal variance, as indicated by narrow shaded regions.

above 0.8) completely forgetting the Asian cluster. One might wonder if this is indeed the right balance between clusters learned by the model, so we checked it by performing a long ($10^6$ AGS steps) run initialized with these PT-generated samples, finding that the missing clusters gradually resettle. Instead, the PTT samples correctly identify all the clusters and the sampled distribution statistics remain unchanged when a normal AGS run is re-run from that initialisation.

## 5 THE LOW-RANK RBM PRETRAINING

In (Decelle & Furtlehner, 2021a), it was shown that an RBM with only a few modes in its $\boldsymbol{W}$ can be trained precisely through a mapping to a Restricted Coulomb Machine and solving a convex optimization problem (see SI A.2). This approach enables training RBMs with simplified $\boldsymbol{W}$:

$$\boldsymbol{W} = \sum_{\alpha=1}^{d} w_\alpha \bar{\boldsymbol{u}}_\alpha \boldsymbol{u}_\alpha^\top, \qquad \text{with} \qquad (\boldsymbol{u}_\alpha, \bar{\boldsymbol{u}}_\alpha) \in \mathbb{R}^{N_{\mathrm{v}}} \times \mathbb{R}^{N_{\mathrm{h}}}, \qquad (4)$$

where $\{w_\alpha\}$ are the singular values of the coupling matrix and the right singular vectors $\{\boldsymbol{u}_\alpha\}_{\alpha=1}^{d}$ correspond exactly to the first $d$ principal directions of the data set. Under this assumption, $p(\boldsymbol{v})$ is a function of only $d$ order parameters, given by the *magnetizations* along each of the $\boldsymbol{u}_\alpha$ components, i.e. $m_\alpha(\boldsymbol{v}) = \boldsymbol{u}_\alpha \cdot \boldsymbol{v}/\sqrt{N_{\mathrm{v}}}$. In particular,

$$H(\boldsymbol{v}) = -\sum_{\alpha=0}^{d} \theta_\alpha m_\alpha - \sum_a \log \cosh\left(\sqrt{N_{\mathrm{v}}} \sum_{\alpha=1}^{d} \bar{u}_{\alpha a} w_\alpha m_\alpha + \eta_a\right) = \mathcal{H}(\boldsymbol{m}(\boldsymbol{v})), \quad (5)$$

where $\boldsymbol{m} = (m_1, \ldots, m_\alpha)$ and $\theta_\alpha$ is the projection of the visible bias onto the extended $\boldsymbol{u}$ matrix. To initialize the visible bias as proposed in (Montavon & Müller, 2012), we add an additional direction

$u_0$ associated with the magnetization $m_0$ generated by the part of the visible bias that is not contained in the intrinsic space of $\boldsymbol{W}$. This is an improvement over (Decelle & Furtlehner, 2021a) because it effectively adds an additional direction at minimal cost. We also found that the addition of the bias term is crucial to obtain reliable low-rank RBMs for image data. The optimal parameters are obtained by solving a convex optimization problem (SI A.2). This yields a probability distribution $p(\boldsymbol{m})$ in a lower-dimensional space than $p(\boldsymbol{v})$, which can be efficiently sampled using inverse transform sampling. However, discretizing the $\boldsymbol{m}$-space limits practicality to intrinsic dimensions $d \leq 4$ due to memory constraints. These low-rank RBMs replicate dataset statistics projected onto the first $d$ principal components and, despite their simplicity, can approximate the dataset, as demonstrated in Supplemental Figure 19 for the 5 datasets.

At the start of RBM training, the model encodes the strongest PCA components through multiple critical transitions (Decelle et al., 2017; 2018; Bachtis et al., 2024). Pretraining with a low-rank construction bypasses these transitions, avoiding out-of-equilibrium effects from critical slowing down, which becomes prohibitive in highly clustered datasets. Once the main directions are incorporated, mixing times stabilize at much lower values, as barriers between clusters shrink when modes align properly Béreux et al. (2023), making it a more reliable starting point for PCD training. However, since mixing times still increase over training, pretraining primarily mitigates poor initialization effects, while issues from insufficient MCMC steps may reappear later. Bypassing these transitions significantly improves model quality and data statistics reproduction, especially for short training times. In Fig. 4, we compare equilibrium samples from two RBMs trained on five datasets with identical settings (see SI C) but different strategies: one trained from scratch using standard PCD (Tieleman, 2008), the other using PCD after low-rank RBM pretraining. We also tested Jarzynski reweighting (Carbone et al., 2024a) (see SI E), but results were unstable on most datasets and are omitted. Across all datasets, pretraining improves sample quality and test log-likelihood. The pre-trained+PCD RBM balances modes effectively, as confirmed by PCA projections, and consistently achieves higher log-likelihood values using online Tr-AIS. Samples are drawn with the PTT algorithm, both described in Section 4.1. Scatter plots highlight the importance of pretraining for balanced models in clustered datasets, significantly improving test log-likelihood. However, for less clustered datasets like CelebA and full MNIST, pretraining offers little benefit, as splitting distributions into isolated modes is unnecessary (see Fig. 20 in SI J).

## 6   Conclusions

In this work, we address the challenges of training, evaluating, and sampling high-quality equilibrium Restricted Boltzmann Machines (RBMs) on highly structured datasets. Our primary focus is on improving the sampling process and accurately estimating the log-likelihood during training. We introduce a novel approach that leverages the progressive feature learning inherent in RBM training to enhance both evaluation and sampling. This method facilitates precise, online computation of log-likelihood and enables an efficient sampling strategy, overcoming the limitations of the standard parallel tempering algorithm. Additionally, our results indicate that for datasets with tightly clustered structures in their principal component analysis (PCA) representation, incorporating a pre-training phase can improve training outcomes. This pre-training involves encoding the principal directions of the data into the model through a convex optimization technique. However, this RBM-specific pre-training method shows limited advantages for less structured datasets or extended training durations, where chains deviate from equilibrium. In the latter case, the presented training approach should be combined in the future with optimized MCMC methods to achieve its maximum capacity.

### Code availability

The code and datasets are available at `https://github.com/DsysDML/fastrbm`

### Acknowledgments

Authors acknowledge financial support by the Comunidad de Madrid and the Complutense University of Madrid through the Atracción de Talento program (Refs. 2019-T1/TIC-13298 & Refs. 2023-5A/TIC-28934), the project PID2021-125506NA-I00 financed by the "Ministerio de Economía y Competitividad, Agencia Estatal de Investigación" (MICIU/AEI/10.13039/501100011033), the Fondo Europeo de Desarrollo Regional (FEDER, UE) and the French ANR grant Scalp (ANR-24-CE23-1320).

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

# Supplemental Information

## A    DETAILS OF THE PRE-TRAINING OF A LOW RANK RBM

### A.1    THE LOW-RANK RBM AND ITS SAMPLING PROCEDURE

Our goal is to pre-train an RBM to directly encode the first $d$ principal modes of the dataset in the model's coupling matrix. This approach avoids the standard procedure of progressively encoding these modes through a series of second-order phase transitions, which negatively impact the quality of gradient estimates during standard training. It also helps prevent critical relaxation slowdown of MCMC dynamics in the presence of many separated clusters.

Given a dataset, we want to find a good set of model parameters ($\boldsymbol{W}$, $\boldsymbol{\theta}$ and $\boldsymbol{\eta}$) for which the statistics of the generated samples exactly match the statistics of the data projected onto the first $d$ directions of the PCA decomposition of the training set. Let us call each of these $\alpha = 1, \dots, d$ projections $m_\alpha = \boldsymbol{u}_\alpha \cdot \boldsymbol{v}/\sqrt{N_\mathrm{v}}$ the *magnetizations* along the mode $\alpha$, where $\boldsymbol{u}_\alpha$ is the $\alpha$-th mode of the PCA decomposition of the dataset. A simple way to encode these $d$-modes is to parameterize the $w$-matrix as:

$$\boldsymbol{W} = \sum_{\alpha=1}^{d} w_\alpha \bar{\boldsymbol{u}}_\alpha \boldsymbol{u}_\alpha^\top, \qquad \text{with} \qquad (\boldsymbol{u}_\alpha, \bar{\boldsymbol{u}}_\alpha) \in \mathbb{R}^{N_\mathrm{v}} \times \mathbb{R}^{N_\mathrm{h}}, \tag{6}$$

where $\boldsymbol{u}$ and $\hat{\boldsymbol{u}}$ are respectively the right-hand and left-hand singular vectors of $\boldsymbol{W}$, the former being directly given by the PCA, while $w_\alpha$ are the singular values of $\boldsymbol{W}$. Using this decomposition, the marginal energy on the visible variables, $\mathcal{H}(\boldsymbol{v}) = \log \sum_{\boldsymbol{h}} \exp \mathcal{H}(\boldsymbol{v}, \boldsymbol{h})$ can be rewritten in terms of these magnetizations $\boldsymbol{m} \equiv (m_1, \dots, m_d)$

$$\mathcal{H}(\boldsymbol{v}) = -\sum_a \log \cosh \left( \sqrt{N_\mathrm{v}} \bar{u}_a \sum_{\alpha=1}^{d} w_\alpha m_\alpha + \eta_a \right) = \mathcal{H}(\boldsymbol{m}(\boldsymbol{v})). \tag{7}$$

Now, the goal of our pre-training is not to match the entire statistics of the dataset, but only the marginal probability of these magnetizations. In other words, we want to model the marginal distribution

$$p_\mathrm{emp}(\boldsymbol{m}) \equiv \sum_{\boldsymbol{v}} p_\mathrm{emp}(\boldsymbol{v}) \prod_{\alpha=1}^{d} \delta \left( m_\alpha - \frac{1}{\sqrt{N_\mathrm{v}}} \boldsymbol{u}_\alpha^T \boldsymbol{v} \right), \tag{8}$$

where $\delta$ is the Dirac $\delta$-distristribution. In this formulation, the distribution of the model over the magnetization $\boldsymbol{m}$ can be easily characterized

$$p(\boldsymbol{m}) = \frac{1}{Z} \sum_{\boldsymbol{v}} e^{-\mathcal{H}(\boldsymbol{v})} \prod_{\alpha=1}^{d} \delta \left( m_\alpha - \frac{1}{\sqrt{N_\mathrm{v}}} \boldsymbol{u}_\alpha^T \boldsymbol{v} \right) \tag{9}$$

$$= \frac{1}{Z} \mathcal{N}(\boldsymbol{m}) \exp \sum_a \log \cosh \left( \bar{u}_a \sum_{\alpha=1}^{d} w_\alpha m_\alpha + \eta_a \right) \tag{10}$$

$$= \frac{1}{Z} e^{-\mathcal{H}(\boldsymbol{m}) + N_\mathrm{v} S(\boldsymbol{m})} = \frac{1}{Z} e^{-N_\mathrm{v} f(\boldsymbol{m})} \tag{11}$$

where $\mathcal{N}(\boldsymbol{m}) = \sum_{\boldsymbol{v}} \prod_{\alpha=1}^{d} \delta \left( m_\alpha - \frac{1}{\sqrt{N_\mathrm{v}}} \boldsymbol{u}_\alpha^T \boldsymbol{v} \right)$ is the number of configurations with magnetizations $\boldsymbol{m}$, and thus $S(\boldsymbol{m}) = \log N(\boldsymbol{m})/N_\mathrm{v}$ is the associated entropy. Now, for large $N_\mathrm{v}$ the entropic term can be determined using large deviation theory, and in particular the Gärtner-Ellis theorem:

$$p_\mathrm{prior}(\boldsymbol{m}) = \frac{e^{N_\mathrm{v} S(\boldsymbol{m})}}{2^{N_\mathrm{v}}} \approx \exp \left( -N_\mathrm{v} \mathcal{I}(\boldsymbol{m}) \right), \tag{12}$$

with the rate function

$$\mathcal{I}(\boldsymbol{m}) = \sup_{\boldsymbol{\mu}} \left[ \boldsymbol{m}^T \boldsymbol{\mu} - \phi(\boldsymbol{\mu}) \right] = \boldsymbol{m}^T \boldsymbol{\mu}^* - \phi(\boldsymbol{\mu}^*), \tag{13}$$

and

$$\phi(\boldsymbol{\mu}) = \lim_{N_{\mathrm{v}} \to \infty} \frac{1}{N_{\mathrm{v}}} \log \left\langle e^{N_{\mathrm{v}} \boldsymbol{m}^T \boldsymbol{\mu}} \right\rangle = \lim_{N_{\mathrm{v}} \to \infty} \frac{1}{N_{\mathrm{v}}} \log \frac{1}{2^{N_{\mathrm{v}}}} \sum_{\boldsymbol{v}} e^{\sqrt{N_{\mathrm{v}}} \sum_{\alpha=1}^{d} \mu_\alpha \sum_i u_{\alpha,i} v_i} \tag{14}$$

$$= \lim_{N_{\mathrm{v}} \to \infty} \frac{1}{N_{\mathrm{v}}} \sum_{i=1}^{N_{\mathrm{v}}} \log \cosh \left( \sqrt{N_{\mathrm{v}}} \sum_{\alpha=1}^{d} \mu_\alpha u_{\alpha,i} \right). \tag{15}$$

Then, given a magnetization $\boldsymbol{m}$, we can compute the minimizer $\boldsymbol{\mu}^*(\boldsymbol{m})$ of $\phi(\mu) - \boldsymbol{m}^T \mu$ which is convex, using e.g. Newton method which converge really fast since we are in small dimension. Note that in practice we will obviously use finite estimates of $\phi$, assuming $N_{\mathrm{v}}$ is large enough. As a result we get $\boldsymbol{\mu}^*(\boldsymbol{m})$ satisfying implicit equations given by the constraints given at given $N_{\mathrm{v}}$:

$$m_\alpha = \frac{1}{\sqrt{N_{\mathrm{v}}}} \sum_{i=1}^{N_{\mathrm{v}}} u_i^\alpha \tanh \left( \sqrt{N_{\mathrm{v}}} \sum_{\beta=1}^{d} u_i^\beta \mu_\beta^* \right). \tag{16}$$

It is then straightforward to check that spins distributed as

$$p_{\mathrm{prior}}(\boldsymbol{v}|\boldsymbol{m}) \propto e^{N_{\mathrm{v}} \boldsymbol{\mu}^{*T} \boldsymbol{m}(\boldsymbol{v})} \tag{17}$$

fulfill well the requirement, as $\left\langle \boldsymbol{u}_\alpha^T \boldsymbol{v} / \sqrt{N_{\mathrm{v}}} \right\rangle_{p_{\mathrm{prior}}} = m_\alpha$. In other words, we can generate samples having mean magnetization $m_\alpha$ just by choosing $v_i$ as

$$p_{\mathrm{prior}}(v_i = 1|\boldsymbol{m}) = \mathrm{sigmoid} \left( 2\sqrt{N_{\mathrm{v}}} \sum_{\alpha=1}^{d} u_{\alpha,i} \mu_\alpha^*(\boldsymbol{m}) \right) \tag{18}$$

The training can therefore be done directly in the subspace of dimension $d$. In Ref. (Decelle & Furtlehner, 2021a), it has been shown that such RBM can be trained by mean of the Restricted Coulomb Machine, where the gradient is actually convex in the parameter's space. It is then possible to do a mapping from the RCM to the RBM to recover the RBM's parameters. In brief, the training of the low-dimensional RBM is performed by the RCM, and then the parameters are obtained via a direct relation between the RCM and the RBM's parameters. The detail of the definition and of the training of the RCM is detailed in the appendix A.2.

## A.2 THE RESTRICTED COULOMB MACHINE

As introduced in (Decelle & Furtlehner, 2021a), it is possible to precisely train a surrogate model for the RBM, called the Restricted Coulomb Machine (RCM), on a low dimensional dataset without explicitly sampling the machine allowing to learn even heavily clustered datasets.

The RCM is an approximation of the marginal distribution of the RBM with $\{-1, 1\}$ binary variables:

$$\mathcal{H}(\boldsymbol{v}) = -\sum_i v_i \theta_i - \sum_a \log \cosh \left( \sum_i W_{ia} v_i + \eta_a \right). \tag{19}$$

We then project both the parameters and variables of the RBM on the first $d$ principal components of the dataset:

$$m_\alpha := \frac{1}{\sqrt{N_{\mathrm{v}}}} \sum_{i=1}^{N_{\mathrm{v}}} v_i u_{i\alpha}, \quad w_{\alpha a} := \sum_{i=1}^{N_{\mathrm{v}}} W_{ia} u_{i\alpha}, \quad \theta_\alpha := \frac{1}{\sqrt{N_{\mathrm{v}}}} \sum_{i=1}^{N_{\mathrm{v}}} \theta_i u_{i\alpha} \tag{20}$$

with $\alpha \in \{1, \ldots, d\}$ and $\boldsymbol{u}$ the projection matrix of the PCA. The projected distribution of the model is then given by

$$p_{\mathrm{RBM}}(\boldsymbol{m}) = \frac{\exp \left( N_{\mathrm{v}} \left[ \mathcal{S}(\boldsymbol{m}) + \sum_{\alpha=1}^{d} \theta_\alpha m_\alpha + \frac{1}{N_{\mathrm{v}}} \sum_{a=1}^{N_{\mathrm{h}}} \log \cosh \left( \sqrt{N_{\mathrm{v}}} \sum_{\alpha=1}^{d} m_\alpha w_{\alpha a} + \eta_a \right) \right] \right)}{Z} \tag{21}$$

where we ignore the fluctuations related to the transverse directions and $\mathcal{S}[\boldsymbol{m}]$ accounts for the non-uniform prior on $\boldsymbol{m}$ due to the projection of the uniform prior on $\boldsymbol{s}$ for the way to compute it.

The RCM is then built by approximating

$$\log \cosh(x) \simeq |x| - \log 2, \tag{22}$$

which is valid for $x$ large enough. The probability of the RCM is thus given by:

$$p_{\text{RCM}}(\boldsymbol{m}) = \frac{\exp\left(N_{\text{v}}\left[\mathcal{S}(\boldsymbol{m}) + \sum_{\alpha=1}^{d}\theta_\alpha m_\alpha + \sum_{a=1}^{N_{\text{h}}} q_a\left|\sum_{\alpha=1}^{d} n_\alpha m_\alpha + z_a\right|\right]\right)}{Z} \tag{23}$$

where

$$q_a = \sqrt{N_{\text{v}}\sum_{\alpha=1}^{d} w_{\alpha a}^2}, \quad n_a = \frac{w_{\alpha a}}{\sqrt{\sum_{\alpha=1}^{d} w_{\alpha a}^2}}, \quad z_a = \frac{\eta_a}{\sqrt{N_{\text{v}}\sum_{\alpha=1}^{d} w_{\alpha a}^2}}. \tag{24}$$

This can be easily inverted as

$$w_{\alpha a} = \frac{1}{\sqrt{N_{\text{v}}}} q_a n_a \qquad \text{and} \qquad \eta_a = q_a z_a,$$

in order to obtain the RBM from the RCM. The model is then trained through log-likelihood maximization over its parameters. However, this objective is non-convex if all the parameters are trained through gradient ascent. To relax the problem, since we're in low dimension, we can define a family of hyperplanes $(\boldsymbol{n}, \boldsymbol{z})$ covering the space and let the model only learn the weights of each to the hyperplane. We can then discard the ones with a low weights to keep the approximation (equation 22) good enough.

The gradients are given by

$$\frac{\partial J(\boldsymbol{\Theta})}{\partial q_a} = \mathbb{E}_{\boldsymbol{m}\sim p_{\mathcal{D}}(\boldsymbol{m})}\left[|\boldsymbol{n}_a^T \boldsymbol{m} + z_a|\right] - \mathbb{E}_{\boldsymbol{m}\sim p_{\text{RCM}}(\boldsymbol{m})}\left[|\boldsymbol{n}_a^T \boldsymbol{m} + z_a|\right], \tag{25}$$

$$\frac{\partial J(\boldsymbol{\Theta})}{\partial \theta_\alpha} = E_{\boldsymbol{m}\sim p_{\mathcal{D}}(\boldsymbol{m})}\left[m_\alpha\right] - \mathbb{E}_{\boldsymbol{m}\sim p_{\text{RCM}}(\boldsymbol{m})}\left[m_\alpha\right]. \tag{26}$$

The positive term is straightforward to compute. For the negative term, we rely on a discretization of the longitudinal space to estimate the probability density of the model and compute the averages.

### A.3 Learning the bias

To initialize the model with its bias set to the dataset's mean, we define the 0-th direction of the longitudinal space as the direction obtained by projecting the configuration space onto the normalized mean vector of the dataset.

The RBM will only contribute to this direction through the visible bias:

$$\mathcal{H}(\boldsymbol{m}(v)) = -\sum_{\alpha=0}^{N_{\text{v}}}\theta_\alpha m_\alpha - \frac{1}{N_{\text{v}}}\sum_{a}\log\cosh\left(\sqrt{N_{\text{v}}}\sum_{\alpha=1}^{d} m_\alpha w_{\alpha a} + \eta_a\right). \tag{27}$$

This implies that this direction is independent of the others within the model. Consequently, we can create an independent mesh for this direction to learn it. The rest of the training is performed using the same procedure as before.

## B Sampling via Parallel Tempering using the learning trajectory

Assuming we have successfully trained a robust equilibrium model, there remains the challenge of efficiently generating equilibrium configurations from this model. Although models trained at equilibrium exhibit faster and more ergodic dynamics compared to poorly trained models, the sampling time can still be excessively long when navigating a highly rugged data landscape. Consequently, we devised a novel method for sampling equilibrium configurations that draws inspiration from the well-established parallel tempering approach. In this traditional method, multiple simulations are conducted in parallel at various temperatures, and configurations are exchanged among them using the Metropolis rule. Unlike this conventional technique, our method involves simultaneously

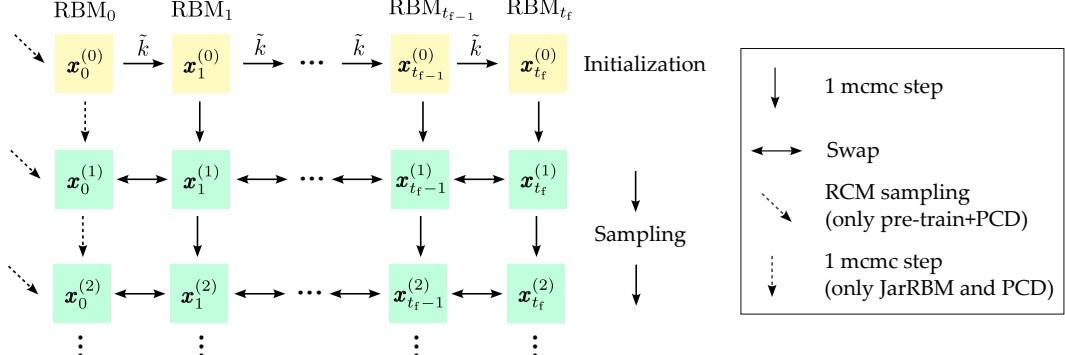

Figure 5: Scheme of PTT. We Initialize the chains of the models by starting from a configuration $\boldsymbol{x}_0^{(0)}$ and passing it through the machines along the training trajectory, each time performing $\tilde{k}$ mcmc steps. For pre-train+PCD, $\boldsymbol{x}_0^{(0)}$ is a sampling from the RCM, otherwise it is a uniform random initialization. The sampling consists of alternating one mcmc step for each model with a swap attempt between adjacent machines. For pre-train+PCD, at each step we sample a new independent configuration for $\mathrm{RBM}_0$ using the RCM.

simulating different models that are selected from various points along the training trajectory. This approach is motivated by the perspective that learning represents an annealing process for the model, encountering second-order type phase transitions during training. In contrast, annealing related to temperature changes involves first-order phase transitions, making traditional parallel tempering less effective for sampling from clustered multimodal distributions.

A sketch of the Parallel Trajectory Tempering (PTT) is represented in fig. 5. Specifically, we save $t_\mathrm{f}$ models at checkpoints $t = 1, \ldots, t_\mathrm{f}$ along the training trajectory. We denote the Hamiltonian of the model at checkpoint $t$ as $\mathcal{H}_t$, and refer to the Hamiltonian of the RCM model as $\mathcal{H}_0$. We define $\mathrm{GibbsSampling}(\mathcal{H}, \boldsymbol{x}, k)$ as the operation of performing $k$ Gibbs sampling updates using the model $\mathcal{H}$ starting from the state $\boldsymbol{x}$. In all our sampling simulations we used $k = 1$.

The first step is to initialize the models' configurations efficiently. This involves sampling $N$ chains from the RCM model, $\boldsymbol{x}_0^{(0)} \sim \mathrm{RCMSampling}(\mathcal{H}^0)$, and then passing the chains through all the models from $t = 1$ to $t = t_\mathrm{f}$, performing $k$ Gibbs steps at each stage: $\boldsymbol{x}_t^{(0)} \sim \mathrm{GibbsSampling}(\mathcal{H}_t, \boldsymbol{x}_{t-1}^{(0)}, k)$.

The sampling process proceeds in steps where we update the configuration of each model except $\mathcal{H}_0$ with $k$ Gibbs steps, and sample a completely new configuration for the RCM model $\mathcal{H}_0$. Following this update step, we propose swapping chains between adjacent models with an acceptance probability given by:

$$p_{\mathrm{acc}}(\boldsymbol{x}_t \leftrightarrow \boldsymbol{x}_{t-1}) = \min\left(1, \exp\left(\Delta\mathcal{H}_t(\boldsymbol{x}_t) - \Delta\mathcal{H}_t(\boldsymbol{x}_{t-1})\right)\right), \tag{28}$$

where $\Delta\mathcal{H}_t(\boldsymbol{x}) = \mathcal{H}_t(\boldsymbol{x}) - \mathcal{H}_{t-1}(\boldsymbol{x})$. We continue alternating between the update step and the swap step until a total of $N_{\mathrm{mcmc}}$ steps is reached.

A comparison of performances between PTT and standard Gibbs sampling is reported in Fig.3.

### B.1   PSEUDO-CODE OF PTT VS PT

We provide here the pseudo code for both Parallel Tempering (Alg. 1) and Parallel Trajectory Tempering (Alg. 2) sampling schemes. The difference between both algorithms lies in the way the parallel models are selected. In the case of Parallel Tempering, an annealing on the temperature is made, whereas for Parallel Trajectory Tempering several models are saved during training.

---

**Algorithm 1** Parallel Tempering

---

**Require:** $\{\beta_j\}_{j=\{1,\ldots,N_\beta\}}, \Theta = \{w, \theta, \eta\}, N_{\texttt{iter}}, N_{\texttt{increment}}, N_{\texttt{sample}}$
   **for** $j = 1, \ldots, N_\beta$ **do**
      $\texttt{chains}[j] = \texttt{RandomBinaryChains}(N_{\texttt{sample}})$
   **end for**
   **for** $i = 1, \ldots, N_{\texttt{iter}}$ **do**
      **for** $j = 1, \ldots, N_\beta$ **do**
         $\texttt{chains}[j] \leftarrow \texttt{AlternateGibbsSampling}(N_{\texttt{increment}}, \beta_j\Theta, \texttt{chains}[j])$
      **end for**
      **for** $j = 1, \ldots, N_\beta - 1$ **do**
         $\texttt{chains}[j], \texttt{chains}[j+1] \leftarrow \texttt{SwapConfigurations}(\texttt{chains}[j], \beta_j\Theta, \texttt{chains}[j+1], \beta_{j+1}\Theta)$
      **end for**
   **end for**

---

**Algorithm 2** Parallel Trajectory Tempering

---

**Require:** $\{\Theta_j = \{w_j, \theta_j, \eta_j\}\}_{j=\{1,\ldots,N_{\texttt{model}}\}}, N_{\texttt{iter}}, N_{\texttt{increment}}, N_{\texttt{sample}}$
   **for** $j = 1, \ldots, N_{\texttt{model}}$ **do**
      $\texttt{chains}[j] = \texttt{RandomBinaryChains}(N_{\texttt{sample}})$
   **end for**
   **for** $i = 1, \ldots, N_{\texttt{iter}}$ **do**
      **for** $j = 1, \ldots, N_{\texttt{model}}$ **do**
         $\texttt{chains}[j] \leftarrow \texttt{AlternateGibbsSampling}(N_{\texttt{increment}}, \Theta_j, \texttt{chains}[j])$
      **end for**
      **for** $j = 1, \ldots, N_{\texttt{model}} - 1$ **do**
         $\texttt{chains}[j], \texttt{chains}[j+1] \leftarrow \texttt{SwapConfigurations}(\texttt{chains}[j], \Theta_j, \texttt{chains}[j+1], \Theta_{j+1})$
      **end for**
   **end for**

---

| dataset | $\tau_{\exp}$ | $\tau_{\text{int}}$ |
|---------|---------------|---------------------|
| MNIST01 | 258(29) | 9.0(15) |
| HGD | 48(10) | 3.8(7) |
| Mickey | 2(0) | 1(0) |
| Ising | 18(3) | 2.4(5) |

Table 1: Autocorrelation times associated to the random walk dynamics between models in the PTT sampling obtained with the models trained with pre-training+PCD on different datasets. Models used in the PTT are all chosen with the $a = 0.25$ criterion.

### B.2 CONTROLLING EQUILIBRATION TIMES WITH PTT

One of the advantages of PT-like approaches is that they allow easy control over thermalization times by studying the random walk in temperatures (or models for the PTT) visited by each run over time, as proposed in Ref. (Banos et al., 2010). From now on, we will refer only to the number of models used in the PTT sampling, denoted as $N_{\text{m}}$. In an ergodic sampling process, each run should spend roughly the same amount of time at each of the $N_{\text{m}}$ different model indices, as all models have the same probability.

The mean temperature/model index is known to be $\langle n \rangle = (N_{\text{m}} - 1)/2$ (assuming the model indices run from 0 to $N_{\text{m}} - 1$). This simplifies the computation of the time-autocorrelation function of the model indices $n_i(t)$ visited by each sample $i$ at time $t$:

$$C(t) = \frac{\overline{(n_i(t + t_0) - \langle n \rangle)(n_i(t_0) - \langle n \rangle)}}{\overline{(n_i(t_0) - \langle n \rangle)^2}}$$

Here, the symbol $\overline{(\cdots)}$ refers to the average over all samples $i$ and initial times $t_0$ obtained from our PTT runs. This allows us to compute the exponential autocorrelation time, $\tau_{\exp}$, from a fit to

$$C(t) \sim A e^{-t/\tau_{\exp}} \quad \text{for large } t,$$

as well as the integrated autocorrelation time, $\tau_{\text{int}}$, using the self-consistent equation (Sokal, 1997; Amit & Martin-Mayor, 2005):

$$\tau_{\text{int}} = \frac{1}{2} + \sum_{t=0}^{6\tau_{\text{int}}} C(t).$$

These two times provide different insights: $\tau_{\exp}$ gives insight into the time needed to thermalize the simulations (setting the length of the runs to at least $20\tau_{\exp}$ is generally a safe (and very conservative) choice to ensure thermalization (Sokal, 1997)), while $\tau_{\text{int}}$ gives the time needed to generate independent samples. To keep track of the fluctuations, we compute a different $C^m(t)$ for each $m = 1, \ldots, N_{\text{m}}$ system index simulated in parallel in the PTT, where each $C^m(t)$ is averaged over 1000 independent realizations (with all runs initialized to the same model index). We then compute the average over all parallel runs $C(t) = \overline{C^m(t)}$ and use this curve to extract $\tau_{\exp}$ and $\tau_{\text{int}}$. The errors in $C$ and the times are then extracted from the error of the mean of the values extracted with each of the $C^m(t)$ curves.

For example, in Fig. 6 (left), we show the averaged curves $C(t)$ obtained with the PTT sampling of the machines discussed in the main text in the MNIST01 (top) and HGD (bottom) datasets. The extracted times are $\tau_{\exp} = 258(29)$ and $\tau_{\text{int}} = 9.0(1.5)$ for MNIST01, and $\tau_{\exp} = 48(10)$ and $\tau_{\text{int}} = 3.8(7)$ for HGD. These numbers guarantee us that the $10^4$ steps used to generate the scatterplots in Fig. 4 were long enough to ensure equilibration in this dataset, but also in the others, as shown in Table 1.

With these measures, we can try to optimize the number of models that we use for the PTT and that are automatically selected during training. The criterion here is that the acceptance of PTT exchanges between neighboring models must not go below a target acceptance $a$. In the figures shown so far, $a = 0.25$. Let us justify this choice. The total physical time we need to simulate $N_{\text{m}}$ models grows proportionally with $N_{\text{m}}$ (unless one manages to properly parallelize the sampling, which is feasible but we did not do it), while the time needed to flow between very well trained and poorly trained models controlling the mixing time scales as $\sqrt{N_{\text{m}}}/a$, where $a$ is the average acceptance of the exchanges between the models. Since $N_{\text{m}}$ grows with $a$ in practice (as shown in the inset of

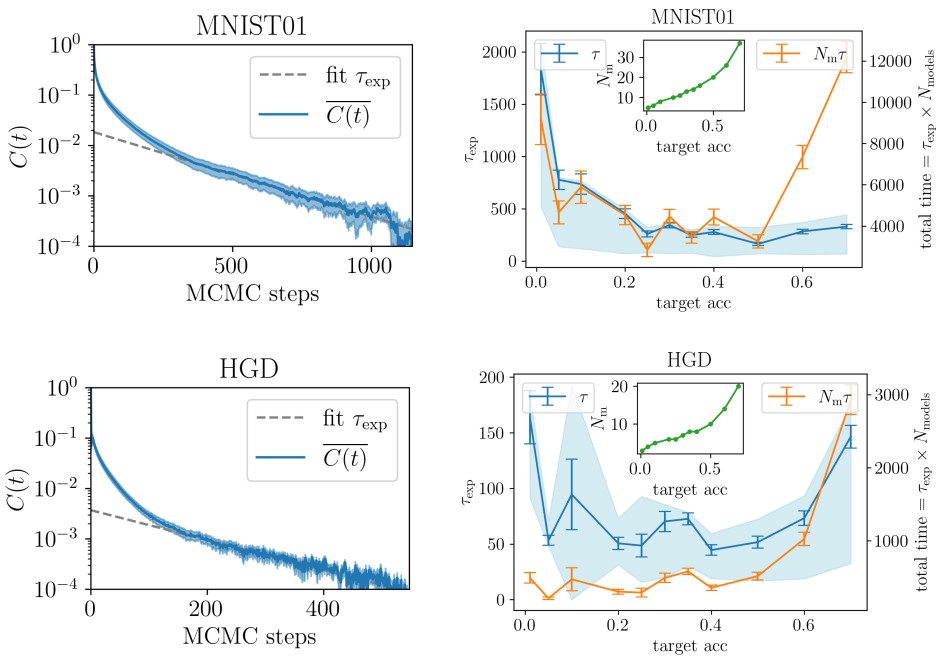

Figure 6: (Left) We show the autocorrelation function $C(t)$ obtained as in Eq. equation B.2 when we analyze the random walk in the model indices during PTT sampling with the same RBMs analyzed in the main text (trained with pretraining+PCD) for the MNIST01 (top) and HGD (bottom) datasets. Both curves were obtained with a PTT run of $5 \cdot 10^4$ MCMC steps, where the models used were automatically selected during training using the PTT moves acceptance criterion of 0.25. (Right) In blue we show the $\tau_{\exp}$ as a function of the target acceptance rate for MNIST01 (top) and HGD (bottom). The error bar results from the error of the mean of the values extracted from the different $C_m(t)$ and the shading marks the range between the largest and the smallest value of the $N_m$ curves. In orange, we show the total simulation costs ($\tau_{\exp}$ multiplied by the number of models to be simulated in parallel) as a function of the target acceptance rate. This number of models saved, as a function of the acceptance rate, is shown in green in the inset.

Fig. 6–right), it is expected that the total time can be optimized for a given value of $a$. In Fig. 6–right we show $\tau_{\exp}$ in blue (and $\tau_{\exp} N_m$ in orange) as a function of $a$ for the data sets MNIST01 (top) and HGD (bottom). These figures show that the choice of $a$ is not particularly crucial as long as it is between 0.2 and 0.5.

## C  TRAINING DETAILS

We describe in Tables 2 and 3 the datasets and hyperparameters used during training. The test set was used to evaluate the metrics.

Table 2: Details of the datasets used during training.

| Name | #Samples | #Dimensions | Train size | Test size |
|---|---|---|---|---|
| CelebA | 30 000 | 1024 | 60% | 40% |
| Human Genome Dataset (HGD) | 4500 | 805 | 60% | 40% |
| Ising | 20 000 | 64 | 60% | 40% |
| Mickey | 16 000 | 1000 | 60% | 40% |
| MNIST-01 | 10 610 | 784 | 60% | 40% |
| MNIST | 50 000 | 784 | 60% | 40% |

Table 3: Hyperparameters used for the training of RBMs.

| Name | Batch size | #Chains | #Epochs | Learning rate | #MCMC steps | #Hidden nodes |
|---|---|---|---|---|---|---|
| CelebA | | | | | | |
| PCD | 2000 | 2000 | 10 000 | 0.01 | 100 | 500 |
| Pre-train+PCD | 2000 | 2000 | 10 000 | 0.01 | 100 | 500 |
| HGD | | | | | | |
| PCD | 2000 | 2000 | 10 000 | 0.01 | 100 | 200 |
| Pre-train+PCD | 2000 | 2000 | 10 000 | 0.01 | 100 | 200 |
| Ising | | | | | | |
| PCD | 2000 | 2000 | 10 000 | 0.01 | 100 | 200 |
| Pre-train+PCD | 2000 | 2000 | 10 000 | 0.01 | 100 | 200 |
| Mickey | | | | | | |
| PCD | 2000 | 2000 | 10 000 | 0.01 | 100 | 200 |
| Pre-train+PCD | 2000 | 2000 | 10 000 | 0.01 | 100 | 200 |
| MNIST-01 | | | | | | |
| PCD | 2000 | 2000 | 10 000 | 0.01 | 100 | 200 |
| Pre-train+PCD | 2000 | 2000 | 10 000 | 0.01 | 100 | 200 |
| MNIST | | | | | | |
| PCD | 2000 | 2000 | 10 000 | 0.01 | 100 | 200 |
| Pre-train+PCD | 2000 | 2000 | 10 000 | 0.01 | 100 | 200 |

All experiments were run on a RTX 4090 with an AMD Ryzen 9 5950X.

## D    COMPUTATION OF THE LIKELIHOOD USING AIS

To compute the likelihood of the RBMs we used the Annealed Importance Sampling method introduced by (Neal, 2001) and recently studied in the case of the RBM (Krause et al., 2020).

### D.1    OFFLINE AIS

We follow the method described in (Krause et al., 2020). The AIS estimate is based on creating a sequence of machines, at one end a machine from which we can compute the partition function and at the other end the machine from which we want to compute the partition function. Let's consider that we have $i = 0, \ldots, N_{\mathrm{rbm}}$ machines described by

$$p_i(\boldsymbol{x}) = \frac{e^{-\mathcal{H}_i(\boldsymbol{x})}}{Z} = \frac{p_i^*(\boldsymbol{x})}{Z},$$

where $p_0 = p_{\mathrm{ref}}$ denote the RBM from which we can compute the partition function (typically a RBM where the weight matrix is zero) and $p_{N_{\mathrm{rbm}}} = p_{\mathrm{target}}$ the machine from which we want to estimate the partition function. For each machine, we denote its set of parameters $\boldsymbol{\Theta}_i = \{\boldsymbol{w}_i, \boldsymbol{\theta}_i, \boldsymbol{\eta}_i\}$. Following (Krause et al., 2020), defining a transition rule $T_i(\boldsymbol{x}_{i-1} \to \boldsymbol{x}_i)$ in order to pass from a configuration $\boldsymbol{x}_{i-1}$ of the system $i - 1$ to $\boldsymbol{x}_i$ of the system $i$, that we have the following identity

$$\frac{Z_{N_{\mathrm{rbm}}}}{Z_0} = \left\langle \exp\left( - \sum_{i=1}^{N_{\mathrm{rbm}}} \mathcal{H}_{i+1}(\boldsymbol{x}_i) - \mathcal{H}_i(\boldsymbol{x}_i) \right) \right\rangle_{\mathrm{ForwardPath}}. \tag{29}$$

There, if one can compute $Z_0$, the estimate of $Z_{N_{\mathrm{rbm}}}$ is obtained by averaging the r.h.s. of Eq. 29 over the forward trajectory starting from a configuration $\boldsymbol{x}_0$ using the operators $T_i$. While the method is exact, the variance of the estimation relies on on close are two successive distribution $p_i$, $p_{i+1}$.

Table 4: Training time of the RBMs

| Name | Training time (s) |
|---|---|
| **CelebA** | |
| PCD | $119.2 \pm 0.8$ |
| Pre-train+PCD | $3134.0 \pm 0.6$ |
| **HGD** | |
| PCD | $106.2 \pm 0.3$ |
| Pre-train+PCD | $266 \pm 3$ |
| **Ising** | |
| PCD | $62 \pm 1$ |
| Pre-train+PCD | $104 \pm 2$ |
| **Mickey** | |
| PCD | $118.3 \pm 3$ |
| Pre-train+PCD | $246.0 \pm 0.7$ |
| **MNIST-01** | |
| PCD | $100.6 \pm 0.3$ |
| Pre-train+PCD | $258.0 \pm 0.9$ |
| **MNIST** | |
| PCD | $105.7 \pm 0.2$ |
| Pre-train+PCD | $233.9 \pm 0.2$ |

Table 5: RCM Hyperparameters

| Dataset name | # Dimensions | # Hidden nodes | Training time (s) |
|---|---|---|---|
| CelebA | 4 | 100 | 3006 |
| HGD | 3 | 100 | 150 |
| Ising | 3 | 100 | 32 |
| Mickey | 2 | 100 | 118 |
| MNIST-01 | 3 | 100 | 148 |
| MNIST | 3 | 100 | 119 |

**Temperature AIS:** a commonly used implementation of AIS, is to multiply the Hamiltonian of the system by a temperature, and to the different systems indexed by $i$, by a set of $\beta_i$ where $\beta_0 = 0$, $\beta_i < \beta_{i+1}$ and $\beta_{N_{\mathrm{rbm}}} = 1$. Therefore, it is equivalent to choosing $\Theta_i = \beta_i\{\boldsymbol{w}, \boldsymbol{\theta}, \boldsymbol{\eta}\}$. Another version of Temperature AIS consist in choosing a slightly different reference distribution $p_0$ that take into account the bias of the model. In this setting, the intermediate distribution are taken as

$$p_i(\boldsymbol{x}) = \frac{p_{\mathrm{ref}}^{(1-\beta_i)} e^{-\beta_i \mathcal{H}(\boldsymbol{x})}}{Z_i}.$$

The main difference is that a prior distribution is used as reference $p_0 = p_{\mathrm{ref}}$ and the temperatures interpolate between this prior and the target distribution. It is quite common to take

$$p_{\mathrm{ref}}(\boldsymbol{x}) = \frac{\exp\left(\sum_i \theta_i v_i + \sum_a \eta_a h_a\right)}{\prod_i 2\cosh(\theta_i) \prod_a 2\cosh(\eta_a)}.$$

**Trajectory AIS:** the general derivation of AIS allow in principle to choose the set forward probability and/or intermediate system as one sees fit. Since again, a change of temperature in the system

can induce discontinuous transition for which, successive distribution will be far away. While, when considering system that are neighbors in the learning trajectory, the system should pass through continuous transition which do not suffer this problem. Hence, we consider a set of RBMs along the learning trajectory. The RBMs are chosen following the prescription that the exchange rate between configuration (see Eq. 3) is about $\sim 0.3$). Then the formula Eq. 29 is used to estimate the partition function.

## D.2 Online AIS

The online AIS (Annealed Importance Sampling) is the trajectory AIS evaluated at every gradient update during training. A set of permanent chains is initialized at the beginning of the training process. Every time the parameters are updated, a step is performed on these chains before using them to compute the current importance weights. This approach is computationally efficient relative to the cost of sampling (1 step vs. 100 steps) and allows for a very precise estimation.

## E  Training of the RBM using the Jarzynski equation

In this section, we describe the procedure introduced in (Carbone et al., 2024b) for training energy-based models by leveraging the Jarzynki equation, and we adapt it to the specific case of the RBM. At variance with (Carbone et al., 2024b), we introduce an additional assumption on the transition probability of the parameters that will lead us to the the reweighting implemented in population annealing methods (Weigel et al., 2021).

In one of its formulations, the Jarzynski equation states that we can relate the ensemble average of an observable $\mathcal{O}$ with the average obtained through many repetitions of an out-of-equilibrium dynamical process. Let us consider a path $\boldsymbol{y} = \{(\boldsymbol{x}^0, \boldsymbol{\theta}^0), (\boldsymbol{x}^1, \boldsymbol{\theta}^1), \dots, (\boldsymbol{x}^t, \boldsymbol{\theta}^t)\}$, where $\boldsymbol{x}^i = (\boldsymbol{v}^i, \boldsymbol{h}^t)$ refers to the model variables and $\boldsymbol{\theta}^i = (w^i, \eta^i, \theta^i)$, the model parameters, both at time $i$.

If we consider the training trajectory of an RBM, $p_0 \to p_1 \to \cdots \to p_{t-1} \to p_t$, we can write the average of an observable $\mathcal{O}$ over the last model of the trajectory $p_t$ as

$$\langle \mathcal{O} \rangle_t = \frac{\left\langle \mathcal{O} \, e^{-W^t} \right\rangle_{\text{traj}}}{\left\langle e^{-W^t} \right\rangle_{\text{traj}}} = \frac{\sum_{r=1}^R \mathcal{O}(\boldsymbol{y}_r) \, e^{-W_r^t}}{\sum_{r=1}^R e^{-W_r^t}}, \tag{30}$$

where $W^t$ is a trajectory-dependent importance factor and the averages on the rhs are taken across many different trajectory realizations. In particular, $W_t$ is given by the recursive formula:

$$\begin{aligned} W^t(\boldsymbol{y}) &= \mathcal{H}^t(\boldsymbol{x}^t) - \mathcal{H}^0(\boldsymbol{x}^0) - \sum_{i=1}^t \log \tilde{T}^i(\boldsymbol{y}^i \to \boldsymbol{y}^{i-1}) + \sum_{i=1}^t \log T^i(\boldsymbol{y}^{i-1} \to \boldsymbol{y}^i) \quad (31) \\ &= W^{t-1}(\boldsymbol{y}) + \mathcal{H}^t(\boldsymbol{x}^t) - \mathcal{H}^{t-1}(\boldsymbol{x}^{t-1}) - \log \frac{\tilde{T}^t(\boldsymbol{y}^t \to \boldsymbol{y}^{t-1})}{T^t(\boldsymbol{y}^{t-1} \to \boldsymbol{y}^t)} \quad (32) \end{aligned}$$

Where $T$ and $\tilde{T}$ are, respectively, the forward and backward transition probabilities.

The transition matrix between two states $\boldsymbol{y}^{t-1} \to \boldsymbol{y}^t$ is factorized into two different subsequent movements:

1. Update the variables $\boldsymbol{x}^{t-1} \to \boldsymbol{x}^t$ on the fixed model $\boldsymbol{\theta}^{t-1}$,
2. Update the parameters $\boldsymbol{\theta}^{t-1} \to \boldsymbol{\theta}^t$ on fixed variables $\boldsymbol{x}^t$.

In other words,

$$T(\boldsymbol{y}^{t-1} \to \boldsymbol{y}^t) = w_{\boldsymbol{\theta}^{t-1}}(\boldsymbol{x}^{t-1} \to \boldsymbol{x}^t) t_{\boldsymbol{x}^t}(\boldsymbol{\theta}^{t-1} \to \boldsymbol{\theta}^t), \tag{33}$$

and

$$\tilde{T}(\boldsymbol{y}^t \to \boldsymbol{y}^{t-1}) = t_{\boldsymbol{x}^t}(\boldsymbol{\theta}^t \to \boldsymbol{\theta}^{t-1}) w_{\boldsymbol{\theta}^{t-1}}(\boldsymbol{x}^t \to \boldsymbol{x}^{t-1}), \tag{34}$$

so that

$$\log \frac{\tilde{T}^t(\boldsymbol{y}^t \to \boldsymbol{y}^{t-1})}{T^t(\boldsymbol{y}^{t-1} \to \boldsymbol{y}^t)} = \log \frac{w_{\boldsymbol{\theta}^{t-1}}(\boldsymbol{x}^t \to \boldsymbol{x}^{t-1})}{w_{\boldsymbol{\theta}^{t-1}}(\boldsymbol{x}^{t-1} \to \boldsymbol{x}^t)} + \log \frac{t_{\boldsymbol{x}^t}(\boldsymbol{\theta}^t \to \boldsymbol{\theta}^{t-1})}{t_{\boldsymbol{x}^t}(\boldsymbol{\theta}^{t-1} \to \boldsymbol{\theta}^t)}. \tag{35}$$

We know the ratio between the transition probabilities between $\boldsymbol{x}^{t-1} \to \boldsymbol{x}^t$, because evolution of the variables during the simulation is performed using MCMC simulations that satisfy detailed balance. This means that:

$$\log \frac{w_{\boldsymbol{\theta}^{t-1}}(\boldsymbol{x}^t \to \boldsymbol{x}^{t-1})}{w_{\boldsymbol{\theta}^{t-1}}(\boldsymbol{x}^{t-1} \to \boldsymbol{x}^t)} = \mathcal{H}^{t-1}(\boldsymbol{x}^t) - \mathcal{H}^{t-1}(\boldsymbol{x}^{t-1}), \tag{36}$$

and we are left with

$$W^t(\boldsymbol{y}) = W^{t-1}(\boldsymbol{x}) + \mathcal{H}^t(\boldsymbol{x}^t) - \mathcal{H}^{t-1}(\boldsymbol{x}^t) - \log \frac{t_{\boldsymbol{x}^t}(\boldsymbol{\theta}^t \to \boldsymbol{\theta}^{t-1})}{t_{\boldsymbol{x}^t}(\boldsymbol{\theta}^{t-1} \to \boldsymbol{\theta}^t)}. \tag{37}$$

The standard formula that we use in the paper assumes that the changes in $\boldsymbol{\theta}$ are so small that we can consider the transition probabilities to be symmetrical and the last term in the rhs cancels out. In that case, we recover the known expressions of the population annealing.

Notice that, since Eq. equation 30 is an exact result, the importance weights should, in principle, eliminate the bias brought by the non-convergent chains used for approximating the log-likelihood gradient in the classical PCD scheme. However, after many updates of the importance weights, one finds that only a few chains carry almost all the importance mass. In other words, the vast majority of the chains we are simulating are statistically irrelevant, and we expect to get large fluctuations in the estimate of the gradient because of the small effective number of chains contributing to the statistical average. A good observable for monitoring this effect is the Effective Sample Size (ESS), defined as (Carbone et al., 2024b)

$$\text{ESS} = \frac{\left(R^{-1} \sum_{r=1}^R e^{-W^{(r)}}\right)^2}{R^{-1} \sum_{r=1}^R e^{-2W^{(r)}}} \in [0, 1], \tag{38}$$

which measures the relative dispersion of the weights distribution. Then, a way of circumventing the weight concentration on a few chains is to resample the chain population according to the importance weights every time the ESS drops below a certain threshold, for instance 0.5. After this resampling, all the chain weights have to be set to 1 ($W^{(r)} = 0 \ \forall r = 1, \ldots, R$).

## F  TUNING OF THE HYPERPARAMETERS

In order to justify our choice of hyperparameters in the main text, we illustrate on Fig. 7 the value of the log-likelihood as a function of the number of hidden nodes and of the learning of the machine. In order to be fair, the comparison is done at a constant number of gradient updates of 10000. In this regime, the machines that perform the best have a learning rate of $\gamma \sim 0.1, 0.005$ and about $N_{\rm h} = 500, 1000$ hidden nodes. We can expect that smaller learning rates reach the same value of the log-likelihood but for a much longer training. We clearly see the benefit of the RCM, where the log-likelihood achieves much higher values in general.

## G  CONTINUOUS PHASE TRANSITION

In statistical physics, and more precisely in mean-field model, a model described by the Gibbs-Boltzmann distribution is studied in the infinite size limit, $N \to \infty$, and characterize the state of the system by investigating the free energy of the system, defined by minus the log partition function

$$F[\boldsymbol{\theta}] = -\log(Z[\boldsymbol{\theta}])$$

One of the goals is thus to determine the typical configurations that should be expected given a set of model parameters. In the case of the RBM, the question is what are the typical visible (and/or hidden) variables that one expects to observe. For example, one may ask what typical energies one expects to observe for the parameters $\boldsymbol{\theta}$. For this purpose, let us rewrite the partition function as a sum over all possible values of the energy

$$Z = \sum_E e^{-E} g(E) = \sum_E e^{-E+S(E)} = \sum_E e^{-F(E)} = \sum_E e^{-Nf(E)}$$

where $g(E)$ the number of states with a given energy $E$, and thus $S(E) = \log g(E)$ is the entropy. Finally, the free energy is defined as $F(E) = E - S(E)$ and $f(E) = F(E)/N$ is the free energy per variable. All $Z$, $E$ and $F$ depend on the model parameters. $N$ refers to the number of variables.

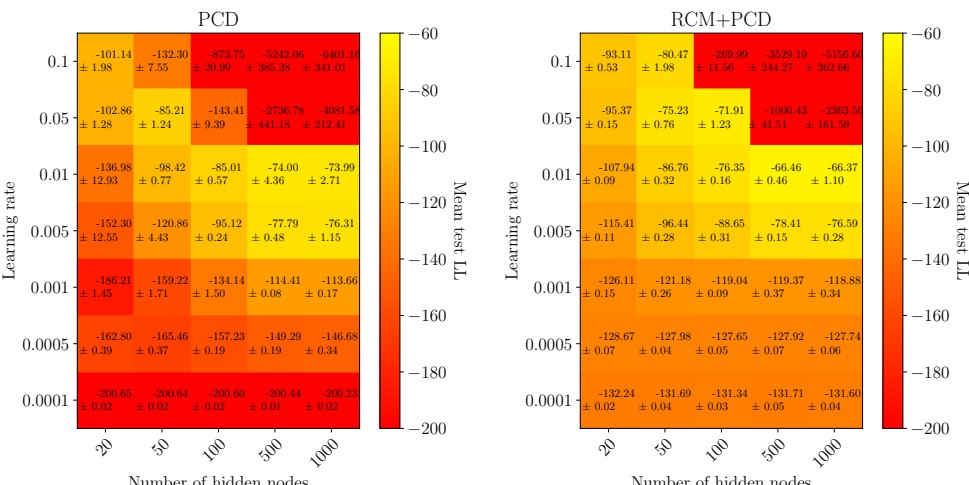

Figure 7: Comparison of the end log-likelihood reached by the RBM using different pairs of values for the learning rate and the hidden layer dimension on MNIST-01. The left panel shows these result when performing PCD training and the right one the results when pre-training the RBM before PCD.

For $N \to \infty$, the sum in Z is then dominated by the states with minimum free energy. A "first order", or discontinuous transition, occurs in the parameters if the first derivative of $f(E)$ is discontinuous with respect to the parameters (in physics, the parameter is typically the temperature). A "second order", or continuous transition, occurs when the first derivative is continuous, but the second derivative is discontinuous.

In the case of the RBM, the first learning transition can be mapped to a very simple model, the so-called Curie-Weiss model, as recently described in Ref. (Bachtis et al., 2024). The Curie-Weiss model is fully solvable, and for this reason the meaning of a continuous transition is perhaps easier to understand there. Let's give it a try. The Curie-Weiss model consists in a set of $N$ discrete variables $s_i = \pm 1$. We define the (exponential) distribution over these variables as

$$p(s) = \frac{1}{Z} \exp(\frac{\beta}{N} \sum_{i<j} s_i s_j + \beta H \sum_i s_i)$$

where $\beta$ and $H$ are parameters of the model and $Z$ the normalization constant also called partition function. The question is to understand the "structure" of the distribution as a function of $\beta$.

**Case $H = 0$:** we expect that for $\beta$ small each spin behaves as an isolated Rademacher random variables (each spin being $\pm 1$ with probability one-half), while for large value of $\beta$, the distribution of the system is dominated by configurations where the variables have (almost) all the same signs. The key mathematical aspect is to study the system in the limit where $N \to \infty$. A way to study this distribution is to investigate the moment generating function. It is possible to show (rigorously in this precise model), that the probability of $m = N^{-1} \sum_i s_i$, also called the magnetization of the system, is given by $p(m) \propto \exp(-N\Omega(m))$ (in the large $N$ limit) where $\Omega(m)$ is a large deviation function. In the CW model, it is given by

$$\Omega(m) = \frac{\beta m^2}{2} - \log\left[2\cosh(\beta m)\right].$$

The structure of $\Omega(m)$ is such that for $\beta < \beta_c = 1$, it is a convex function with only one minimum in $m = 0$, while for $\beta > \beta_c$ it has two symmetric minima $m = \pm m(\beta) \neq 0$ with $\Omega(m(\beta)) = \Omega(-m(\beta))$ (the two minimum are equally probable). The values (in function of $\beta$) of the $m$ are continuous: $m = 0$ for $\beta \in [0, 1]$, and then positive until it saturates to 1 as $\beta \to \infty$. In practice, it means that the distribution $p(m)$ pass from an unimodal distribution to a bimodal one as $\beta$ increases. The point where it happens is at $\beta_c = 1$ and is called a second order (continuous) phase transition because de value of $m$ changes continuously from zero to non-zero values. A particular interesting properties is that at $\beta_c$, the function $\Omega(m)$ develops a non-analyticity in its second-order derivative,

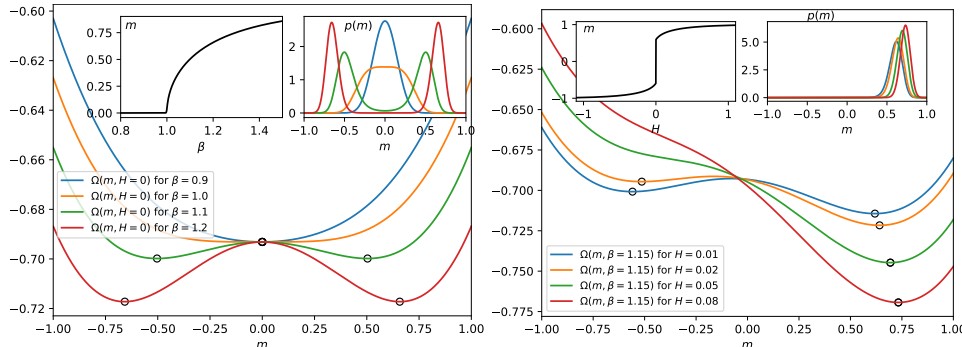

Figure 8: **Left:** The case $H = 0$ for various temperature. We illustrate the symmetric shape of the large deviation function and how the two minima emerges from the centers. The minima of $m$ go continuously from zero to non-zero values. In the inset we also illustrate the shape of the probability distribution for moderately large size $N = 500$. For $N \to \infty$ they converge toward a $\delta$ distribution. **Right:** The case $H \neq 0$. In this example, we illustrate what happen where the bias parameter $H$ is changed at a fix temperature $\beta = 1.15 > 1$. We see that for small values of $H$, $\Omega$ has two minima, yet the dominant one suppresses exponentially the other one for large system size. For larger values of the bias the metastable minimum disappears. We show how the place of the minimum jump when the bias goes from negative to positive values.

that can be linked with long-range fluctuations. For a recent review (Kochmański et al., 2013), and for more rigorous results (Sinai, 2014).

**Case $H \neq 0$:** in this case we expect that the system will always be biased toward configurations where the spins have the same direction(sign) as the bias (or field) $H$. Yet, for small value of $\beta$, the bias will be quite small, hence the magnetization will be small but not zero, $m \sim \beta H$. When $\beta$ is large, the system will now have a stronger and strong magnetization toward the magnetic field $m \sim \text{sign}(H)$. Now we can again compute the large deviation function of the system

$$\Omega(m) = \frac{\beta m^2}{2} - \log\left[2\cosh(\beta m)\right] + Hm.$$

Again, it is possible to analyze the function $\Omega$ as a function of $\beta$ and $H$. For $\beta < 1$ the function is convex and there is only one minimum located at $m > 0$. When $\beta > 1$ two situations can be distinguished. First, let's denote

$$H_{\text{sp}}(\beta) = \sqrt{\frac{\beta - 1}{\beta}} - \text{atanh}\left(\frac{\beta - 1}{\beta}\right).$$

If $H > H_{\text{sp}}(\beta)$, then the large deviation function has only one minimum for positive $m$. Now if $H < H_{\text{sp}}(\beta)$, then $\Omega$ will present two minima. A first one for positive $m$ and a second one for negative $m$. The reason is that the pairwise interactions "manage" to create a metastable state for negative magnetization even in the presence of a positive field or bias. It is clear that this state will be unstable w.r.t. the one with positive magnetization which traduces in a minimum of higher value in the large deviation function. By changing the value of $H$ from large positive values toward large negative ones, we then pass from a system with only one state, then two states with one being subdominant, and finally back to one state but where the magnetization has changed its sign. In the infinite size limit, the system will always dominated by the state where $\Omega$ is at its lowest value. However, if one prepares a system close to a metastable state, the system will remain trapped there until it disappears.

**How is this related to the training of the RBM ?** First, it has been shown in previous works (Decelle et al., 2018; Bachtis et al., 2024) that the RBM's training undergoes a set of continuous transitions, at least at the beginning of the learning. Hence, changing the parameters of the RBM following the learning trajectories, the transitions are continuous. Then in (Decelle & Furtlehner, 2021a) it is shown that the relaxed version of the RBM, namely the RCM undergoes a first-order transition when changing the temperature. Schematically to understand that, it is enough to see how

the parameters $\beta$ acts on the system. In the case of the RCM, the energy term behaves in the standard way, i.e. linearly with the inverse temperature $\beta$ and the free energy takes the ordinary form

$$\log(Z) = -\beta F = -\beta E + S.$$

Therefore $\beta$ act as a trade-off parameters between the energy $E$ and the entropy $S$. By varying $\beta$, we change the contribution of the energy while keeping the entropy fixed. Hence, when the corresponding large deviation function of the RCM has reached a point having many minima with the same value of the free energy up to $\mathcal{O}(1/N)$ corrections, all these minima realize in principle different trade-off between energy and entropy. As a result a slight change in $\beta$ will offset this sensible balance and the respective free energies will acquire differences of order $\mathcal{O}(N)$ leading to favor dramatically one state against the others, i.e. the one having originally the lowest [resp. the largest] energy when $\beta$ is increased. It is very likely that by changing $\beta$ a discontinuous transition is encountered, where the minima of $\Omega$ vanishes far away from each other rather than a continuous one. Concerning the RBM the argument for the occurence of such a phase transition, is only slightly more involved than for Coulomb machine because for RBM the energy (obtained after summing over latent variables) has a non-linear behavior w.r.t. $\beta$. A formal argument would go schematically as follows: just writing the local minima of the free energy (function of magnetization in the reduced intrinsic space) and looking at how the equilibrium is displaced with temperature shows that their energy vary indeed in a non-linear way with temperature and non-uniformly regarding their position on the magnetization manifold, while the entropy contribution again do not change. This again necessarily implies that changing even slightly the temperature will break the highly sensitive equilibrium obtained between these state corresponding to the multimodal distribution and will again typically favor one state among all. This scenario is expected to occur as soon as the data are located on low dimensional space ($d = O(1)$) compared to a large embedding space ($d = O(N)$, $N \gg 1$) which is quite common in our point of view, at least for the type of data we are interested in.

To conclude, moving the parameters along the learning trajectory, leads the system to undergo continuous transition where $p_{\boldsymbol{\Theta}(t)}(\boldsymbol{s})$ is very close to $p_{\boldsymbol{\Theta}(t+\delta t)}(\boldsymbol{s})$. Changing the temperature tends to produces discontinuous transition where $p_{\beta(t)\boldsymbol{\Theta}}(\boldsymbol{s})$ can happen to be very far from $p_{\beta(t+\delta t)\boldsymbol{\Theta}}(\boldsymbol{s})$.

## H    FIRST ORDER TRANSITIONS IN RBMS

In this section, we examine the appearance of first-order transitions when adjusting the temperature of a well-trained RBM on clustered datasets—a phenomenon that significantly hinders the performance of thermal sampling algorithms. We first illustrate this issue using an analytically tractable toy model, followed by an analysis of the first-order temperature transition observed in the RBM trained on the HGD dataset, as discussed in the main text.

### H.1    THEORETICAL ANALYSIS IN A SIMPLE MODEL

We propose a simple model of dataset on which we will show that the learned RBM is having a first order transition in temperature. We first consider an artificial dataset using a Curie-Weiss model. We defined the following Hamiltonian on the variables $s_i = \pm 1$

$$\mathcal{H} = -\frac{1}{2N_{\mathrm{v}}} \left( \sum_i s_i \right)^2, \quad p(\boldsymbol{s}) = \frac{1}{Z} \exp(-\beta_T \mathcal{H}).$$

When the inverse temperature $\beta_T$ is below one, the distribution in the $N_{\mathrm{v}} \to \infty$ limit is unimodal, with the Boltzmann average of the variables $s$ being $\langle s \rangle = 0$. For $\beta_T > 1$, the system becomes bimodal, with $\langle s_i \rangle = m$, where $m$ satisfies the self-consistent equation $m = \tanh(\beta_T m)$. We propose to model this dataset using a Bernoulli-Gauss RBM with one hidden node. The visible nodes, $\sigma = \{0, 1\}$, follow a Bernoulli distribution, while the hidden node, $\tau$, follows a Gaussian distribution with zero mean and variance $1/N_{\mathrm{v}}$, combined with a local visible bias. When the inverse temperature $\beta_T$ is below one, the distribution in the $N_{\mathrm{v}} \to \infty$ limit is unimodal and the average w.r.t. the Boltzmann distribution of the values of the variables $s$'s is $\langle s \rangle = 0$. When $\beta_T$ is above one, the system becomes bimodal with $\langle |s_i| \rangle = m$ when $m$ is defined by the self-consistent equation $m = \tanh(\beta_T m)$. We now propose to learn this dataset using a Bernoulli-Gauss RBM with one hidden node, where the visible nodes are $\sigma = \{0, 1\}$, and the hidden one $\tau$ is Gaussian distribution

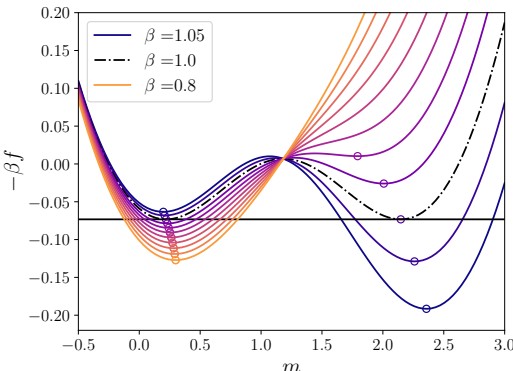

Figure 9: Free energy $-\beta f(m_\tau)$ for $\beta \in [0.8, 1.05]$ for the optimal RBM's parameters learned on a CW model with $\beta_T = 1.4$. At $\beta = 1.0$ the two minima are perfectly adjusted. As we change the temperature of the RBM, they get destabilized showing the presence of a first order transition.

of zero mean and variance $1/N_v$, and a local visible bias. The Hamiltonian is then

$$\mathcal{H} = -\sum_i \sigma_i W_i \tau - \sum_i \eta_i \sigma_i,$$

where $W_i$ is the weight matrix and $\eta_i$ the biases. The leaning then consist in converting the binary variable of the Curie-Weiss model into $\{0, 1\}$ and to learn the coresponding dataset. By a rapid inspection (in the infinite size limit), we can easily show that the optimal learned parameters for the RBM are

$$w_i = 2\sqrt{\beta_T} \text{ and } \eta_i = -2\beta_T.$$

Now, thanks to the simplicity of the model, we can actually compute the free energy of the model where the Hamiltonian has been rescaled by an annealing temperature $\beta$, we obtain

$$-f(m_T) = \frac{m_\tau^2}{2} - N^{-1} \sum_i \log\left[1 + \exp(\beta w_i m_\tau + \beta \eta_i)\right]$$

$$\text{where } m_\tau = \frac{1}{N} \sum_i \beta w_i \text{sigm}\left[1 + \exp(\beta w_i m_\tau + \beta \eta_i\right]$$

In Fig. 9, we plot $-\beta f(m_T)$ for ten different values of $\beta$ within the range $[0.8, 1.05]$. It is evident that $\beta = 1$ represents the coexistence point, indicating a first-order transition in temperature. Note that the probability $p(m) \approx e^{-N\beta f(m_T)}$ for large $N$, implying that for $\beta > 1$, the typical configuration corresponds to $m > 1$, whereas for $\beta < 1$, it corresponds to $m < 1$. This clearly shows that even on simple case we observe such transition when doing annealing.

## H.2 FIRST ORDER PHASE TRANSITION IN TEMPERATURE ON A TRAINED RBM ON A REAL DATASET

We now consider the RBM trained on the HGD dataset using the pretraining and 50 000 updates of PCD training. As we vary the temperature, we recover the probability $p(\boldsymbol{m})$ learned by the model using the TMC method (Béreux et al., 2023), projected along the second principal component in Fig. 11 and on both first and second principal components in Fig. 10. While $\beta < 1.0$, the probability distribution is dominated by the central mode, equilibrating between all modes $\beta = 1.0$. As $\beta$ goes above 1.0, the distribution becomes dominated by the two external clusters.

The effect is clearly visible when sampling the model using the PT algorithm (Fig. 12). The sampling was performed with 200 temperatures with a minimum acceptance rate of 0.78 between two temperatures. We see a dataset cluster being ignored with only a few samples in it. However, when resampling sampling the model for $10^6$ AGS steps starting from the samples obtained through PT, we see a clear change in the distribution, equilibrating with the missing cluster. In contrast, performing the same experiment with PTT instead of PT, we observe no shift in the empirical distribution of the samples after $10^6$ AGS, as the upper cluster is already sampled.

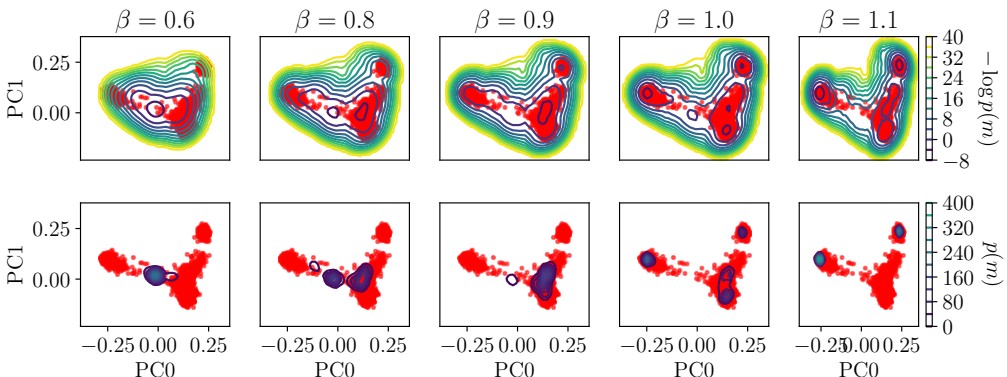

Figure 10: Projection of the negative log probability (top) and probability (bottom) learned by the model, constrained to visible magnetizations $m_0$ and $m_1$, for various values of $\beta$. The red dots represent the training data projected onto the first two principal components.

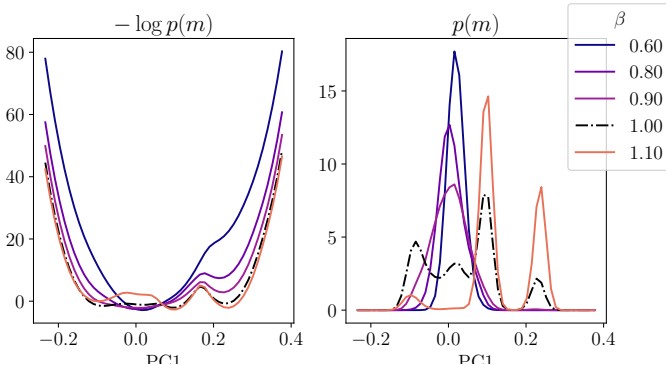

Figure 11: Negative log probability (left) and probability (right) learned by the model constrained to have a given $m_1$ for several values of $\beta$. The equilibrium distribution of the model is displayed as the black dotted line ($\beta = 1$).

## I  OVERFITTING AND PRIVACY LOSS AS QUALITY INDICATORS

In this section, we examine the quality of the samples generated, regarding overfitting and privacy criteria which have been defined for genomic data in particular. We look at this on the models trained with PCD with and without pre-training. We focus on the human genome dataset, as shown in Fig. 1–C, to evaluate the ability of various state-of-the-art generative models to generate realistic fake genomes while minimizing privacy concerns (i.e., reducing overfitting). Recent studies (Yelmen et al., 2021; 2023) have thoroughly investigated this for a variety of generative models. Both studies concluded that the RBM was the most accurate method for generating high-quality and private synthetic genomes. The comparison between models relies primarily on the Nearest Neighbor Adversarial Accuracy ($AA_{\text{TS}}$) and privacy loss indicators, introduced in Ref. (Yale et al., 2020), which quantify the similarity and the level of "privacy" of the data generated by a model w.r.t. the training set. We have $AA_{\text{TS}} = \frac{1}{2}\big(AA_{\text{True}} + AA_{\text{Synth}}\big)$ where $AA_{\text{True}}$ [resp. $AA_{\text{Synth}}$] are two quantities in $[0, 1]$ obtained by merging two sets of real and synthetic data of equal size $N_s$ and measuring respectively the frequency that a real [rep. synthetic] has a synthetic [resp. real] as nearest neighbor. If the generated samples are statistically indistinguishable from real samples, both frequencies $AA_{\text{True}}$ and $AA_{\text{Synth}}$ should converge to 0.5 at large $N_s$. $AA_{\text{TS}}$ can be evaluated both with train or test samples and the privacy loss indicator is defined as $\text{Privacy loss} = AA_{\text{TS}}^{\text{test}} - AA_{\text{TS}}^{\text{train}}$ and is expected to be strictly positive. Fig. 14 shows the comparison of $AA_{\text{TS}}$ and privacy loss values obtained with our two models, demonstrating that the pre-trained RBM clearly outperforms the

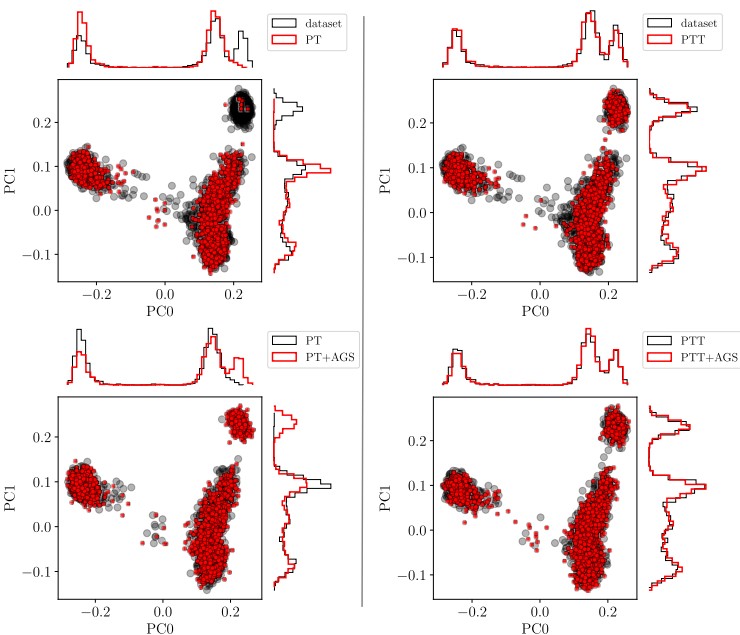

Figure 12: Comparison of the sampling results of PT (left) and PTT (right). The top row corresponds to sample generated with each methods compared with the dataset distribution. The bottom row compares samples obtained by performing $10^6$ AGS starting from the PT and PTT samples with their starting position.

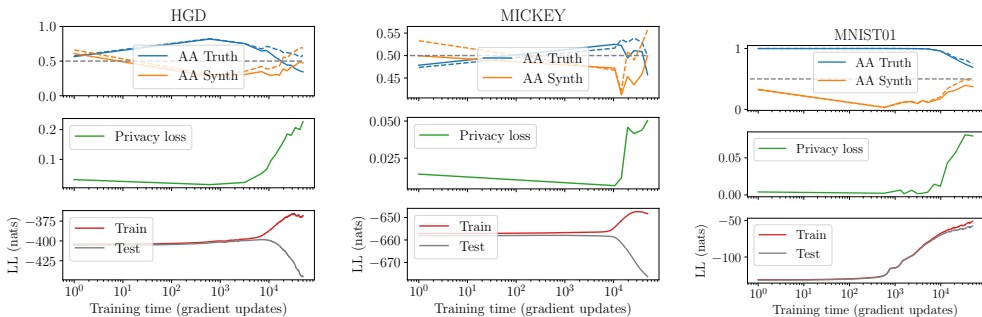

Figure 13: Evolution of the AATS (first row) and privacy loss (second row) indicator during the training of pretrained RBMs. The online estimation of the train and test LL is given at the third row. The overfitting is clearly detected in the HGD and MICKEY dataset and the divergence between the train and test LL coincides with a sharp rise in the privacy loss.

other model, and even achieves better results ($AA_{\mathrm{TS}}$ values much closer to 0.5) than those discussed in (Yelmen et al., 2021).

## J EXTRA FIGURES

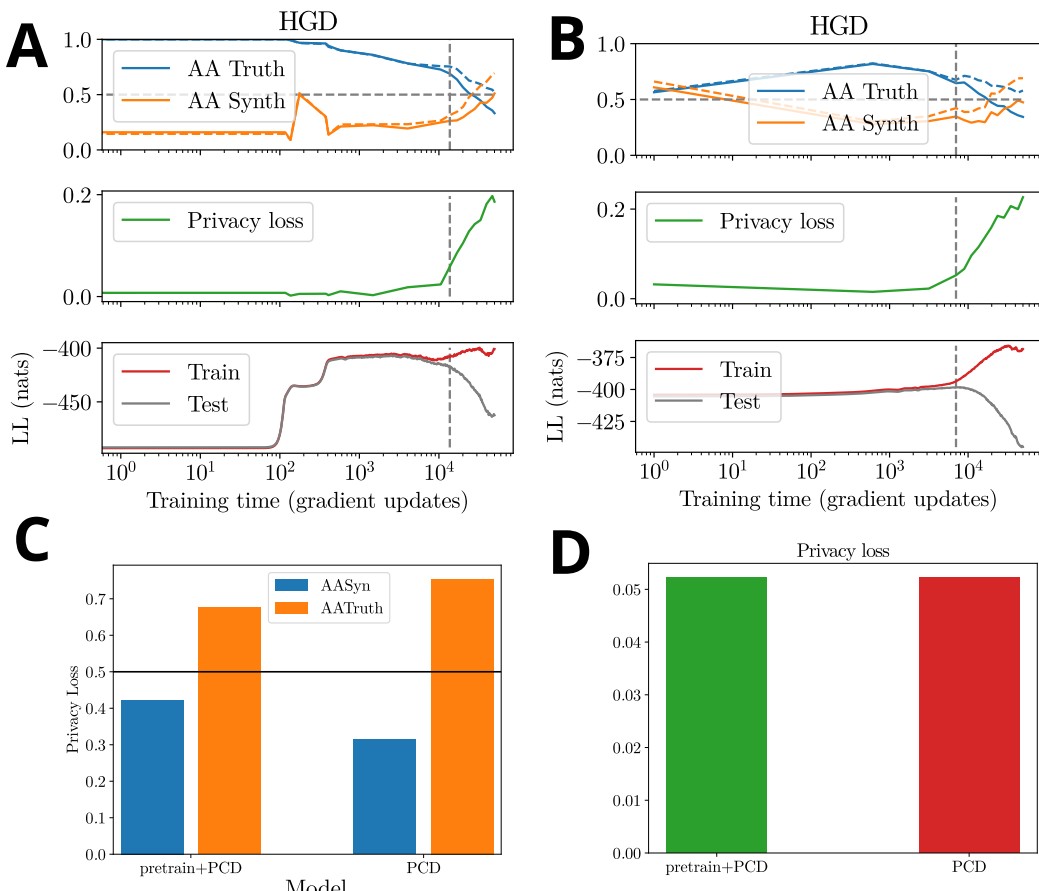

Figure 14: Comparison of the evolution of overfitting metrics along the log-likelihood between a RBM trained from scratch using PCD (**A**) and a pretrained RBM (**B**). (**C**) shows the comparison between the AASyn and AATruth between the two machines. The machines were selected in order to have a comparable privacy loss (**D**), right before starting to overfit.

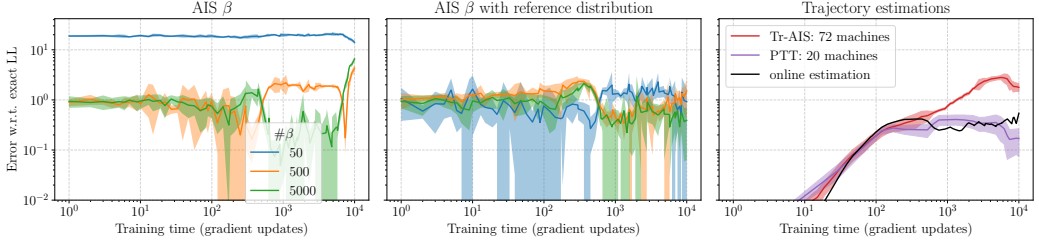

Figure 15: Comparison of log-likelihood estimation w.r.t. the exact ll on the MNIST-01 dataset.

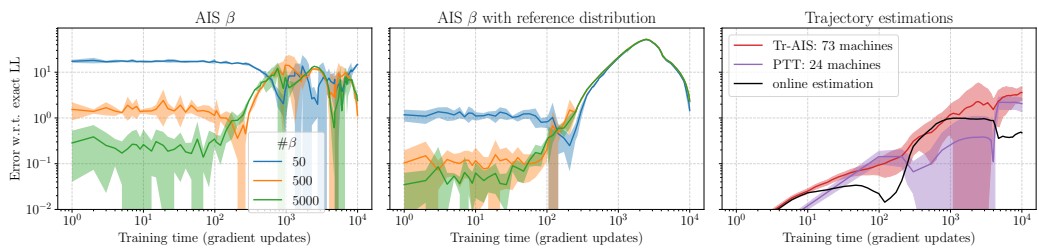

Figure 16: Comparison of log-likelihood estimation w.r.t. the exact ll on the MNIST dataset.

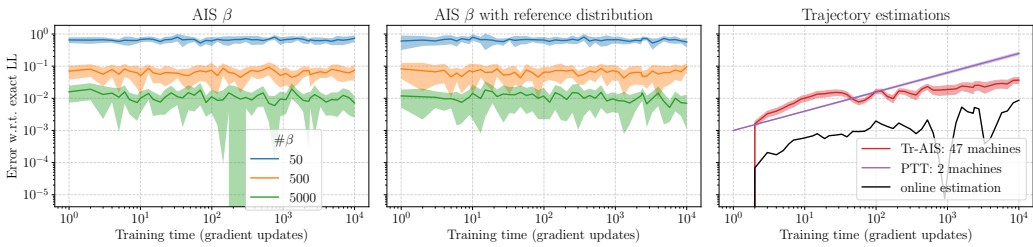

Figure 17: Comparison of log-likelihood estimation w.r.t. the exact ll on the Ising dataset.

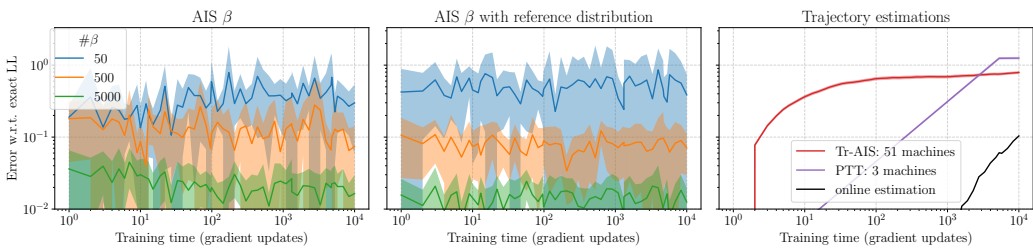

Figure 18: Comparison of log-likelihood estimation w.r.t. the exact ll on the MICKEY dataset.

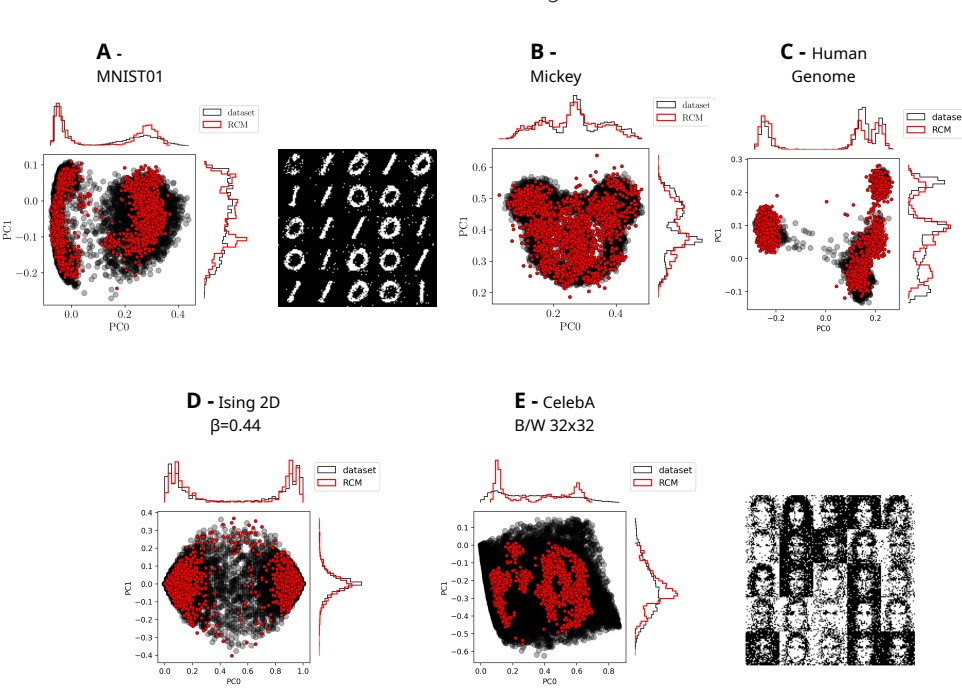

Figure 19: Samples generated with the low-rank RBM trained with 3 directions plus a bias. We show the samples generated by the pre-trained machined for the 5 datasets discussed in the paper and projected on the PCA of the dataset. For the image datasets, we also show the images generated.

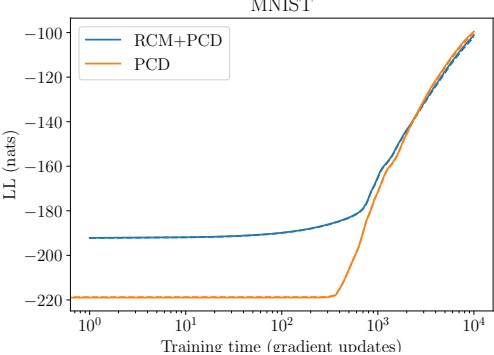

Figure 20: We show the evolution of the tr-AIS log-likelihood in the training of the entire MNIST dataset (containing digits from 0 to 9) with and without the pre-training. The pre-training does not suppose a particular advantage but neither is a disadvantage.

