# OpenReview forum: "Fast training and sampling of Restricted Boltzmann Machines"
_ICLR.cc/2025/Conference — ICLR 2025 Poster_

### Official Review · Reviewer_mx3V · 2024-10-27

**Soundness:** 2
**Presentation:** 2
**Contribution:** 2
**Rating:** 3
**Confidence:** 4

**Summary:**

This paper studies algorithms for the training of Restricted Boltzmann Machines (RBMs).  It argues that "highly structured" data require different algorithms than those that have been successful for, e.g., image datasets.  There are three algorithmic ideas that are discussed: 1) Pre-training an RBM using an "exact" procedure that produces low-rank weight matrices; 2) Estimating log-likelihoods using annealed importance sampling across steps of a training run; and 3) Using parallel tempering for sampling, again using different steps of training. Evidence for the efficacy of these procedures is provided in the form of curves from training runs on a few small datasets.

**Strengths:**

1. The idea of low-rank pre-training is interesting and seems like it could be useful if it scaled up.

2. The idea of doing AIS across the training run is creative and clever.

3. Parallel tempering across training steps seems new.

**Weaknesses:**

1. I think this paper has a somewhat limited audience.  It mostly builds upon work from a small group of authors, using language most familiar to that community.  (For example, one person's work is cited thirteen times in the references.)  A significant amount of jargon is used that keeps this from being a readable stand-alone paper.  This is coupled with heuristic explanations for things that appear to rely on sharing the particular statistical mechanical point of view of this subcommunity.

2. Much of the motivation for the work centers on "highly structured" data, which is not defined clearly.  The authors indicate that this corresponds to the existence of clusters.  The paper does not show examples of the methods succeeding or failing in the presence of this structure.  For example, the Celeb-A dataset is given as an example of a dataset in which there are not clusters and so it is not "highly structured".  However, Figure 2 does not seem to show us that this matters for the pre-training procedure.  Figure 15 is
similar.  Why does one conclude that the bottom row of Fig 2 and Fig 15 are significantly different from what we see in the top row of Fig 2?

3. The main text is highly verbose, with most of the actual concrete content being in the appendices.  I don't think anything novel is introduced until page six.

4. I find it difficult to appreciate precisely what the contribution of Section 4 is.  As I understand it, the insight is "do Decelle & Furtlehner (2021a) before you do PCD".  This is useful information, but between this section and Appendix A, I'm not sure where the boundary is between this and D&F (2021a).

5. While the ideas of section 5 are interesting and Figure 3 is intriguing, the empirical results are at the level of "preliminary findings" on a single small problem.  Even with the vastly smaller compute resources of 15 years ago, RBM researchers were studying larger problems.

6. The title is too broad relative to what the paper delivers.

Typos:
 - L161-162: "two slow"
 - L478: "exchanges parameters" but I think you mean "exchanges configuration".
 - L775-776: \bar{u} vs \hat{u}.
 - L836-837: "gradient is convex" -- surely you mean the training objective is convex in the parameters.

**Questions:**

1. Why didn't you apply this to larger problems?

2. What are situations where the pre-training fails?

3. Is PTT useful for generating samples during training, using only earlier parts of the training run?

---

> ### Author Response · Authors · 2024-11-14
> **Answer to the weaknesses and questions (part 1/2)**
>
> **Weaknesses**
>   1. We appreciate the reviewer’s feedback on the accessibility of our paper. We know how important it is to make our work as understandable as possible for a wide audience. Therefore, we have included an appendix to explain the definitions and concepts from phase transition theory. We are happy to provide further explanation if there are particular terms or sections that could benefit from additional context. Although our work contains elements from the field of statistical physics, we believe that it also makes an important contribution to the fields of computer science and machine learning by providing a practical algorithm for training, evaluating, and sampling Restricted Boltzmann Machines that has numerous applications.
>
>
>   2. We appreciate the reviewer’s comments on the term "highly structured dataset". While there is no precise mathematical definition, we would like to clarify what we mean by this. By "highly structured" we refer to datasets that exhibit certain notable characteristics: (i) the presence of visible and well-separated clusters in PCA projections and (ii) the difficulty Monte Carlo methods have in jumping between isolated clusters, often due to excessively long mixing times. As for "unstructured" datasets, we show that our method is effective on these as well, although it does not offer much advantage for long training times, since PCD has difficulty thermalizing both with and without pre-training, rendering pre-training mostly useless for long training times. Fig. 15 shows the training of the full MNIST dataset, where the pre-training becomes useless (in terms of matching log-likelihoods) very early in the training, from $10^3$ updates. Fig. 15 illustrates the training of RBMs on the full MNIST dataset, where the advantage of pre-training diminishes as early as $10^3$ updates, making it ineffective in terms of matching log-likelihoods beyond this point. In contrast, for the MNIST 0-1 subset shown in Figure 2 (top), the pretrained model consistently outperforms the standard PCD model in terms of log-likelihood and in a proper balance of the different peaks of the histogram of the projected generated data. For the CelebA dataset (Figure 2), the pre-trained RBM also reaches a higher log-likelihood than the standard model, but both approaches seem to converge toward similar values over time.
>
>   3. Our paper focuses on the sampling problems that highly structured datasets pose when training and evaluating RBMs, and suggests strategies to overcome these problems. The long introduction focuses on both reviewing previous work and explaining the physical reason why many of these sampling strategies may fail at the different training stages. Based on these conclusions, we propose a more appropriate training and sampling strategy as well as a new method to compute log-likelihood during training. We want to stress that while  Monte Carlo is an easy method to implement, in many cases, like in RBMs, it is extremely difficult to implement and control it correctly. While we do not believe that pre-training is the main contribution of our work (we rather think that it is the trajectory AIS measure of log-likelihood or the PTT), we have made innovations that are necessary to make the mapping of D&F 2021 usable in real datasets, which we present on page 5. We can try to make them clearer.
>
>   4. The work D&F 2021 propose a rather theoretical setting to learn a low-rank RBM and it is tested only on very simple, low-dimensional synthetic data sets with specific modes and regular features to cover the low-dimensional space, and mainly focuses on theoretical aspects. The first added value of the present work is to move from theory to practice, where many details need to be specified to make the technique work for arbitrary data. In this previous work, the algorithm was tested up to only 2 constrained directions. In our construction, we reach 4 constrained dimension thanks to including the possibility of adding a trainable bias, which also turns out to be crucial to train decent low-rank RBMs with image data. Moreover, D&F's work never controlled the quality of the low-rank RBM generated samples but only that of the low-rank Coulomb machines. It turned out that when dealing with real data, one needs to carefully correct the entropy term to ensure that true RBM equilibrium configurations can be obtained by the static Monte Carlo procedure, which is crucial to sample fast the trained machines, but also to properly train the low-rank RBMs. We will present these improvements in more detail in the final version.

---

> > ### Author Response · Authors · 2024-11-14
> > **Answer to the weaknesses and questions (part 2/2)**
> >
> > **Weaknesses**
> >
> > 5. We are unsure if we fully understand the reviewer's comment. Calculating the exact log-likelihood is only feasible for a small number of hidden or visible nodes, as it requires an exhaustive enumeration of all $2^{\mathrm{min}(N_v,N_h)}$ possible states. Since this count grows exponentially, even modest increases in $\mathrm{min}(N_v,N_h)$ quickly make exact calculations infeasible, regardless of computer advances over the past 15 years. For larger RBMs, as shown in Fig. 2, we can estimate the log-likelihood; however, these estimates can only be compared to other approximate methods rather than exact values. Figure 3 illustrates a comparison with exact values, which is possible only when the full enumeration is computationally feasible. This approach of validating approximate methods with exact values where possible is common practice in papers introducing new techniques for estimating the partition function. We would be grateful if the reviewer could share any specific types of analysis they had in mind.
> >
> >
> > **Questions**
> >
> >   1. We apply it to CelebA $N_v=32^2$ and $N_h=500$, how large should we go ?
> >   2. The problem in general came from numerical issue rather than from the method. We observed that when doing the projection in the low-dimensional space, if part of the dataset lied on the border of the domain, there can be convergence issue, mainly due to the saturation of the hyperbolic tangent. Otherwise if the discretization step is excessively large, it may lead to the appearance of spurious probability modes that can't be detected, and some data clusters might be overlooked.
> >   3. Basically yes, you need to have several machines over the learning trajectory to sample.

---

### Official Review · Reviewer_dgLM · 2024-10-28

**Soundness:** 2
**Presentation:** 3
**Contribution:** 3
**Rating:** 5
**Confidence:** 5

**Summary:**

This research proposes an efficient training approach for structured data in RBMs by employing pre-training based on simple convex optimization, which significantly facilitates learning for structured datasets. Furthermore, the study introduces a novel sampling and log-likelihood evaluation method that leverages the model's learning process, differing from conventional Parallel Tempering.

**Strengths:**

- The paper offers a novel contribution by proposing a pre-training technique and a new sampling approach for RBMs inspired by their thermodynamic properties. This builds on the existing theoretical analyses of RBMs.
- To my knowledge, extending replica Monte Carlo methods to a learning trajectory is original and intriguing.
- Including a specialized physics background in the Appendix makes the paper accessible even to readers without a physics background.

**Weaknesses:**

The distinction between theoretical claims and empirical findings is not clear. It would be beneficial for the authors to clarify which parts of the study are based on theoretical analysis and which are supported by numerical experiments, particularly in the context of related work. For instance, the first- and second-order phase transition claims pertain to equilibrium properties. However, it is unclear how these phase transitions are justified when updating parameters with limited samples.

- In Section 4, the paper introduces pre-training for low-rank RBMs with singular value decomposition (SVD)--based weights, aiming to avoid continuous phase transitions (second-order transitions) as structural patterns gradually emerge. It is further claimed that training can proceed quickly using the PCD method after post-pre-training. Could the authors provide a more detailed explanation for this intuition? Even if second-order transitions are avoided, if there are multiple stable clustered states, capturing multiple modes with the PCD method may be challenging and could introduce bias in the estimation. However, the paper claims, "Once the main directions are incorporated, training can efficiently continue with standard algorithms like PCD, as the mixing times of pre-trained machines tend to be much shorter than at the transitions." I believe that simulating clustered models with simple PCD often results in impractically long mixing times. Indeed, in Section 5.2, it is argued that mixing is very slow for AGS in clustered data.

- The statement "It’s also often ineffective with highly clustered data due to first-order phase transitions in EBMs, where modes disappear abruptly at certain temperatures, as discussed by Decelle & Furtlehner (2021a)" suggests that using PT becomes challenging because the learned RBM exhibits a first-order transition at specific temperatures. However, does the existence of a first-order transition in the learned RBM typically occur regardless of the statistical model being learned? For example, if learning a model without a first-order transition, such as the Ising model without a local field, does a first-order transition still arise in the learned RBM? This seems somewhat nontrivial.

- In the phase diagram of A. Decelle’s Thermodynamics of Restricted Boltzmann Machine and Related Learning Dynamics does not appear to be a first-order transition, and the AT line may suggests continuous phase transitions dominated by Full-step RSB. Thus, the claim regarding first-order transitions requires further elaboration. If a first-order transition is present, it would be essential to validate this by examining the free energy from the equilibrium state of the learned model, which could likely be accomplished by evaluating the partition function using the proposed method.
- If a first-order transition does exist, then the exchange probability in PT would approach zero near the transition. Has this phenomenon been observed? Additionally, it would be helpful to evaluate the round-trip rate of PT and PTT.
- While it is argued that preparing models at different temperatures is challenging for PT, it should be noted that the proposed approach also requires storing models during the learning process.
- The CelebA data in Figure 2 appears to be truncated.

Because the high performance has been verified numerically, the score can be raised if the above statement is cleared.

**Questions:**

- Does critical slowing down occur in the energy-based model when the hidden variables are traced out, or does it occur in the joint distribution that includes the hidden variables? If the phase transition occurs in the joint measure, does the traced-out distribution also exhibit a phase transition?
- What is the definition of $\bar{u}$?
- Could the authors provide a detailed derivation of Equation (4)? The terms $\bar{u}_{a}$ and $\eta_{a}$ are currently undefined.
- The phrase "a direction $\bar{u}_0$ is used for the magnetization $m_0$ that is only present in the bias term" is unclear. Could you explain this in more detail?
- Is it possible to learn DBM without pre-training using the pre-training with weights introduced by [1] ?

[1] Yuma Ichikawa and Koji Hukushima, Statistical-mechanical Study of Deep Boltzmann Machine Given Weight Parameters after Training by Singular Value Decomposition.

---

> ### Author Response · Authors · 2024-11-16
> **1/4**
>
> We thank the referee for all the interesting question posed. We answer all them one by one below.
>
> > *The distinction between theoretical claims and empirical findings is not clear. It would be beneficial for the authors to clarify which parts of the study are based on theoretical analysis and which are supported by numerical experiments, particularly in the context of related work. For instance, the first- and second-order phase transition claims pertain to equilibrium properties. However, it is unclear how these phase transitions are justified when updating parameters with limited samples.*
>
> Concerning the 1st and 2nd order phase transitions, we agree that the existence of such transitions is a purely equilibrium phenomenon. However, our study focuses on the associated dynamical effects in the sampling process, which should still manifest even if the transition is a crossover due to limited sample sizes or other out-of-equilibrium phenomena. At the dynamical level (where the temperature is changed, or the parameters adjusted), the effect of the transition takes place not only at the critical point by also in its neighborhood (for instance in 2nd order phase transition the increase in the correlation length manifests itself as we approach the transition). It is also important to note that the existence of at least the first learning transitions in finite minibatch training has already been confirmed in (Bachtis et al., 2024) using finite-size scaling techniques.
>
> > *In Section 4, the paper introduces pre-training for low-rank RBMs with singular value decomposition (SVD)--based weights, aiming to avoid continuous phase transitions (second-order transitions) as structural patterns gradually emerge. It is further claimed that training can proceed quickly using the PCD method after post-pre-training. Could the authors provide a more detailed explanation for this intuition? Even if second-order transitions are avoided, if there are multiple stable clustered states, capturing multiple modes with the PCD method may be challenging and could introduce bias in the estimation. However, the paper claims, "Once the main directions are incorporated, training can efficiently continue with standard algorithms like PCD, as the mixing times of pre-trained machines tend to be much shorter than at the transitions." I believe that simulating clustered models with simple PCD often results in impractically long mixing times. Indeed, in Section 5.2, it is argued that mixing is very slow for AGS in clustered data.*
>
> The reviewer's intuition is correct. The effectiveness of pretraining strongly depends on the properties of the dataset. In many cases, even after learning the first modes, the mixing time following the transitions remains too long for PCD to function properly. This is why the benefits of pretraining cannot be extended throughout the entire training process. However, we typically observe that after crossing the first transitions, the mixing time decreases by several orders of magnitude compared to the values observed during the transitions, which, for some datasets, makes it feasible to safely continue using PCD for a portion of the training. This was for instance measured in  (Béreux 2023).  That said, the mixing time generally increases as training progresses, eventually driving the system out of equilibrium, requiring alternative algorithms. We have now tried to clarify this point in the revised version of the manuscript.

---

> ### Author Response · Authors · 2024-11-16
> **2/4**
>
> >*The statement "It’s also often ineffective with highly clustered data due to first-order phase transitions in EBMs, where modes disappear abruptly at certain temperatures, as discussed by Decelle \& Furtlehner (2021a)" suggests that using PT becomes challenging because the learned RBM exhibits a first-order transition at specific temperatures. However, does the existence of a first-order transition in the learned RBM typically occur regardless of the statistical model being learned? For example, if learning a model without a first-order transition, such as the Ising model without a local field, does a first-order transition still arise in the learned RBM? This seems somewhat nontrivial.*
>
> Our claim is based on the work from D\&F 2021 for which it is quite clear that a change of temperature (a global parameter $\beta$ in front of the energy) can unbalance the minima when we have a bias (see app. G line 1328 to 1348). From a mathematical point of view, it is possible to provide the further analysis that is now included in the new SI-H "FIRST ORDER TRANSITIONS IN RBMS" Section.
>
> We consider a Curie-Weiss model over $s_i = \pm 1$ with $E= -\sum_{i<j} s_i s_j$ at a given temperature $\beta_{CW}$. We proceed with learning using Bernoulli $\sigma_i = \{0,1\}$ visible variables and one hidden gaussian node $\tau$ of variance $1/N$ (for simplicity) and with a field on the visible nodes. The Hamiltonian is given by $E = -\sum_{i} \sigma_i w_i \tau - \sum_i s_i \eta_i$. In such setting, the free energy is given by
>     \begin{align*}
>         -f &= \frac{m_{\tau}^2}{2} - N^{-1} \sum_i\log\left[ 1 + \exp(w_i m_{\tau} + \eta_i)\right] \text{ with } m_{\tau} = \frac{1}{N}\sum_i w_i {\rm sigm}\left[1 + \exp(w_i m_{\tau} + \eta_i)\right]
>         %m_{\tau} &= \frac{1}{N}\sum_i w_i {\rm sigmoid}\left[1 + \exp(w_i m_{\tau} + \eta_i)\right]
>     \end{align*}
>     and we can identify the optimal learned parameters of the RBM
>     \begin{equation*}
>         w_i = 2 \sqrt{\beta_{CW}} \text{ and } \eta_i = -2 \beta_{CW}
>     \end{equation*}
>     Starting from the optimal RBM, we can multiply the energy of the system by a factor $\beta$ and compute the free energy. We show on Fig. 9 of the SI H, we illustrate the result on ten different values of $\beta$ from $[0.8,1.05]$ and we clearly observe the presence of a first order transition: when $\beta$ is lowered the local minima with the large magnetization is gradually destroyed: first it is subdominant and then disappear. When $\beta$ is increased, we observe the exact contrary behavior, thus showing a clear first order transition at $\beta=1$.
>
> We also demonstrate the same phenomenon using an RBM trained on a real dataset (HGD). Since this dataset is intrinsically low-dimensional, we efficiently apply the Tethered method from (Béreux 2023) to compute the potential and probability distribution as a function of \(m\). In Fig. 11, we show that performing an annealing experiment by multiplying the energy by a factor \(\beta\) reveals a first-order transition. For smaller values of \(\beta\), the cluster (top-right) becomes sub-dominant and eventually disappears below $\beta=0.7$. For larger \(\beta\) values, the central cluster vanishes, providing a clear indication of another first-order transition.
>
> > *In the phase diagram of A. Decelle’s Thermodynamics of Restricted Boltzmann Machine and Related Learning Dynamics does not appear to be a first-order transition, and the AT line may suggests continuous phase transitions dominated by Full-step RSB. Thus, the claim regarding first-order transitions requires further elaboration. If a first-order transition is present, it would be essential to validate this by examining the free energy from the equilibrium state of the learned model, which could likely be accomplished by evaluating the partition function using the proposed method.*
>
> The phase diagram in this reference is presented within the proper context, where the absence of biases generally results in only second-order phase transition lines. For the occurrence of first-order phase transitions, we refer the reviewer to our response to their previous comment and the new SI-H.

---

> > ### Author Response · Authors · 2024-11-16
> > **3/4**
> >
> > > *If a first-order transition does exist, then the exchange probability in PT would approach zero near the transition. Has this phenomenon been observed? Additionally, it would be helpful to evaluate the round-trip rate of PT and PTT.*
> >
> > We agree with the referee that this is the typical scenario observed in physics, where only two states are considered. However, in the training of RBMs, this is not always the case, as only one of many clusters might disappear instead. This implies that the acceptance rate of PT moves can remain high because most states are still valid for both temperatures. Nonetheless, the long-term dynamics eventually forget the disappearing cluster because the simulation does not spend enough time at  low temperatures for the cluster to be renucleated.
> >
> >  This is precisely what we observe when working with the HGD dataset. While the acceptance rate suggests that PT is functioning correctly, inspecting the samples generated after a long PT run reveals that a cluster is missing. To confirm that these final samples do not accurately represent the model's equilibrium, we run a long standard Alternate Gibbs Sampling (AGS) process ($10^6$ iterations) initialized from these PT sampled configurations. In this case, the missing cluster slowly re-emerges.
> >
> > In contrast, when running the same AGS process starting from PTT-generated samples, no significant changes are observed. These findings are presented in the new Fig. 12 in the SI.
> >
> > >*While it is argued that preparing models at different temperatures is challenging for PT, it should be noted that the proposed approach also requires storing models during the learning process.*
> >
> >  We are not sure we fully understand the reviewer's comment. The issue with PT is not the preparation of models at different temperatures, which is straightforward, but rather the need to sample them across many temperatures, which are not particularly useful beyond estimating the log-likelihood of one model through thermodynamic integration.
> >
> > In contrast, samples generated at different training times are generally more valuable, as they allow tracking the learning process and investigating aspects such as overfitting and the benefits of early stopping. While it is true that this approach requires saving a few models during the training trajectory, this is a standard practice in most training workflows.
> >
> >
> > > *The CelebA data in Figure 2 appears to be truncated.*
> >
> > We thank the referee for pointing this out. We have revised the figure, and the updated version is included in the revised manuscript.
> >
> > > **Questions**
> >
> > > *Does critical slowing down occur in the energy-based model when the hidden variables are traced out, or does it occur in the joint distribution that includes the hidden variables? If the phase transition occurs in the joint measure, does the traced-out distribution also exhibit a phase transition?*
> >
> > During the training of the RBM, sampling is performed on the joint distribution, so critical slowing down occurs within this context. The properties of the distribution after tracing out the hidden nodes are typically not studied. In Ref. [Roussel et al., PRE 2021], the authors show that this approach appears to improve sampling compared to Alternate Gibbs Sampling, though this improvement may not be directly related to the presence or absence of transitions. Furthermore, the authors do not explicitly compute the mixing time in their analysis.
> >
> > On the other hand, theoretical arguments presented in (Bachtis 2024) map the problem of learning the first mode in the traced-out probability of the RBM (initialized from $W$ very small) to a Mattis model (see SI F in that work), which also undergoes a second-order phase transition when the first singular value exceeds the critical value \(w_\mathrm{c}=4\).
> >
> > >*What is the definition of $\bar{u}$?*
> >
> > The definition of \(\bar{u}\) is provided in Eq. (3) as the left singular vector of \(W\), but we have identified an error. We have corrected this in the revised manuscript.
> >
> > > *Could the authors provide a detailed derivation of Equation (4)? The terms $\bar{u}_{a}$ and $\eta_{a}$ are currently undefined.*
> >
> > The term \(\eta_a\) refers to the hidden bias, as defined in Eq. (1). The term \(\bar{u}_a\) is indeed a typo; it should be \(\bar{u}_{\alpha a}\) and placed within the summation symbol. Beyond this, the only derivation involves substituting \(\bm{u}_\alpha \cdot \bm{v} = \sqrt{N_v} m_\alpha(\bm{v})\) into the marginalized RBM energy, using \(W\) as defined in Eq. (3).
> >
> > We have added a new paragraph in the new version of the manuscript and corrected the formula.

---

> > > ### Author Response · Authors · 2024-11-16
> > > **4/4**
> > >
> > > > *The phrase "a direction is used for the magnetization that is only present in the bias term" is unclear. Could you explain this in more detail?*
> > >
> > > When the visible bias of the RBM is initialized using the empirical mean, it may not lie within the intrinsic space defined by the first directions of the PCA. To address this, we expand the intrinsic space by adding a new direction. This direction is the component of the empirical mean that is orthogonal to the existing intrinsic space. However, only the visible bias lives on this component since we decompose the weight matrix on the PCA. Thus only the visible bias interacts with the magnetization on this direction (indexed as direction 0).
> > >
> > > > *Is it possible to learn DBM without pre-training using the pre-training with weights introduced by [1] ?*
> > >
> > > To be honest, it is not clear for us if it is possible or not. We would have to check precisely how it can be done to give a precise answer.

---

> > > > ### Comment · Reviewer_dgLM · 2024-11-20
> > > > **Response to 4/4**
> > > >
> > > > Thank you very much for your thorough explanation.
> > > > I now have a clear understanding of this point.

---

> > > > > ### Author Response · Authors · 2024-11-20
> > > > > **Response to "Response to 4/4"**
> > > > >
> > > > > We thank the reviewer for their comments and their willingness to understand. If the reviewer feels that a particular experiment or figure might help the discussion, we will be happy to try to produce it.

---

> > > > > > ### Comment · Reviewer_dgLM · 2024-11-25
> > > > > > **response**
> > > > > >
> > > > > > Thank you for your response.
> > > > > >
> > > > > > I now have a good understanding of the advantages of PTT over PT. Additionally, the possibility that PTT can avoid the issue of first-order transitions is quite appealing.
> > > > > >
> > > > > > While it is interesting to develop algorithms based on equilibrium properties, the approach is fundamentally empirical. Therefore, having a theoretical analysis that elucidates the superiority of PTT would be beneficial.
> > > > > >
> > > > > > Taking everything into account, I will raise my score by one point.

---

> > > > > > > ### Author Response · Authors · 2024-11-25
> > > > > > >
> > > > > > > We sincerely thank the reviewer for revisiting the review and for the kind words about the PTT.
> > > > > > >
> > > > > > > We would like to address the point regarding the "fundamentally empirical approach". For simple datasets, there are several rigorous analytical studies showing that the training dynamics undergo several second-order transitions during learning. While it is true that such precise analytical descriptions are not feasible for complex, arbitrary datasets or for training with non-convergent MCMC or small minibatches, it was recently shown in Bachtis et al, NeurIPS 2024 that the cascade of second-order phase transitions remains consistent in all these practical cases. And not only that, the characterization of these transitions was achieved in real and non-ideal trainings using well established methods of physics, such as finite-size scaling. So there is a solid theoretical basis for understanding why PTT works effectively.

---

> > > > > > > > ### Comment · Reviewer_dgLM · 2024-11-25
> > > > > > > > **response**
> > > > > > > >
> > > > > > > > Thank you for your response.
> > > > > > > >
> > > > > > > > I understand that RBMs encounter phase transitions during dynamic processes, which contributes to the difficulty. However, the claim that PPT typically works well due to the presence of these phase transitions seems to be based on empirical intuition rather than a theoretically guaranteed method.
> > > > > > > > In other words, there is a gap between the existence of phase transitions and the performance of algorthm.

---

> > > ### Comment · Reviewer_dgLM · 2024-11-20
> > > **Response to 2/4 and 3/4**
> > >
> > > Thanks to the additional revisions, I now have a better understanding of the claims regarding first-order transfer. Additionally, the explanation of mode collapse, particularly the statement "In this case, the missing cluster slowly re-emerges," has become much clearer and easier to follow.
> > >
> > > However, I still do not fully understand the advantages of saving the weights during the training process to perform parallel tempering. What are the non-trivial benefits of performing parallel tempering among models during the training process, as opposed to the more conventional approach of conducting parallel tempering along the axis of inverse temperature in a quantitative manner?
> > >
> > > I understand your point that saving models during the training flow is a standard practice. However, it is also straightforward to prepare models with different temperatures for conventional parallel tempering. Furthermore, in the case of inverse temperature, high-temperature limits are largely independent of training data, allowing clusters to be smoothed out for effective sampling. There are also systematic methods proposed for adjusting the exchange probability.
> > >
> > > I would like to understand the quantitative advantages of performing replica exchange Monte Carlo methods among models—for example, achieving faster mixing or a significantly higher round-trip rate with an extremely small number of replicas. At present, it feels like there is a heuristic element involved.
> > >
> > > If this point becomes clearer, I plan to raise my score.

---

> > > > ### Author Response · Authors · 2024-11-20
> > > > **Response to "Response to 2/4 and 3/4"**
> > > >
> > > > First a note: we do not perform PTT during training. This is something that needs to be tested in the future, but this requires a specific study comparing training methods. Rather, the point of this paper was to show that using the training trajectory for generating samples and estimating the partition function is much more efficient than using standard temperature scheme. This statement is quantitatively supported by several figures in the paper.
> > > >
> > > > Second note: The less-trained model of the PTT allows to decorrelate the chains as fast as the high temperature limit for PT. If we consider a normal PCD training, the first model is exactly the same high temperature limit (with a reference configuration), since the couplings are initially set to zero and the biases are chosen to coincide with the center of the dataset. If we consider pre-training, independent samples are obtained in the first model with 1 Monte Carlo step, since the model can be sampled with a static Monte Carlo process (not with a Markov chain MC).
> > > >
> > > > As for PTT, the comparison with standard PT is shown in Fig. 4 C1,C2 and C3, where the number of jumps between clusters is measured as a function of the total number of simulation steps performed (i.e. the total number of models or temperatures used, multiplied by the number of AGS steps performed at each model). In this figure, we compare the PTT with the standard PT and the optimized PT with a non-flat reference probability using a different number of temperatures in the ladder. In all cases, we achieve large speed increases with PTT. The reviewer suspects that the problem is that we are not properly controlling the acceptance PT rate, but we can tell him/her that this is not the case. For this project, we have tried to optimize the PT as much as possible. In particular, we considered both regular temperature ladders (as shown in the manuscript) and sets of temperatures selected so that one could minimize the number of temperatures by imposing a fixed acceptance rate of 0.24-0.3 (same criteria used for PTT). We did not find any improvement in performance of the second and the same problems. We did not show this data in the paper because it complicates the discussion too much, but it could be shown.
> > > > The problem with PT is the appearances of first order transitions at temperature. Unless we consider an extremely dense ladder of temperatures around the phase transition point(s), we are not able to get reliable results, and as samples associated to isolated clusters disappear from the sampled equilibrium distribution even if they should not according to their statistical weight, as described in the new Fig. 12.
> > > >
> > > > Concerning the online computation of the log-likelihood. For the AIS trajectory, the log-likelihood at different models is obtained using $O(N_m)$ calculations, with $N_m$ the number of models, while to do the same with temperature requires $O(N_m\times N_T)$ with $N_T$ the number of temperatures. This last scaling makes the LL online computation (that is, at all the training updates) impossible in practice using the standard AIS method. In Fig.3 (and Figs.15-18 in the SI) we show that the traj AIS is not only much faster, but also much more reliable than previous methods in controlled experiments where the exact value is known.
> > > >
> > > > PT works well if the temperature transitions are second order. This is the case, for example, with the Ising model or the Edwards-Anderson model in statistical physics. The problem is that for multimodal distributions that are not centered around zero (as opposed to $p(m)$ or $p(q)$ in the previously mentioned models at low temperature), a change in temperature means the discontinuous disappearance or appearance of a lump. In this case, the exchange of configurations between neighboring temperatures is not only not useful, but can even be harmful, as shown in Fig. 12 for the HGD. The PTT trajectory, on the other hand, does not undergo first-order phase transitions, so it does not suffer from this obstacle and operates in the traditional basic structure, which is perfect for PT-like algorithms.
> > > >
> > > > In practice, much fewer models need to be used when using PTT than PT. The comparison of the computational cost of both sampling methods is quantitatively compared in Fig.3 when the time is multiplied by the number of parallel models used for sampling. If the reviewer wants to see the number of jumps per sampling time (not multiplied by the number of models), we can show that. We can also compare the autocorrelation times of the PT trajectory, as was done for the PTT in SI B.2. The only problem is that the PT relaxation time is much slower, so the analysis takes a bit more time if we want to control the performance with $N_T$. But we can show preliminary figures at this stage (that we already have) if the reviewer thinks they are useful to clarify the issue.

---

> ### Comment · Reviewer_dgLM · 2024-11-20
> **Response to 1/4**
>
> > Concerning the 1st and 2nd order phase transitions, we agree that the existence of such transitions is a purely equilibrium phenomenon.
>
> In other words, is it accurate to interpret that, instead of focusing on the dynamic phase transitions arising in non-equilibrium processes, your algorithmic improvements are grounded in the properties of equilibrium states and their associated critical phenomena?
>
> > The mixing time decreases by several orders of magnitude compared to the values observed during the transitions, which, for some datasets, makes it feasible to safely continue using PCD for a portion of the training.
>
> I am not entirely sure I understand. Once clusters form, the RBM's block Gibbs sampling becomes confined to those clusters. For example, with MNIST, starting with an image of the digit "1" and running the RBM rarely results in transitions to images of other digits.
> With respect to your comment that "the mixing time is reduced by several orders of magnitude compared to the values observed during the transition," could it be that pretraining biases the states of the learned clusters, leading to transitions that remain confined within those clusters? This wouldn’t typically be described as faster mixing of the Markov chain.
> In PCD and CD, Gibbs sampling is conducted in parallel using diverse initial values, with the randomness from these initializations enabling the practical generation of high-quality images.

---

> > ### Author Response · Authors · 2024-11-20
> > **Response to "Response to 1/4"**
> >
> > > *In other words, is it accurate to interpret that, instead of focusing on the dynamic phase transitions arising in non-equilibrium processes, your algorithmic improvements are grounded in the properties of equilibrium states and their associated critical phenomena?*
> >
> > Yes, our algorithmic improvements are indeed based on the equilibrium properties of the model at different training times. We rationalize the performance of the MCMC sampling methods in terms of the equilibrium phases that occur during training and the evolution of the landscape. In doing so, we need to address key challenges such as critical slowdown and phase coexistence, which hinder both efficient sampling and accurate gradient estimation.
> >
> > > *I am not entirely sure I understand. Once clusters form, the RBM's block Gibbs sampling becomes confined to those clusters. For example, with MNIST, starting with an image of the digit "1" and running the RBM rarely results in transitions to images of other digits. With respect to your comment that "the mixing time is reduced by several orders of magnitude compared to the values observed during the transition," could it be that pretraining biases the states of the learned clusters, leading to transitions that remain confined within those clusters? This wouldn’t typically be described as faster mixing of the Markov chain. In PCD and CD, Gibbs sampling is conducted in parallel using diverse initial values, with the randomness from these initializations enabling the practical generation of high-quality images.*
> >
> > At the beginning of training, the distribution of the model is essentially Gaussian, so it is easy to sample. For clustered datasets, the initial encoding of the dominant modes leads to models that resemble Gaussian mixtures, with distinct and isolated clusters gradually emerging as training progresses. This behavior, which is described in practice in Béreux (2023) and also analytically in Bachtis (2024), poses a major challenge: Transitions between clusters require crossing intermediate regions with a probability close to zero, which effectively prevents jumps. Thus, the difficulty lies not only in the critical slowdown caused by the emergence of new modes, but also in the exponential slowdown caused by sampling a disjoint, multimodal distribution.
> >
> > As training progresses and more modes are learned, the initially infinite barriers between clusters become large but finite to fit the statistics of the dataset. Although the distribution remains multimodal, the transitions between clusters occur on much shorter time scales. In Fig. 8 of Béreux (2023) for MNIST-01, the integrated autocorrelation time before the training update exceeds $10^7$ MCMC steps (to the extent that it cannot be accurately estimated due to time constraints). After the first two transitions, the autocorrelation time decreases to about $10^4$ MCMC steps. This emphasizes the effectiveness of pre-training by circumventing the need to sample extremely difficult models for gradient estimation.
> >
> > By initializing the persistent chains in standard PCD training with equilibrium samples from the pre-trained low-rank model, we ensure two important results: (1) the model is properly trained at this stage and (2) the persistent chains serve as accurate representations of the model's equilibrium measure. We also emphasize that the samples included in the persistent chain are randomly drawn by a static Monte Carlo process, so there is no particular difference in terms of randomness compared to a standard PCD. We would also like to clarify that the randomness of CD does not help to generate high quality samples. CD generally trains models that are not able to generate samples at the level of the equilibrium measure as discussed in many works  (Salakhutdinov & Murray, 2008; Desjardins et al., 2010; Decelle et al., 2021 in the paper).
> >
> > Finally, we understand that the reviewer’s concerns relate to the post-training phase, particularly with respect to mixing times during PCD training. While we only perform 100 MCMC steps per update, even if the mixing times are longer, it is important to emphasize that PCD involves a slow annealing of the permanent chain as the model parameters are slowly changed. This gradual adjustment makes it easier to approximate equilibrium samples than when starting from random initial configurations. Since the parameters evolve very slowly, the initial equilibrium measure is changed only slightly at each step, and it is important to remember that newly emerging modes continuously arise from splittings of previously existing modes. In general, PCD can reliably approximate equilibrium models as long as the mixing time is not many orders of magnitude larger than the number of MCMC steps performed.

---

### Official Review · Reviewer_axdy · 2024-10-30

**Soundness:** 3
**Presentation:** 3
**Contribution:** 3
**Rating:** 6
**Confidence:** 4

**Summary:**

The claimed novelties of this work are twofold.
First, this paper proposes low-ranking training of RBMs by directly encoding the principal components throughout a convex-optimization process. This pre-training component proves to be very efficient when data are particularly clustered. In such cases, target densities are highly multimodal, and the model struggles to "discover"all the modes from scratch during training without the pre-training phase. This autonomous discovery of new modes is often associated with second-order phase transitions, similar to systems from statistical mechanics, where critical slowing down prevents the discovery of all modes in finite time efficiently.

As a second contribution, the paper also investigates how to use a variation of parallel tempering (PT) algorithms, termed parallel trajectory tempering, to sample more efficiently and obtain log-likelihoods estimates. In simple terms, parallel trajectory tempering (PTT) essentially relies on the same idea of parallel tempering of swapping between models at different temperatures using the Metropolis rule (and therefore retaining detailed balance). However, differently from PT, PTT swaps a full set of parameters $\Theta^t$ instead of the temperature $\beta$ only. In that sense, it can be thought of as a generalization of PT.

Numerical experiments in Fig. 2 prove the pre-trained low-rank RBM to be more capable of identifying all modes in highly clustered data, while Figs. 3-4 show that PTT allows more accurate loglikelihood estimation and faster yet more efficient sampling from all modes of distribution compared to standard alternate Gibbs sampling (AGS).

**Strengths:**

- The paper is well-written and easy to follow.
- It represents a pleasant read that is accessible to a broad audience.
- The literature review and related work section read well and are exhaustive.
- The idea of pre-training the RBM to encode the principal components is simple yet very effective.
- Leveraging the analogy between critical slowing down and the struggle of RBM during training to be ergodic and discovering all modes of the distributions is elegant and intuitive (though I suppose this is not a novelty of this paper, it is very nicely pictured in the introduction).
- The numerical experiments look solid and aligned with the theoretical insights given in the main text.
- I have not thoroughly checked the mathematical details in the appendix, but at first glance, they look good.

**Weaknesses:**

- I find it a bit challenging to identify the two main contributions in the paper as those are totally disentangled in their presentation between Sec. 4 and Sec. 5.2. I strongly recommend adding a list of bullet points at the end of section 1 to clearly list the contributions of work and crossref to the corresponding point in the paper. This would substantially help navigate the paper.
- I find that the structure of sections 5.2 and 5.2.1 can be improved. In particular, I find it confusing that Parallel Trajectory Tempering is introduced in section 5.2, and Parallel Tempering approaches are discussed in section 5.2.1. I find this logically inefficient as I believe that a more natural yet easier-to-follow flow would be to first introduce Parallel Tempering approaches and then explain what makes PTT different compared to existing approaches from the literature. As this is one fundamental contribution of this work I believe it is crucial to rework these sections such that the actual novelty emerges more clearly from the discussion.
- The discussion around eq. (4) is rather crucial for the paper as it represents one of the main contributions of this work. Currently, the novelty with respect to Decelle and Furtlehner (2021a) is not very clear to me, and I would appreciate it if the authors could elaborate more on this. Moreover, what's the intuition behind the "magnetizations" along each of the singular vectors? Is there any correspondence with the magnetization as a physical observable? As far as I understand, those should be the projections along the unitary vectors of the visible variable. Is that correct? If all my understanding is correct, then the new contribution of this work is to use a bias initialization along a direction $\boldsymbol{u}_0$, which augments the dimensionality of the system by one in the bias direction. If all above is still all correct, I wonder the following:
    - How beneficial is to have such an augmented direction for the bias compared to the naive approach proposed in Decelle and Furtlehner (2021a)?
    - Have the authors conducted any ablation studies to compare the differences in performances between Decelle and Furtlehner (2021a) and their new approach from an empirical standpoint?

This latter point is crucial in assessing the effective novelty of this work. At the moment the reason for the lower score is primarily due to my perception of limited novelty. I am more than happy to discuss this with the authors during the rebuttal and revisit my score upon clarification of my concerns above (and below, see, e.g., the first bullet point in the **Questions** cell).

**Questions:**

## Questions, Small comments and typos
- Would it be possible for the authors to provide a sketch and pseudocode for their PTT algorithm as a standalone and in comparison to PT? This would be very helpful to get a better understanding of the contribution of this work.
- Is there any intuition behind the bump observed in Figure 3 at around $10^3$ gradient updates (left and middle plot).
- Layout: there's a problem with Figure 2. The x-axis is sometimes completely or partly cut. I strongly recommend carefully checking this, aligning the plots, and making sure such problems are removed.
- In general the authors often refer to the Appendix as SI (I assume Supplement Information). I guess this acronym has not been defined anywhere. I identify its first occurrence in line 96. Perhaps the authors can define what SI is or, alternatively just all it appendix.
- Line 235: I'd recommend adding a reference for critical slowing down. This comment applies to earlier occurrences of this concept.
- Line 459: grew -> grey
- Line 512: Banos et al. (2010) might need to be wrapped in parenthesis \cite -> \citep

---

> ### Author Response · Authors · 2024-11-16
> **1/2**
>
> We thank the reviewer for the careful reading and useful comments.
>
> **Weaknesses** (we answer following the order of the bullet points)
>
>  > *I find it a bit challenging to identify the two main contributions in the paper as those are totally disentangled in their presentation between Sec. 4 and Sec. 5.2. I strongly recommend adding a list of bullet points at the end of section 1 to clearly list the contributions of work and crossref to the corresponding point in the paper. This would substantially help navigate the paper.*
>
> We thank the reviewer for this excellent suggestion. We have now added a list of bullet points at the end of Section 1 to clearly outline the contributions and included cross-references to the relevant sections for easier navigation. We have made a common answer about the novelties of this work in the official comments section.
>
> > *I find that the structure of sections 5.2 and 5.2.1 can be improved. In particular, I find it confusing that Parallel Trajectory Tempering is introduced in section 5.2, and Parallel Tempering approaches are discussed in section 5.2.1. I find this logically inefficient as I believe that a more natural yet easier-to-follow flow would be to first introduce Parallel Tempering approaches and then explain what makes PTT different compared to existing approaches from the literature. As this is one fundamental contribution of this work I believe it is crucial to rework these sections such that the actual novelty emerges more clearly from the discussion.*
>
> We sincerely thank the reviewer for this valuable suggestion. In the revised manuscript, we have reorganized the discussion to make it more linear and better aligned with the proposed structure.
>
> > *The discussion around eq. (4) is rather crucial for the paper as it represents one of the main contributions of this work. Currently, the novelty with respect to Decelle and Furtlehner (2021a) is not very clear to me, and I would appreciate it if the authors could elaborate more on this. Moreover, what's the intuition behind the "magnetizations" along each of the singular vectors? Is there any correspondence with the magnetization as a physical observable? As far as I understand, those should be the projections along the unitary vectors of the visible variable. Is that correct? If all my understanding is correct, then the new contribution of this work is to use a bias initialization along a direction , which augments the dimensionality of the system by one in the bias direction. If all above is still all correct, I wonder the following:*
>
> > *How beneficial is to have such an augmented direction for the bias compared to the naive approach proposed in Decelle and Furtlehner (2021a)?*
>
> > *Have the authors conducted any ablation studies to compare the differences in performances between Decelle and Furtlehner (2021a) and their new approach from an empirical standpoint?*
>
> The benefit of having an augmented direction is two-fold:
>
> * The method requires evaluating a discretized multidimensional integral. Having the augmented direction allows one to learn an extra dimension without the computational overhead of discretizing it since this direction is independent of the rest of the model.
> * This extra dimension is not learned using the hidden features of the RBM but only with the bias, resulting in RBMs with far fewer hidden nodes when trained with this additional direction.
>
> The work by D&F never attempted to train the low-rank RBM on real datasets, focusing solely on simple artificial clustered data. Without incorporating the learning of biases and additional directions, their approach fails to produce low-rank RBMs capable of adequately generating images, even for a basic dataset like MNIST01. Moreover, the D&F method faces issues in generating samples from the low-rank RBM, as the generated samples do not align with the statistics of the model's equilibrium distribution. This discrepancy can be verified by running a standard MCMC simulation on the samples generated using the static Monte Carlo method proposed by D&F. To resolve this issue, it is necessary to correct the entropy computation to address the mismatch caused by the fact that the generated samples do not have the same exact magnetization as those sampled from $p(m)$.
>
> Regarding the second question, we have not yet conducted a systematic study on the performance impact of adding more directions. Preliminary results suggest that the improvement strongly depends on the dataset. For MNIST01, having at least 4 directions was crucial to achieve better performance than PCD training, whereas for the HGD dataset, 2 or 3 directions appeared to yield similar results. We plan to include an ablation study in an updated version of the paper.
> Concerning the term magnetization, the referee is right it comes from the study of spins systems in physics, we can here see it simply as the projection of the spins on a particular direction as the referee's said.

---

> > ### Author Response · Authors · 2024-11-16
> > **2/2**
> >
> > >**Questions** (we again answer following the order of the bullet points)
> >
> > > *Would it be possible for the authors to provide a sketch and pseudocode for their PTT algorithm as a standalone and in comparison to PT? This would be very helpful to get a better understanding of the contribution of this work.*
> >
> > We thank the reviewer for this suggestion. In response, we have included the pseudo-code for Parallel Tempering (PT) and Parallel Trajectory Tempering (PTT) in Section B1 of the Supplemental Information in the updated version of the manuscript. The key difference between the two algorithms lies in how the parallel models are selected. In PT, a single model is simulated in parallel at different temperatures, whereas in PTT, different models with different parameters saved at different stages of the training process are selected.
> >
> > > *Is there any intuition behind the bump observed in Figure 3 at around $10^3$ gradient updates (left and middle plot).*
> >
> > The bump could come from the effective number of independent chains used for calculation, which can decrease significantly during the annealing process if the machine has been trained beyond this time. This number of effective chains is the number of chains that has an appreciable weight in the AIS measure. We will check this.
> >
> > > *Layout: there's a problem with Figure 2. The x-axis is sometimes completely or partly cut. I strongly recommend carefully checking this, aligning the plots, and making sure such problems are removed. *
> >
> > Thank you for pointing this out. We have prepared a new figure and thoroughly checked for any errors. The revised figure is included in the updated version of the manuscript.
> >
> > > *In general the authors often refer to the Appendix as SI (I assume Supplement Information). I guess this acronym has not been defined anywhere. I identify its first occurrence in line 96. Perhaps the authors can define what SI is or, alternatively just all it appendix. *
> >
> > Thank you for pointing this out. The acronym for Supplemental Information (SI) is now defined in Section 1.
> >
> > > *Line 235: I'd recommend adding a reference for critical slowing down. This comment applies to earlier occurrences of this concept.*
> >
> > We have added the reference [Hohenberg, P. C., \& Halperin, B. I. (1977). Ts, 49(3), 435.] for the critical slowing down.
> >
> > > *Line 459: grew -> grey*
> >
> > Corrected
> >
> > >*Line 512: Banos et al. (2010) might need to be wrapped in parenthesis \cite -> \citep*
> >
> > Now all references are cited with citep

---

> > > ### Comment · Reviewer_axdy · 2024-11-19
> > > **Acknowledgment of rebuttal**
> > >
> > > I thank the authors for thoroughly updating the manuscript and answering all my concerns in great detail.
> > >
> > > I think this work is valuable and brings some new interesting insights.
> > > However, I perceive the novelty of the work is still somehow limited, and for this reason, I will increase my score  to 6.

---

> > > > ### Author Response · Authors · 2024-11-22
> > > >
> > > > Thank you very much for your valuable feedback and for increasing the score of our manuscript.
> > > >
> > > > We noticed that you mentioned that the novelty of our work is still somewhat limited. Could you please give us more details or clarify this point? This would help us to further improve the manuscript.
> > > >
> > > > We propose a new sampling method that clearly outperforms all available methods and a log-likelihood estimation method that accomplishes the same. Both address two major challenges in working with EBMs, and the methods are applicable beyond RBMs. Recent work at ICLR, such as the work by Roussel (2023) mentioned in the manuscript, focuses exclusively on efficient sampling for RBMs, and our PTT method outperforms it in all datasets.
> > > >
> > > > Do you think that additional experiments or specific analyzes could make these contributions more convincing? Your suggestions would be invaluable for refining our work.
> > > >
> > > > Thank you again for your time and constructive feedback.

---

> > > > > ### Comment · Reviewer_axdy · 2024-11-26
> > > > > **Reply to the authors**
> > > > >
> > > > > Dear authors,
> > > > >
> > > > > Thank you for the follow-up question.
> > > > >
> > > > > As you can gauge from my evaluation, I find the paper good and interesting enough to be accepted to ICLR.
> > > > > Nevertheless, I perceive point 1. in the list of contributions to be fairly limited to be claimed as a "novelty."
> > > > > While I indeed understand the improvement, extending  [D&F 2021]  from toy data to real datasets seems a natural extension rather than a "novel contribution".
> > > > >
> > > > > In my opinion, though I might be wrong, so please take this as a personal consideration, I believe the story of the paper should have had more emphasis on the parallel tempering trajectory and log-likelihood estimation in order to make it the actual novelty more evident.
> > > > >
> > > > > Up to page 6, which is more than half the paper, only introduction, related work, and pre-training of RBMs are discussed and I think this prevents the reader to appreciate the novelty of the other contributions which came later on.
> > > > >
> > > > > The fact the the contributions are not yet entirely clear because of the "story" of the paper prevents me form gving a score higher than 6.
> > > > >
> > > > > For sure adding more analyses, that specifically focus on the benefit on the latter contribution and rewrite the story accordingly would strenghten the paper.
> > > > >
> > > > > i hope this clarifies and answer the authors' question.

---

> > > > > > ### Author Response · Authors · 2024-11-26
> > > > > > **Revised version**
> > > > > >
> > > > > > Thank you for your feedback and constructive suggestions.
> > > > > > We understand your concerns about the emphasis on the pretraining contribution in the previous version of the paper. On reflection, we agree with you that the presentation may have inadvertently given the impression that pre-training is a more central contribution than intended. We appreciate your suggestion to focus more on the parallel tempering trajectory and log-likelihood estimation, as these aspects are indeed more substantial contributions of our work. In response to your comments, we have restructured the manuscript to address these concerns:
> > > > > >
> > > > > > 1. **Revised storyline:** We have restructured the flow of the paper to ensure that the novel contributions — particularly the analysis of parallel tempering and log-likelihood estimation — are emphasized earlier and given the prominence they deserve. The discussion of pretraining has been moved to the end of the article to better align it with its subordinate role in the overall narrative.
> > > > > >
> > > > > > 2. **New analysis:** We have added a new subsection that addresses the challenges of the standard parallel tempering and the presence of discontinuous transitions. This section discusses in more detail how these issues were addressed and the impact they had on the training and sampling process. This should make the novelty of our contributions clearer to the reader.
> > > > > > We hope that these changes improve the clarity and narrative of the article, better highlight the new contributions, and address your concerns.

---

### Official Review · Reviewer_jkKM · 2024-10-31

**Soundness:** 2
**Presentation:** 2
**Contribution:** 2
**Rating:** 3
**Confidence:** 3

**Summary:**

The paper discusses approximations to train a restricted Boltzmann machine (RBM). The first is to pre-train the RBM by fitting a constrained (low-rank) form of the RBM to the low-dimensional PCA space of the data. This can help with finding a good initial solution. After this various MCMC approaches are considered to continue training.

**Strengths:**

RBMs are an important model and finding appropriate ways to train them is a topic of significant interest. The paper highlights the phenomenon of critical slowing down and how pre-training the model with a low-rank approximation of the parameter matrix can help the model overcome some of the slowing down effects.

**Weaknesses:**

The paper suffers from a lack of clarity of presentation and lack of clarity of novelty.

The paper mentions that the idea of a low-rank approach has already been used by others and it's unclear to me what novelty there is in any of the sampling approaches used after the pre-training phase.

In terms of presentation, there are notational inconsistencies and a general lack of clarity in terms of the main ideas. Fundamentally the approach of fitting a constrained model seems straightforward and indeed I believe there is a simple way to compute the projected distribution in the PCA space (using the Fourier integral representation of the Dirac delta function) which the authors do not discuss.

**Questions:**

*** introduction

Whilst the RBM is well known, it would be helpful I feel for a reader to have the definition of the model earlier in the text. It currently isn't defined until near the end of page 4. Please introduce the RBM formally earlier in the text.

Notation: inconsistent use of $N_v$ and $N_{\text{v}}$ throughout, similarly for $N_h$.

Equation 1: it might be better to write W_{i\alpha}, rather than w_{i\alpha} since w is used later for the "singular values".

*** page 2

Figure 1 isn't very easy to parse. For example the panel on race is placed more in the Mickey column than the human genome column.

*** page 5

Please clarify the difference between "model averages" and "observable averages" and the difference between using N_s independent MCMC processes and R parallel chains.

Please clarify for the reader the meaning of <v_ih_a>_D

Section 4: It is not correct that it is possible to train "exactly" an RBM with a reduced number of modes. Approximations are required, as explained in the supplementary material.

Please state what the free parameters to learn are in equation 3. If u and \bar{u} are the singular directions, then the free parameters would be w_\alpha?

In general I found the description of the low-rank approach unclear and this important section needs work to make it simpler and more clear to the reader.

For figure 14 it would be useful to show the distribution of the PCA projected data to see how well the RBM matches the projected data distribution.

It's unclear to me what contribution the authors are claiming to make. They state that the learning of the low rank parameterisation of W has been done before. Please clarify what the contributions of the paper are.


*** Section 5

I find it hard to follow why the authors are considering different sampling schemes and therefore what the aim of this section is. I presume this is considering alternative sampling approaches after the low-rank pre-training has been applied. However, I struggle to follow a clear recommendation or conclusion as to which method might be more suitable.

*** Section 6

In the conclusion the authors claim to have introduced a method that enables "precise computation of log-likelihood". I cannot see anything in the main text that relates to this. There is no experiment I can see that measures the quality of the log-likelihood approximation. Please give some evidence to support this assertion.


*** Supplementary material

The use of the term "mode" isn't very clear. The phrasing suggests that the first d modes of the maximum likelihood trained RBM should correspond to the d "modes" of the PCA solution. I'm not sure I know what this means. What are modes of a PCA solution?

The notation \hat{u} is confused with \bar{u}.

Why use $w$ here whereas $W$ is used in the main text?

The derivation is quite confusing. For example the dependence on \bar{u} in equation 7 disappears without explanation. Indeed \bar{u} seems to be never properly defined.

Please state clearly what are the parameters of the model that are being learned.

Section A.2. The claim as before of exact training is incorrectly made here.

The notation in equation 20 is confusing, such as w_{\alpha,a}=\sum_i w_{ia}u_{i\alpha} -- are arabic and latin indices meant to indicate referencing a different entity, even though both objects are labelled w?

In general I find the supplementary material confusing. I believe it is trying to fit an RBM projected to the d-dimensional subspace defined by PCA of the data to the empirical data distribution in that same subspace. However, approximations are clearly required in order to compute the projected RBM distribution. Given that, for a very low dimension d then one can easily discretise the model and carry out a simple maximum likelihood fit. If that is what is being done, it is not well explained and rather misleading (since this requires approximations itself).

An alternative (and standard) way to compute the marginal p(m) is to use the integral (Fourier) representation of the Dirac delta function. This means that the summation over v can be then carried out exactly, leaving only a d-dimensional integral to exactly compute p(m). This can also be carried out using discretisation for small d. The authors are (as I can understand) also using discretised integrals, so I'm unclear why they don't employ the standard Fourier Delta representation approach to compute p(m) -- this would seem to involve less approximations that the approach the authors consider.

---

> ### Author Response · Authors · 2024-11-16
> **1/2**
>
> We again thank the reviewer for their careful reading and constructive comments. We will attempt to respond to all comments individually below.
>
> >*The paper suffers from a lack of clarity of presentation and lack of clarity of novelty.*
>
> We apologize for any lack of clarity in the paper, which may stem from differences in language between communities. We have tried to reorganize and rephrase some parts of the manuscript to make it clearer overall. However, if the reviewer could point out certain unclear sections, we would be happy to revise them to make them more accessible.
>
> Regarding the novelty, as discussed in our general response, we respectfully disagree with the reviewer’s assessment. While the reviewer’s focus appears to be on the pretraining proposal, we consider our primary contributions to be the new algorithms for estimating the log-likelihood and accelerating the sampling process. These innovations provide substantial improvements over the state-of-the-art and are broadly applicable to energy-based models beyond RBMs. To make these diverse contributions clear and accessible, we will highlight them in bullet points at the beginning of the paper.
>
> >*The paper mentions that the idea of a low-rank approach has already been used by others and it's unclear to me what novelty there is in any of the sampling approaches used after the pre-training phase.*
>
> To our knowledge, the low-rank initialization approach has only been proposed theoretically in Decelle-Furtlehner (2021) and has not been applied to real data or used as a pre-training method. Additionally, leveraging the training trajectory to compute the log-likelihood and perform parallel tempering is a novel contribution of this work. This approach significantly enhances the performance of previously optimized methods across all datasets considered, marking an important new contribution of this work.
>
> >*In terms of presentation, there are notational inconsistencies and a general lack of clarity in terms of the main ideas*
>
> We thank you again for your careful reading and your suggestions for improvement. We have revised the paper to address almost all of your comments. In doing so, we have ensured consistent spelling and improved the clarity of key ideas in the new, quick version. The remaining suggestions will be incorporated later this week or in the final version.
>
> >*Fundamentally the approach of fitting a constrained model seems straightforward and indeed I believe there is a simple way to compute the projected distribution in the PCA space (using the Fourier integral representation of the Dirac delta function) which the authors do not discuss.*
>
> We agree with the reviewer that \( p(m) \) could indeed be computed using a Fourier representation rather than large deviations. However, the Fourier transform would still require a saddle point approximation, effectively equivalent to our large $N$ approximation in the large deviation formalism. Importantly, the Coulomb machine mapping is not primarily aimed at estimating \( p(m) \) for the RBM—this can be done through various methods. Instead, it serves to make the training process a convex optimization problem, as training the parameters of \( p(m) \) directly in the RBM framework is often unstable and challenging. We will clarify this point in the paper.
>
> >*QUESTIONS*
>
> >*Figure 1 isn't very easy to parse. For example, the panel on race is placed more in the Mickey column than the human genome column.*
>
> Thank you for pointing this out. We have adjusted the layout to improve clarity and ensure each panel aligns with the intended column.
>
> *Please clarify the difference between "model averages" and "observable averages" and the difference between using $N_s$ independent MCMC processes and R parallel chains.    Please clarify for the reader the meaning of $<v_ih_a>_D$*
>
> Thank you for pointing out these areas needing clarification. We will ensure these terms and variables are properly defined and consistently used in the final version of the paper. We have included a sentence explaining the meaning of $<v_ih_a>_D$  below Eq.(2) in the new version.
>
> >*Section 4: It is not correct that it is possible to train ``exactly" an RBM with a reduced number of modes. Approximations are required, as explained in the supplementary material.*
>
> We agree with the reviewer regarding the comment on the exactness of low-rank training. The training is performed precisely rather than exactly, with residual errors arising from the asymptotic approximation at finite \(N\) and the finite resolution of the mesh. We will systematically revise the paper to remove all the references to the exact training.

---

> > ### Comment · Reviewer_jkKM · 2024-11-22
> >
> > Thanks for making the efforts on this.
> >
> > In terms of the approach itself, my essential point is that we can (naturally) carry out the summation over h first. This leaves an expression for the partition function of the form \sum_z \prod_a cosh(W_a'v) where W_a is the vector indexed by the hidden unit a (ignoring biases for the moment). If we assume a low rank W = \sum_l a_l b_l' for any vectors a_l and b_l, then the order parameters are  b_l'v, one for each l. We can then use any method to approximate the remaining sum over v, giving an approximate value for the likelihood of p(v) under this low rank assumption. For example the Fourier approach means that we would be left with a rank d complex integral to approximate p(v). This can indeed be approximated by a saddle point method; it can be approximated by discretising the integral appropriately through the saddle point. My basic point is that this suggests that one can therefore learn any low rank W by this approach -- there's no need to consider projections to PCA directions. After learning the optimal low rank approximation, this can then be used to start the sampling for learning the full W (after relaxing the low rank constraint).  Would the authors be able to comment on this suggestion (which seems very natural to me) and why the approach based on projecting to PCA directions was taken as an alternative.

---

> ### Author Response · Authors · 2024-11-16
> **2/2**
>
> >*Please state what the free parameters to learn are in equation 3. If u and $\bar{u}$ are the singular directions, then the free parameters would be $w_\alpha$? In general I found the description of the low-rank approach unclear and this important section needs work to make it simpler and more clear to the reader.*
>
> In Eq. (3), for the low-rank method, the matrix $\boldsymbol{u}$ is determined using the PCA while $\bar{\boldsymbol{u}}$ and $\eta_a$ are randomly initialized in the Coulomb Machine and not learned. So only $w_\alpha$ and $\theta_\alpha$ are indeed adjusted through gradient ascent. We have included some sentences explaining this in the new version. Nevertheless, we will try to make an effort to improve the explanation of the pre-training method.
>
> >*For figure 14 it would be useful to show the distribution of the PCA projected data to see how well the RBM matches the projected data distribution.*
>
> Thank you for the comment. The new figure is now in the new version of the paper.
>
> >* I find it hard to follow why the authors are considering different sampling schemes and therefore what the aim of this section is. I presume this is considering alternative sampling approaches after the low-rank pre-training has been applied. However, I struggle to follow a clear recommendation or conclusion as to which method might be more suitable*
>
> We believe the reviewer is referring to Section 5, where we compare various sampling methods—Alternating Gibbs Sampling (AGS), Parallel Tempering (PT), Stacked Tempering (ST)—with the algorithm proposed in this paper: **Parallel Trajectory Tempering (PTT)**.
>
> In this section, we assume a well-trained model (obtained using the pretraining + PCD procedure described earlier) and focus on sampling independent equilibrium configurations. For highly structured (i.e., clustered) datasets, this task is inherently challenging. In standard local Monte Carlo methods like AGS, the mixing time is ruled by the typical times for chains to jump between clusters, resulting in very long equilibration times. In these kinds of models, neither PT performs well because of the existence of first order transitions in temperature: the fact that some modes of the probability measure just disappear at neighboring temperatures. This last phenomenon is now explained in detail in a new SI H section about first order transitions.
>
> To address this problem, we propose the PTT, an algorithm designed to drastically reduce equilibration time by leveraging the training trajectory of the model. To evaluate the efficiency of sampling algorithms, we track the average number of cluster jumps made by the Markov chains in average. In Figure 4, we compare PTT to previously developed methods, including AGS, ST, PT, and its variations. Our results demonstrate that PTT is significantly more efficient—often by several times or even orders of magnitude—compared to other methods.
>
> We have revised and reorganized the entire section in the new version of the manuscript to clarify the contributions and enhance the discussion.
>
> >*In the conclusion the authors claim to have introduced a method that enables "precise computation of log-likelihood". I cannot see anything in the main text that relates to this. There is no experiment I can see that measures the quality of the log-likelihood approximation. Please give some evidence to support this assertion.*
>
> We are quite puzzled by this comment. Section 5.1 in the previous version of the manuscript was devoted to using the training trajectory as an annealing process to estimate the log-likelihood (LL). We explained how this approach allows the LL to be computed online during training, as well as offline or through thermodynamic integration of configurations obtained with the PTT sampling.
>
> Figs. 3 and 11–14 in the SI (Figs. 3 and 15–18 in the current version) systematically compare in all datasets the trajectory-based LL estimates with the exact values in RBMs with a small number of hidden nodes, where all states can be enumerated. These figures also contrast our trajectory-based method with the standard temperature AIS approach, which is not only less precise but also significantly more computationally expensive than the trajectory-based method.
>
> We have included a set of bullet points in Section 1 of the paper to highlight that this is one of the important contributions of this work.

---

> ### Author Response · Authors · 2024-11-22
>
> We thank the reviewer for the suggestion, now we understand better. The reason why we want to encode the first PCA components in the $W$ matrix is to mimic what standard maximum likelihood training does in the first moments of training time, as shown in several previous theoretical and numerical works, which is encoding the first PCA components. We proposed to use this pre-training to simply skip this first part of the training and directly start training where one should have arrived if the gradient had been accurately estimated in a standard training. The reviewer suggests starting the training with another general low-rank model that is not necessarily associated with PCA. We do not know what would happen if a standard PCD training was restarted from there (whether or not the training dynamics would try to re-learn the initial PCA components and thus destroy the pre-trained model). The reliability of this idea should be tested in practice as we have no experience with it. Yet we expect this to work poorly as a projection of the data on small set of random directions usually don't show its structure which instead tend to appear on the first PC. In that case the pre-training become useless.
>
> The low-rank construction method we used can be performed with any orthonormal basis. We use the PCA because we want to mimic the training.

---

> ### Author Response · Authors · 2024-11-29
> **New version of our manuscript**
>
> Dear Reviewer,
>
> We believe that we have addressed most of your questions and concerns about the weaknesses of our work. We would also like to emphasize that while your suggestions on low-rank construction approaches could be considered for alternative methods of constructing rank-low RBMs, the actual development of low rank RBMs was not a key contribution of our work.
>
> To clarify the focus of the work, we have prepared a revised version of the manuscript. In this new version, we have placed more emphasis on sampling and log-likelihood computation to illustrate how these processes exploit the continuous nature of phase transitions during learning. We have also included a new figure (Fig. 4AB) and a new subsection discussing the limitations of the standard parallel tempering algorithm in the context of first-order transitions. In addition, we have added a new section in the SI (Section H) where we analyze in detail the landscape of probability density in a simplified environment where temperature changes.
>
> We hope that these updates add more clarity and depth to our work, and ask you to reconsider our manuscript in light of these changes.

---

> > ### Comment · Reviewer_jkKM · 2024-12-02
> >
> > SID typo
> >
> > Please stick with a single terminology -- either "appendix" or "supplementary information (SI)"
> >
> > Figure 2:
> >
> > The text refers to panels in figure 2. No such panels are labelled.
> >
> > I'm not sure it's fully explained in what is happening here. Is it that a method (such as AIS) is used to approximate the gradient and update the parameters, with the difference between the true LL and the approximate LL being plotted?
> >
> > What is the source of the error bars in this figure? There's no mention of repeated experiments.
> >
> > Panel "A" will present to the reader an obvious question, namely why does AIS with more steps actually perform worse than AIS with fewer steps around 1000 gradient updates? It would be good if possible to relate this to a phase transition.
> >
> > Please be more specific about the non-flat reference distribution and state in the figure caption exactly what this is.
> >
> > Panel "C" isn't properly explained and I don't understand what is being shown here. In general I feel it is poor practice to present results out of order -- showing things that cannot be understood by the reader until further material has been revealed.
> >
> >
> > Page 6
> >
> > I don't feel that TR-AIS is properly explained here. Please write an algorithmic description of this approach.
> >
> > What is "Alternative" or "Alternate" Gibbs sampling (the authors seem to flip between these two terms)? Do the authors mean "alternating"? Please also explain this to a non-expert in this area. Also the authors assume that there is only one way to do Gibbs sampling, whilst in practice there are many ways to implement Gibbs sampling. The authors choose the standard approach (alternating between conditioning on all hiddens and then conditioning on all visibles) but never explain this to the reader.
> >
> > I don't understand what "equilibrium measure of RBMs" means. The RBM is a defined distribution -- there is no "equilibrium". I feel the authors are confusing terms for the RBM itself with the sampling distribution of the RBM.
> >
> > The reader at this stage doesn't know what low-rank pretraining is.
> >
> > Figure 3
> >
> > I'm not sure what is intended here, but what I presume is the true distribution is almost completely obscured by the trajectories, making it hard for the reader to understand much.
> >
> > I don't understand panel C. What does "Nmodels x AGS steps/model" mean? Specifically, what does Nmodels mean here?
> >
> > "averages number"
> >
> > Page 7
> >
> > "a serie"
> >
> > Still a lack of consistency in notation and lack of care. For example equations 3 and 28.
> >
> > The authors ask the reader to refer to appendix B.1 to understand how to do PTT for the RBM. However, the appendix discusses the RCM - a different model. Again, no reader unfamiliar with this will be able to follow this non sequitur.
> >
> > I stopped re-reading the paper at page 7 since it seems that the same overall presentation issues are still present.  I would have preferred the authors to get to the point more quickly and explain to the reader the central contributions (one substantial contribution would have been sufficient) and have a clear presentation of the actual algorithms used.

---

> > > ### Author Response · Authors · 2024-12-02
> > > **Reply to "Official Comment by Reviewer jkKM" (1/2)**
> > >
> > > > *SID typo
> > > Please stick with a single terminology -- either "appendix" or "supplementary information (SI)"*
> > >
> > > We thank the reviewer for pointing out this typo. While we had attempted to unify the terminology in the revised version, we unfortunately missed two instances of 'appendix' in the text. Although we are unable to submit a new version of the manuscript at this stage of the process, we have corrected this error in our internal version.
> > >
> > > > *Figure 2:
> > > The text refers to panels in figure 2. No such panels are labelled.*
> > >
> > > We thank the reviewer for bringing this to our attention. The panels in Figure 2 have now been properly labeled.
> > >
> > > > * I'm not sure if it's fully explained in what is happening here. Is it that a method (such as AIS) is used to approximate the gradient and update the parameters, with the difference between the true LL and the approximate LL being plotted? *
> > >
> > > No, it is not. The updates of parameters is done using the PCD scheme. The evolution of the parameters during the training trajectory is used to obtain better estimates of the log-likelihood, and to sample faster the distribution, by replacing the temperature ladder scheme by different models during the trajectory.
> > >
> > > > *What is the source of the error bars in this figure? There's no mention of repeated experiments.*
> > >
> > > We thank the reviewer for pointing this out. We had repeated trainings 10 times and obtained a log likelihood curve for each. The shadow shows the standard deviation between trainings. It is now explained in the caption of our last version (we cannot longer update the pdf) with the sentence “The lines represent the average LL obtained from 10 independent training runs, while the shaded areas indicate one standard deviation.”
> > >
> > > > *Panel "A" will present to the reader an obvious question, namely why does AIS with more steps actually perform worse than AIS with fewer steps around 1000 gradient updates? It would be good if possible to relate this to a phase transition. *
> > >
> > > We agree with the referee that it is intriguing, but this is a mixed effect between the evolution of the free energy landscape during the training and the evolution of that landscape when temperature is changed. This particular ordering in the quality of the AIS is very particular to this dataset and this number of hidden nodes, and not a general feature.
> > >
> > > > * Please be more specific about the non-flat reference distribution and state in the figure caption exactly what this is.*
> > >
> > > It was defined in the main text and explicitly in SI-D. It is now briefly mentioned the caption of the last version: “AIS with a reference distribution fixed to independent site distribution that matches the empirical center of the dataset (middle)”
> > >
> > > > * Panel "C" isn't properly explained and I don't understand what is being shown here. In general I feel it is poor practice to present results out of order -- showing things that cannot be understood by the reader until further material has been revealed.*
> > >
> > > Panel C is explained in the main text just after panels A and B. The red curve is obtained with the method just proposed, and the purple line is obtained with the sampling method that is explained in the section just after that. This is a matter of taste of the reviewer, we think it is more logical to compare the PTT estimate with the rest of estimates when the LL is discussed. The comparison of AIS with estimates based on PT is common in the literature, so we do not think it is a weird connection.
> > >
> > > > * I don't feel that TR-AIS is properly explained here. Please write an algorithmic description of this approach.*
> > >
> > > The Tr-AIS method is exactly the same as the AIS method, where the different temperatures are replaced by different models saved during the training trajectory. We can of course add a review of the AIS and Tr-AIS method in the Appendix.
> > >
> > > > * What is "Alternative" or "Alternate" Gibbs sampling (the authors seem to flip between these two terms)? Do the authors mean "alternating"? Please also explain this to a non-expert in this area. Also the authors assume that there is only one way to do Gibbs sampling, whilst in practice there are many ways to implement Gibbs sampling. The authors choose the standard approach (alternating between conditioning on all hiddens and then conditioning on all visibles) but never explain this to the reader.*
> > >
> > > We thank the reviewer for identifying this spelling error, which appears to have been introduced by the spell checker. We have corrected all instances to 'alternating Gibbs sampling.' Additionally, we have included the sentence: 'The AGS procedure involves iteratively alternating between two steps: conditioning on all hidden variables given fixed visible variables, and then conditioning on all visible variables given fixed hidden variables,' immediately after the first occurrence of the acronym.

---

> > > > ### Author Response · Authors · 2024-12-02
> > > > **Reply to "Official Comment by Reviewer jkKM" (2/2)**
> > > >
> > > > > * I don't understand what "equilibrium measure of RBMs" means. The RBM is a defined distribution -- there is no "equilibrium". I feel the authors are confusing terms for the RBM itself with the sampling distribution of the RBM. *
> > > >
> > > > We agree with the reviewer that the term 'equilibrium measure' should not be necessary. However, the literature is full of authors generating with RBMs using non-convergent MCMC processes, often involving extremely short sampling runs. In statistical physics, the distinction between studying the Boltzmann distribution and examining the dynamical properties of an out-of-equilibrium process with MCMC is emphasized through the use of the term 'equilibrium'. By adopting this terminology, we aim to highlight the importance of focusing on the correct underlying distribution rather than on non-equilibrium dynamics.
> > > >
> > > > > * The reader at this stage doesn't know what low-rank pretraining is.*
> > > >
> > > > The pretraining is briefly introduced in the paragraph just before section 4 and for a more detailed discussion, the reader is referred to section 6.
> > > >
> > > > > * Figure 3
> > > > I'm not sure what is intended here, but what I presume is the true distribution is almost completely obscured by the trajectories, making it hard for the reader to understand much.*
> > > >
> > > > We present two independent Markov chains to demonstrate that the PTT method can ergodically sample the phase space, whereas the AGS method cannot.
> > > >
> > > > > * I don't understand panel C. What does "Nmodels x AGS steps/model" mean? Specifically, what does Nmodels mean here?*
> > > >
> > > > This is defined in the main-text, the number of models simulated in parallel by each algorithms. For AGS Nmodels=1 because just one model is simulated, for PT it's the number of temperatures, and for ST or PTT the number of RBM models used for each algorithm.
> > > >
> > > >
> > > > > * "averages number"
> > > > Page 7
> > > > "a serie"*
> > > >
> > > > both corrected, thanks.
> > > >
> > > > > * Still a lack of consistency in notation and lack of care. For example equations 3 and 28.*
> > > >
> > > > We are not sure to understand this comment. Both equations are the same. We suppose the referee find the explanation not clear enough, it seems that the phrase $\Delta \mathcal{H}_t(\bm{x}) = \mathcal{H}_t(\bm{x}) - \mathcal{H}_{t-1}(\bm{x})$ got deleted in one of the updates by error. It is now included in Eq. (3).
> > > >
> > > > > * The authors ask the reader to refer to appendix B.1 to understand how to do PTT for the RBM. However, the appendix discusses the RCM - a different model. Again, no reader unfamiliar with this will be able to follow this non sequitur.*
> > > >
> > > > Section SI B.1 is “B.1 PSEUDO-CODE OF PTT VS PT” and the RCM is not mentioned there. We suppose the referee refers to section B where the use of the low-rank RBM (sometimes referred in short as RCM in the paper). We will revise the paper to avoid using this naming to avoid confusion.
> > > >
> > > > > * I stopped re-reading the paper at page 7 since it seems that the same overall presentation issues are still present. I would have preferred the authors to get to the point more quickly and explain to the reader the central contributions (one substantial contribution would have been sufficient) and have a clear presentation of the actual algorithms used.*
> > > >
> > > > The proposed algorithms, along with the explanation of why commonly used algorithms often fail, are grounded in the physical characterization of the phases encountered during training. To the best of our knowledge, such explanations have not been previously discussed in the context of RBM training, and we believe they offer valuable insights into the underlying dynamics of the training process.

---

### Author Response · Authors · 2024-11-16
**General answer for all the reviewers**

We thank the reviewers for their careful comments and valuable suggestions. Several reviewers inquired specifically about which methods in our work are novel contributions. We would like to address this shared concern collectively and will also make every effort to clearly highlight these contributions in the final version.

The new contributions of this work are as follows:

- We build on recent advances in understanding the phase transitions and phases encountered during RBM training to establish a general framework for analyzing the dynamical behavior of MCMC algorithms used in both training and generation. This framework provides insights into why certain widely-used algorithms in the literature—both for sampling and for estimating the partition function—succeed or fail. Leveraging these insights, we propose new methods for training, evaluation, and sampling, as outlined in the following points.

- We design a pre-training strategy based on the mapping between the RBM and the Coulomb machine proposed in the [D\&F 2021] paper, significantly extending the applicability of this method by adjusting various components to make it suitable for real data. We outline these improvements as follows. First, while D\&F’s work focuses primarily on low-dimensional synthetic datasets with specific modes and regular features, our work shifts from theory to practical application, requiring numerous detailed adjustments to broaden the technique’s applicability. Notably, we extend the technique to handle up to four intrinsic dimensions, including a specially treated bias direction; the previous method was limited to two dimensions. This bias adjustment is essential for processing image data, as it enables image generation with low-rank models. Additionally, we correct D\&F’s entropy calculation to allow true equilibrium samples of the low-rank RBM to be obtained via a static Monte Carlo procedure, which is crucial for efficient sampling of the trained machines. These enhancements for real data applicability will be highlighted in the revised version.

- We introduce a novel framework for estimating log-likelihood (LL) by leveraging the learning trajectory's softness, rather than relying on temperature integration. This approach allows for a reliable, cost-effective LL estimation either online during training or, alternatively, after training by simply saving the model parameters at various stages. We validate this new LL estimation method by comparing it to exact LL values obtained through exhaustive state enumeration in controlled training scenarios using RBMs with a few hidden nodes trained on real data. The results demonstrate an unprecedented level of accuracy relative to standard methods, particularly when applied to highly structured datasets.

- We propose a variation of the standard parallel tempering algorithm in which exchanges occur between the parameters of models trained at different stages, rather than across temperatures. We demonstrate that this new algorithm significantly accelerates simulations—achieving speed improvements by several orders of magnitude compared to standard alternate Gibbs sampling—and also provides substantial speed gains over optimized methods such as parallel tempering and the recently proposed Stack Tempering algorithm (ICLR 2023).

We have included a list of bullet points at the end of Section 1 to highlight our contributions and direct the reader to the relevant sections, as suggested by Reviewer Axdy. Given these points, we believe that concerns about a lack of novelty in our work are unfounded. RBMs are notoriously challenging to train in the equilibrium regime of interest (as discussed in the paper), primarily due to the slow convergence of Monte Carlo gradient estimation. Our new sampling method offers a significant advancement for RBM training by enabling efficient exploration across diverse clusters. This approach is rooted in well-identified phenomena and introduces a novel method for evaluating the partition function, an exceptionally challenging task. Our results demonstrate that this method outperforms previous state-of-the-art techniques in this context.

We have produced a new version of the manuscript that attempts to address almost all of your comments and concerns. We will try to complete the remaining comments later this week.

---

### Author Response · Authors · 2024-11-26
**General summary of the modifications**

Dear reviewers,

In addition to all the answers to your questions, and considering that the rebuttal time is reaching to an end, we would like to give you a brief summary of the various changes we have made to the article during this period.

* We have added a short section after the introduction describing our various contributions, paragraph line 92.

* As suggested, we have moved the "Pre-training" section to the end of the main text so that we focus more on the trajectory sampling perspective and the contributions on the log-likelihood estimation and the PTT algorithm.

* We have reordered the presentation of the sampling methods to highlight our contribution and the pitfalls of previous methods (paragraph line 316).

* We have added to paragraph line 416 (failure of the thermal PT algorithm) an explanation of why the trajectory PT performs better and illustrated the phenomenon of discontinuous transition in the case of the genetic dataset, which we have added in Fig. 4, A-B. In this last figure, we clearly show the discontinuous transition and compare the results between PTT and PT, noting that the latter misses a small cluster when it tries to capture the model's equilibrium distribution.

We also added
* appendix B.1. to describe our algorithm with pseudocode;
* appendix H: discuss a simple example where we can show exactly that a simple RBM (a hidden Gaussian node learns on a simple dataset) exhibits a first-order transition when the temperature of the  learned machine is changed. We also add more details about the first order phase transition observed on the HGD dataset in this section.

---

### Meta-Review · Area_Chair_SE3x · 2024-12-29

**Metareview:**

The paper introduces a pretraining strategy for RBMs that enables better coverage of all the modes of a target density and more accurate partition function estimation. The paper also proposes a new sampling algorithm that outperforms existing MCMC algorithms. Experimental results demonstrate the validity of these claims.

**Additional Comments On Reviewer Discussion:**

The authors provided a rebuttal, engaging with all the reviewers, and thoroughly addressed their comments and questions.

---

### Decision · Program_Chairs · 2025-01-22

Accept (Poster)